# Online images amplify gender bias

Douglas Guilbeault[1✉], Solène Delecourt[1], Tasker Hull[2], Bhargav Srinivasa Desikan[3], Mark Chu[4] & Ethan Nadler[5]

Each year, people spend less time reading and more time viewing images[1], which are proliferating online[2–4]. Images from platforms such as Google and Wikipedia are downloaded by millions every day[2,5,6], and millions more are interacting through social media, such as Instagram and TikTok, that primarily consist of exchanging visual content. In parallel, news agencies and digital advertisers are increasingly capturing attention online through the use of images[7,8], which people process more quickly, implicitly and memorably than text[9–12]. Here we show that the rise of images online significantly exacerbates gender bias, both in its statistical prevalence and its psychological impact. We examine the gender associations of 3,495 social categories (such as 'nurse' or 'banker') in more than one million images from Google, Wikipedia and Internet Movie Database (IMDb), and in billions of words from these platforms. We find that gender bias is consistently more prevalent in images than text for both female- and male-typed categories. We also show that the documented underrepresentation of women online[13–18] is substantially worse in images than in text, public opinion and US census data. Finally, we conducted a nationally representative, preregistered experiment that shows that googling for images rather than textual descriptions of occupations amplifies gender bias in participants' beliefs. Addressing the societal effect of this large-scale shift towards visual communication will be essential for developing a fair and inclusive future for the internet.

Images increasingly pervade the information we consume and communicate daily. The number of images in online search engines has leapt from thousands to billions in just two decades[2]. Every day, millions of people view and download images from platforms such as Google and Wikipedia[5,6], and millions more are socializing through hyper-visual platforms such as Instagram, Snapchat and TikTok, which are based predominantly on the exchange of images. This growing trend is widely recognized by the tech and venture capital industries[3,4], as well as by news agencies and advertisers who are now relying more heavily on images to attract people's attention online[7,8]. This trend is also reflected by changes in the habits of the average American. A longitudinal survey from the American Academy of the Arts and Sciences shows that the amount of time Americans spend reading text is steadily declining[1], whereas the time they spend producing and viewing images continues to rise[2,4]. What consequences does this unprecedented shift towards visual content have on how we 'see' the world? At the dawn of photography, Frederick Douglass—esteemed writer and civil rights leader—forewarned of the potential for images to reinforce social biases at large, arguing in his 1861 lecture 'Pictures and Progress' that "the great cheapness and universality of pictures must exert a powerful though silent influence on the ideas and sentiment of present and future generations"[19]. Since Douglass' time, the internet has made it only cheaper and easier to circulate images on a massive scale[3,4], potentially intensifying the impact of their silent influence. In this study, we explore the impact of online images on the large-scale spread of gender bias.

Despite the swelling proliferation of online images, most quantitative research into online gender bias focuses on text[13,15,20–22]. Only a few recent studies examine gender bias in a small sample of Google images[16–18], without comparing the prevalence of gender bias and its psychological impact across images and text. Yet numerous psychological studies suggest that images may provide an especially potent medium for the transmission of gender bias. Research into the 'picture superiority effect' shows that images are often more memorable and emotionally evocative than text[9,10,23], and may implicitly underlie the comprehension of text itself[11,12,24,25]. Images also differ from text in the salience with which they present demographic information. A textual description of a person can easily minimize gender bias by leveraging gender-neutral terminology or by omitting references to gender. For example, the sentence 'The doctor administered the test' makes no mention of the doctor's gender. By contrast, an image of a doctor directly transmits demographic cues that elicit perceptions of the doctor's gender. In this way, images strengthen the salience of gender in the representation of social categories. These intrinsic differences between images and text point to the prediction that online images amplify gender bias, both in its statistical prevalence and in its psychological impact on internet users.

## Comparing gender bias in images and text

In this study, we developed computational and experimental techniques for comparing gender bias and its psychological impact across massive online corpora of images and texts. Our main analyses compared images and text data from the world's most popular search engine, Google. Our findings were replicated using more than half a million images

[1]Haas School of Business, University of California, Berkeley, Berkeley, CA, USA. [2]Psiphon Inc., Toronto, Ontario, Canada. [3]Institute For Public Policy Research, London, UK. [4]School of the Arts, Columbia University, New York, NY, USA. [5]Department of Physics, University of Southern California, Los Angeles, CA, USA. ✉e-mail: douglas.guilbeault@haas.berkeley.edu

and billions of words from Wikipedia and Internet Movie Database (IMDb)[26–28] (Extended Data Figs. 1 and 2; see Supplementary Information sections A.1.1 and A.1.2 for details). We implemented our model at scale by examining the gender biases in images and texts associated with all 3,495 social categories drawn from WordNet, a canonical database of categories in the English language[29]. These categories include occupations—such as doctor, lawyer and carpenter—and generic social roles, such as neighbour, friend and colleague.

To measure gender bias in online images, we automatically retrieved the top 100 images from Google corresponding to each social category in Google Images (Extended Data Fig. 3; see 'Data collection procedure for online images' in Methods). Collecting 100 images for 3,495 categories yielded 349,500 images. In the Supplementary Information, we report analyses showing that our results held when we increased the number of images collected for each category, and when we used gender-specific Google searches for each category (for example, female doctor), which yielded an extra 491,169 images (Supplementary Figs. 1 and 2). The scale of our image dataset is orders of magnitude larger than prior studies of gender bias in Google Images, which have typically examined 50 occupations or fewer, using only a few thousand images in total[16–18]. Each search was implemented from a fresh Google account with no prior history to avoid the uncontrolled effects of Google's recommendation algorithm, which customizes results based on browsing history[30]. Searches were run by ten distinct data servers in New York City. All image data were collected in August 2020. Our results were replicated when collecting Google images using Internet Protocols from five further locations around the world: Amsterdam (the Netherlands), Bangalore (India), Frankfurt (Germany), Singapore (Singapore) and Toronto (Canada) (Supplementary Figs. 3 and 4).

To identify the gender of faces in each image, we hired a team of 6,392 human coders from Amazon Mechanical Turk (MTurk). The gender of each face was determined by identifying the majority (modal) gender classification selected by three unique coders who labelled faces as 'female', 'male' or 'non-binary' (2% of classification judgements indicated 'non-binary'; these were excluded from our analyses). Our focus is not on how people self-identify in terms of gender. Rather, we focus on the gender that internet users perceive in online images. We replicated our findings using a canonical image dataset[28] of 72,214 celebrities depicted across IMDb and Wikipedia (511,946 images), where each image is associated with the self-identified gender of the person depicted (Extended Data Fig. 2 and Supplementary Information section A.1.2). All coders were fluent English speakers based in the USA, and our results are robust to controlling for coder demographics and the rate of intercoder agreement (Supplementary Tables 1 and 2; see 'Demographics of human coders' in Methods). Coders reached unanimous agreement in their gender classifications for 91% of images. A standard chance-corrected measure of classification agreement (Gwet's Agreement Coefficient, AC) indicates satisfactory intercoder reliability in our sample (Gwet's AC1 = 0.48). For each category, we calculated the gender balance of the faces in its top 100 Google Image search results. We normalized this measure such that −1 indicates 100% female representation, 0 indicates perfect gender balance (50%/50%) and 1 indicates 100% male representation.

To measure gender bias in online texts, we leveraged word embedding models that construct a high-dimensional vector space based on the co-occurrence of words (for example, whether two words appear in the same sentence), such that words with similar meanings are closer in this vector space. Harnessing recent advances in natural language processing[22,31], we identified a gender dimension in word embedding models that captures the extent to which each category co-occurs with textual references to either women or men. This method allows us to position each category along a −1 (female) to 1 (male) axis, such that categories closer to −1 are more commonly associated with women and those closer to 1 are more commonly associated with men (see 'Constructing a gender dimension in word embedding space' in

Methods). We focus here on applying this method to the canonical word2vec model[32] trained on the 2013 Google News corpus consisting of more than 100 billion words. Our results hold when comparing against our own word2vec model trained on a more recent sample of online news published between 2021 and 2023 (Extended Data Fig. 4). We also replicated our findings when comparing online images with a range of word embedding models, including Global Vectors for Word Representation (GloVe), Bidirectional Encoder Representations from Transformers (BERT), FastText, ConceptNet and Generative Pre-trained Transformer 3 (GPT-3), which vary in their dimensionality, their data sources (including Twitter and a random sample of the web) and the time period during which their training data were collected, ranging from 2013 to 2023 (Supplementary Table 3 and Supplementary Fig. 5).

Both our image-based and text-based measures capture the frequency with which each social category co-occurs with representations of each gender, along a −1 (female) to 1 (male) continuum, where 0 indicates equal association with each gender. To maximize the correspondence between our image-based and text-based measures, we apply minimum–maximum normalization to our text-based measure, so that −1 and 1 represent the most female and male categories, respectively, according to each method (results are robust to alternative normalization procedures; Supplementary Fig. 6). We were able to associate 2,986 social categories in WordNet with word embeddings in the Google News corpus, so we focus our comparisons on these categories (our image results are robust to including all 3,495 categories; Supplementary Fig. 7).

Using these measures, we quantify gender bias as a form of statistical bias along three dimensions. First, we examine the extent to which social categories are associated with a specific gender in images and texts. Second, we examine the extent to which women are represented, compared with men, across all social categories in images and texts. Third, we compare the gender associations in our image and text data with the empirical representation of women and men in public opinion and US census data on occupations. This allows us to test not only whether gender bias is statistically stronger in images than texts, but also whether this bias reflects a distorted representation of the empirical distribution of women and men in society.

## Gender bias is stronger in images

To begin, we confirm that the gender associations for each social category are highly correlated across online images (Google Images) and texts (Google News) ($P < 0.0001$, $r = 0.5$, Fig. 1a, Pearson correlation, two-tailed, $n = 2,986$ categories), indicating shared patterns of gender representation across these sources. Yet the gender associations in images from Google Images are statistically more extreme than those in texts from Google News. Figure 1b shows that the magnitude of gender bias is significantly stronger in images than text for both female-skewed ($P < 0.0001$) and male-skewed categories ($P < 0.0001$) (Wilcoxon signed-rank test, $n = 2,986$ categories, two-tailed). This result holds when comparing only categories for which the gender associations agree across images, texts and human judgements (Extended Data Fig. 5). Figure 1c highlights this gap by showing the gender associations in these images and texts for an illustrative sample of occupations.

Yet we also find that, on average, women are underrepresented in images, compared with texts (Fig. 2). Figure 2a shows that texts from Google News exhibit a relatively weak bias towards male representation (average bias ($\mu$) = 0.03, $P < 0.0001$), whereas this male bias is more than four times stronger in images from Google Images ($\mu = 0.14$, $P < 0.0001$), marking a highly significant increase (mean difference = 0.11, $P < 0.0001$) (Wilcoxon signed-rank test, two-tailed, $n = 2,986$ categories). According to Google News, 56% of categories are male-skewed, whereas 62% are male-skewed according to Google Images ($P < 0.0001$, proportion test, two-tailed, $n = 2,986$ categories). The underrepresentation of women is accentuated when using a deep learning algorithm to classify gender in these online images

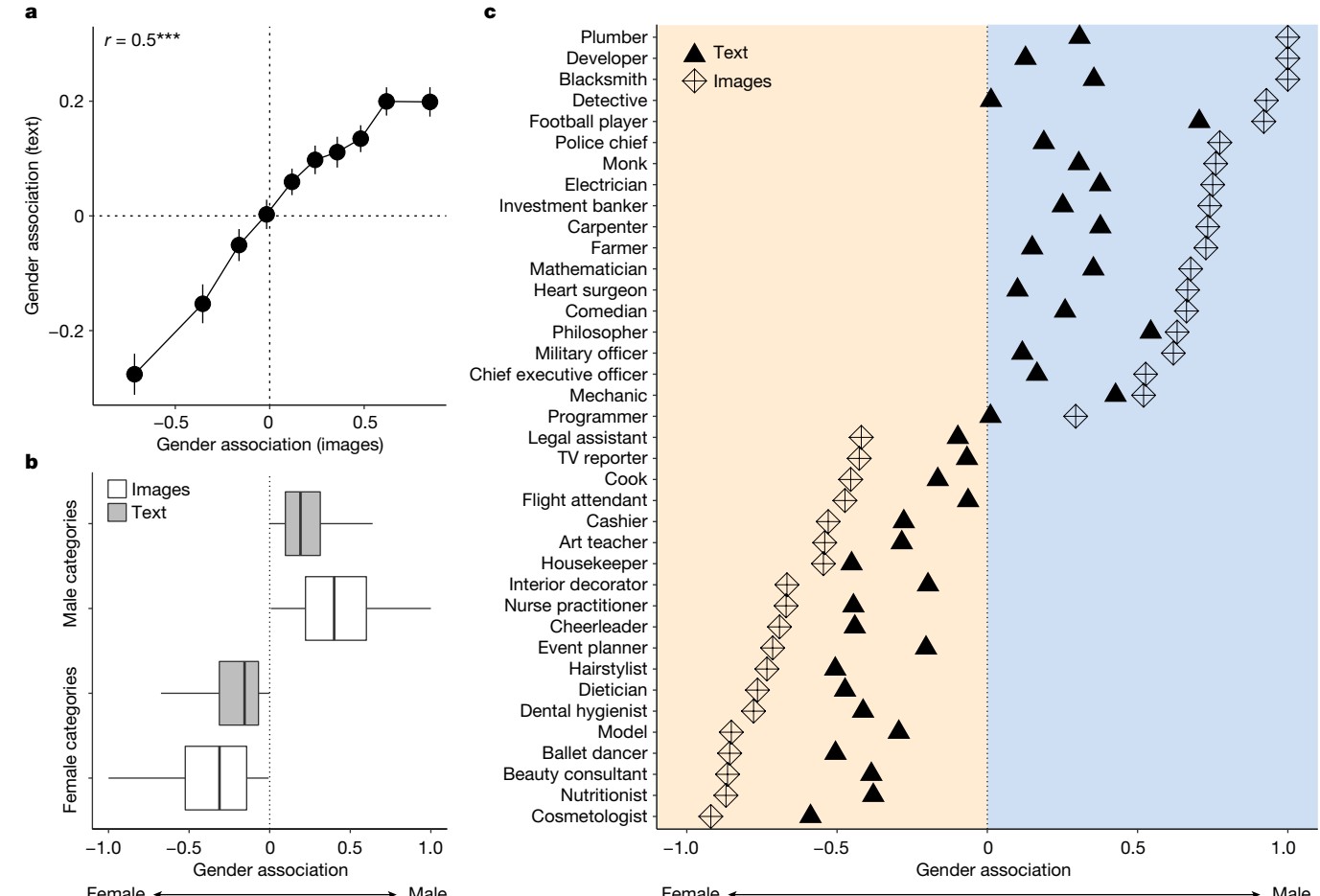

**Fig. 1 | Gender bias is more prevalent in online images (from Google Images) and online texts (from Google News) for both male- and female-typed social categories. a**, The correlation between gender associations in images from Google Images and texts from Google News for all social categories (*n* = 2,986), organized by deciles. Our image-based measure captures the frequency of female and male faces associated with each category in Google Images (−1 means 100% female; 1 means 100% male). Our text-based measure captures the frequency at which each category is associated with men or women in the Google News corpus (−1 means 100% female; 1 means 100% male; measure is minimum–maximum normalized; 'Constructing a gender dimension in word embedding space'). Data are shown as mean values, and error bars represent 95% confidence intervals. ***P* = 2.2 × 10⁻¹⁶ (Pearson correlation, two-tailed). **b**, The strength of gender association in these online images and texts for all categories (*n* = 2,986), split into whether these categories are female- or male-skewed according to each measure separately. Box plots show interquartile range (IQR) ±1.5 × IQR. **c**, The gender associations for a sample of occupations according to these online images and texts; this sample was manually selected to highlight the kinds of social categories and gender biases examined.

(Supplementary Figs. 8–10). This inequality even persists when searching explicitly for 'female' and 'male' images of each category in Google (Supplementary Figs. 1 and 2).

Our findings continue to hold when controlling for (1) linguistic features of categories, such as ambiguity, word frequency and gender connotation (for example, uncle) (Supplementary Figs. 11 and 12 and Supplementary Table 4); (2) the method for constructing the gender dimension in embedding space (Supplementary Figs. 6 and 13–15); (3) the frequency at which each category is searched in Google Images across the USA (Supplementary Figs. 16 and 17 and Supplementary Table 5); (4) the number of faces (Supplementary Fig. 18) and images (Supplementary Fig. 19) associated with each category, and the number of categories examined (Supplementary Fig. 7); (5) the ranking of images in Google search results (Supplementary Fig. 19 and Supplementary Table 7); (6) whether faces are automatically cropped from images before they are classified by human annotators (Supplementary Fig. 20) or a deep learning classifier (Supplementary Fig. 9); (7) whether images repeat in and across searches (Supplementary Table 8); (8) the number of faces associated with each Google search (Supplementary Table 7); and (9) whether images contain photographed or animated people (Supplementary Table 8).

Although these analyses support our prediction that online gender bias is more prevalent in images than texts, an open question is whether online images present a biased representation of the empirical distribution of gender in society. Next, we show that online images exhibit significantly stronger gender bias than public opinion and 2019 US census data on occupations.

To compare our results with public opinion, we hired a separate panel of 2,500 coders from MTurk who used the same −1 (female) to 1 (male) scale to provide their opinions about the gender they most associate with each category in our dataset (see 'Collecting human judgements of social categories' in Methods). Although both our image and text measures are highly predictive of gender associations in public opinion, Fig. 2b shows that texts significantly underestimate male bias in public opinion (by −0.084 on average, *P* < 0.001), whereas images significantly overestimate it (by 0.025 on average, *P* < 0.001) (Wilcoxon signed-rank test, two-tailed, *n* = 2,986 categories).

We also compare our measures with the frequency of genders across occupations according to the 2019 census by the US Bureau of Labor Statistics (*n* = 685 occupations could be matched between our data and the census). Figure 2c shows that, according to texts from Google News, the gender association of these occupations is neutral (*μ* = 0, *P* = 0.65)

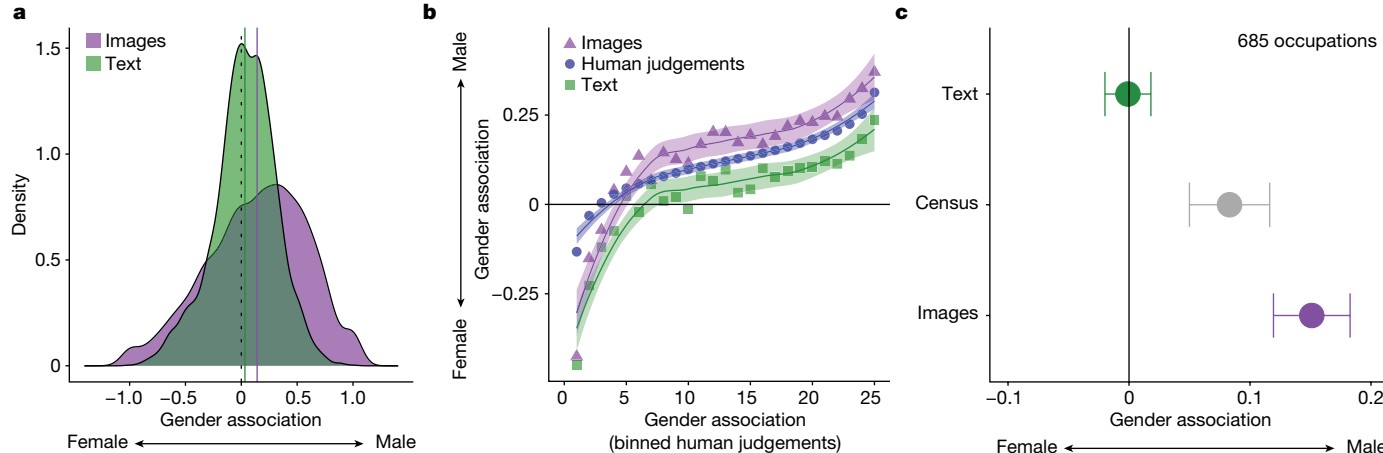

**Fig. 2 | The underrepresentation of women is stronger in online images (from Google Images) than in online texts (from Google News), public opinion and US census data on occupations. a**, The distribution of gender associations for social categories ($n = 2,986$) in images from Google Images and texts from Google News. The image-based measure captures the frequency of female and male faces associated with each category in Google Image search results (−1 means 100% female; 1 means 100% male); the text-based measure captures the frequency with which each category is associated with men or women in the Google News corpus (−1 means 100% female; 1 means 100% male associations). The solid lines indicate the average gender association according to text (green) and images (purple). **b**, The correlation of gender associations, paired at the category level ($n = 2,986$), as measured by these online images and texts, as well as by internet users' ($n = 2,500$) judgements of each category. Human coders indicated their beliefs about the gender representation of each category by moving a slider along the same −1 (female) to 1 (male) scale (horizontal axis shows the average human judgement across evenly spaced bins). Data points show mean values for each bin, and error bands show 95% confidence intervals for the fitted curve defined by a locally estimated scatterplot smoothing (LOESS)-smoothed regression (span = 0.75). **c**, The gender association of all matched occupations ($n = 685$) according to (1) textual patterns in Google News (green), (2) the empirical distribution of gender in the 2019 US census Bureau of Labor Statistics (grey) and (3) Google Images (purple). Data are shown as mean values and error bars show 95% confidence intervals calculated using a Student's $t$-test (two-tailed).

and significantly less male than the census (census $\mu = 0.08$, $P < 0.001$) and Google Images (images $\mu = 0.15$, $P < 0.001$) (Wilcoxon signed-rank test, two-tailed, $n = 685$ occupations). By contrast, although these occupations are male-skewed in both the census and Google Images, the same occupations are significantly more biased towards male representation in Google Images (mean difference = 0.07, $P < 0.001$, Wilcoxon signed-rank test, two-tailed, $n = 685$ occupations). Comparing images and texts separately for female- and male-typed occupations reinforces these findings (Supplementary Fig. 21).

## Testing psychological effects of images

What consequences do these biases in online images have on internet users? Here we report the results of a preregistered experiment designed to test the impact of online images on gender bias in people's beliefs ('Data availability'). In this experiment, we recruited a nationally representative sample of US participants from the online platform Prolific ($n = 450$), who were tasked with using Google to search for descriptions of occupations relating to science, technology and the arts (Extended Data Fig. 6; see 'Participant pool' in Methods). A total of 423 participants completed the task. Each participant used Google to retrieve descriptions of 22 randomly selected occupations from a set of 54 (see 'Participant experience' in Methods). Participants were randomized into either (1) the Text condition, in which participants used Google News to search for and upload textual descriptions of these occupations, or (2) the Image condition, in which participants used Google Images to search for and upload images of occupations. After uploading the description for each occupation, each participant was asked to rate which gender they most associate with the occupation being described, using a −1 (female) to 1 (male) scale. To evaluate these experimental effects, participants were also randomized into the Control condition that used the same task design, except that participants used Google to search for and upload either images or textual descriptions of basic, unrelated categories (for example, apple and guitar) before rating the gender they associate with each occupation.

In the Supplementary Information, we report the results of an extra condition in which a separate randomized group of participants were tasked with searching for textual descriptions using the generic Google search bar rather than the Google News search bar; altering the search bar had no effect on the outcomes (Supplementary Fig. 22). Across all conditions, our main outcome variable of interest is the absolute strength of participants' gender associations for each occupation.

After completing the search task for all occupations, participants undertook an implicit association test (IAT)[33], a standard method in psychology for detecting implicit biases (see 'Measuring implicit bias using the IAT' in Methods). We adopted an IAT designed to detect the implicit bias towards associating women with liberal arts and men with science (Extended Data Figs. 7 and 8), because prior work demonstrates the ability of this IAT to predict human judgements and behaviours[34,35] relating to a consequential pattern of inequality in industry and academic institutions[36,37]. We administered the IAT to participants immediately after the experiment, and 3 days later. Participants' implicit bias was measured using the standard IAT $D$ score[33]; positive $D$ scores indicate that participants are faster at associating women with liberal arts and men with science. We acknowledge important continuing debate about the reliability of the IAT[38–40]. Our specific choice of IAT is supported by (1) prior work demonstrating its stable results across decades[34] and (2) a separate preregistered study we conducted that yielded consistent results with a similar design (Methods). We note, however, that the distribution of participants' implicit bias scores was less stable across our preregistered studies than the distribution of participants' explicit bias scores. Given these considerations, we view our implicit bias results as suggestive and emphasize our measure of participants' explicit bias as our primary and most robust outcome of interest.

We begin by examining the extent of gender bias in the descriptions participants uploaded. A team of annotators labelled each textual description as female, male or neutral on the basis of whether it used female or male pronouns or names to describe the occupation (for example, a description referring to a 'doctor' as 'he' would be coded as 'male'); textual descriptions were identified as neutral if they did

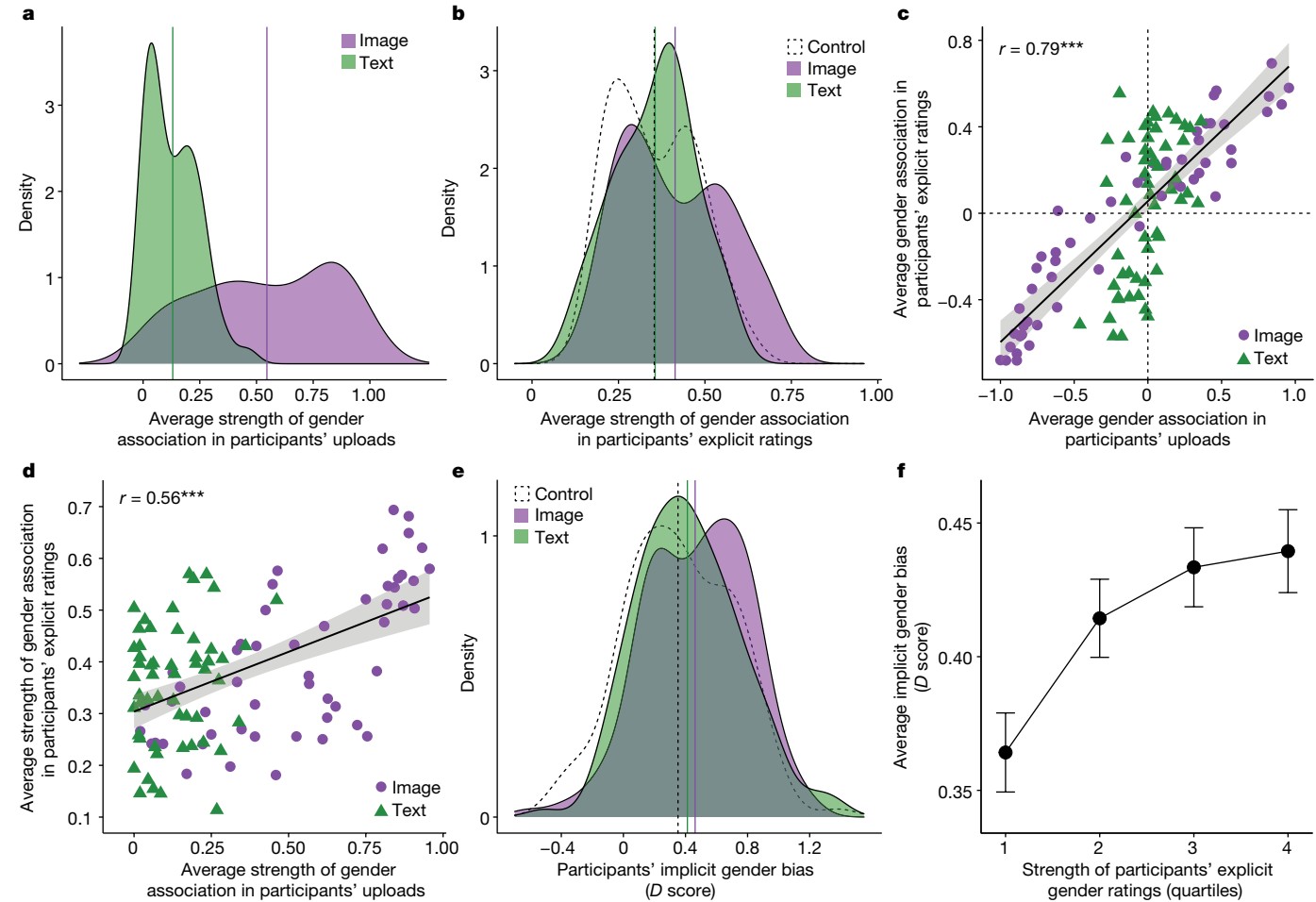

**Fig. 3 | Googling for images rather than textual descriptions of occupations amplifies gender bias in people's beliefs. a–f,** Participants (*n* = 423) from a nationally representative sample were randomized to one of the following: the 'Image' condition, in which they googled for images of occupations; the 'Text' condition, in which they googled for textual descriptions of occupations from Google News; or the 'Control' condition, in which they googled for either image-based or text-based descriptions of random categories (for example, 'apple') unrelated to occupations. The green, purple and dotted vertical lines indicate the mean results for the Text, Image and Control conditions, respectively. **a,** The average absolute strength of the gender associations in participants' uploads for each occupation (*n* = 54; averaged at the occupation level) in both the Text and Image conditions (not applicable to the Control condition). **b,** The average absolute strength of the gender associations that participants reported for each occupation (*n* = 54; averaged at the occupation level) in each condition. **c,** The linear correlation between the average gender association of the descriptions that participants uploaded and the average gender association they explicitly reported for each occupation, coloured by condition. ***$P = 2.2 \times 10^{-16}$ (Pearson correlation, two-tailed). **d,** The correlation between the average strength of the gender association of the descriptions that participants uploaded and the average strength of the gender association they explicitly reported for each occupation, coloured by condition. ***$P = 6.2 \times 10^{-11}$. **e,** The implicit gender bias (*D* score) that participants (*n* = 405) exhibited in each condition. **f,** The correlation between the strength of participants' self-reported gender associations for each occupation and their implicit bias (*D* score) towards associating women with liberal arts and men with science (*n* = 9,167 observations across all participants). Error bars show 95% confidence intervals calculated using a Student's *t*-test (two-tailed).

not ascribe a particular gender to the occupation. Similarly, a team of annotators labelled the gender of the focal face in each uploaded image as female, male or neutral; images were coded as neutral if they contained no face or an undecipherable face. Then, for each occupation, we calculated the gender balance of the descriptions provided by participants by computing the average gender association across all descriptions. This approach compares gender associations across images and texts without relying on word embedding models, while also ensuring that the images and texts being compared were collected by users during the same time period.

## Images amplify explicit gender bias

Consistent with our observational results, Fig. 3a shows that the descriptions participants uploaded were significantly more gendered in the Image condition than in the Text condition (mean difference = 0.42,

*P* < 0.0001, Wilcoxon signed-rank test, two-tailed). Figure 3b shows that exposure to more gendered stimuli in the Image condition led participants to report significantly stronger explicit gender associations than those in the Text (mean difference = 0.06, *P* < 0.001) and Control (mean difference = 0.06, *P* < 0.001) conditions, whereas there was no significant difference between those in the Text and Control conditions (mean difference = 0.001, *P* = 0.56) (Wilcoxon signed-rank test, two-tailed; Wilcoxon equivalence test, *P* < 0.05 for all bounds greater than or equal to |0.11|, *n* = 54 occupations). For example, participants in the Text condition rated the category 'model' as female-skewed ($\mu = -0.32$), but the female-skew of this rating nearly doubled in its intensity among participants in the Image condition ($\mu = -0.62$). These findings hold when controlling for the number of online sources that participants encountered, the amount of time they spent evaluating descriptions and participants' gender (Supplementary Fig. 23 and Supplementary Tables 9–11). Notably, the gender associations in

participants' uploads and self-reported beliefs are highly correlated with the gender associations detected for the same occupations in our observational analyses of Google Images and textual data from Google News (Extended Data Fig. 9).

## Images prime gender bias more strongly

These findings suggest that exposure to gendered descriptions in the Image condition more strongly primed participants' explicit gender ratings of occupations. This priming mechanism is supported by Fig. 3c, which shows a high correlation between the gender associations in the descriptions that participants uploaded and the gender associations in their own explicit gender ratings across occupations ($r = 0.79$, $P < 0.0001$), and by Fig. 3d, which shows a strong correlation between the absolute strength of gender associations in participants' uploads and the absolute strength of the average gender associations they explicitly reported across occupations ($r = 0.56$, $P < 0.0001$) (Pearson correlation, two-tailed, $n = 54$ occupations). These results hold across occupations for both the Image and the Text conditions (Supplementary Table 12).

We found further evidence suggesting that images differ from text not only in the prevalence of gender bias they contain, but also in their ability to prime gender bias in people's beliefs, holding prevalence constant (Extended Data Fig. 10). Participants who uploaded gendered images explicitly reported significantly stronger gender bias ($\mu = 0.41$) than those who uploaded gendered textual descriptions of the same occupations ($\mu = 0.35$; mean difference = 0.06, $P < 0.0001$, $t = 4.58$, Student's $t$-test, two-tailed, $n = 54$ occupations). This holds even when controlling for the amount of gender bias in the distribution of images and texts to which participants were exposed (Supplementary Table 13). Thus, even when gender was salient in both text and images, exposure to images led to stronger bias in people's self-reported beliefs about the gender of occupations.

## Images amplify implicit gender bias

Finally, we report suggestive results indicating that extended exposure to online images may have also amplified participants' implicit gender bias. Participants across all conditions exhibited significant implicit bias towards associating men with science and women with liberal arts ($P < 0.0001$ in all conditions, Wilcoxon signed-rank test, two-tailed, $n = 423$). Yet Fig. 3e shows that participants in the Image condition exhibited stronger implicit bias. There was no significant difference between participants' implicit bias in the Text and Control conditions (mean difference = 0.06, $P = 0.24$, Wilcoxon rank-sum test, two-tailed; Wilcoxon equivalence test, $P < 0.05$ for all bounds greater than or equal to |0.13|). However, participants in the Image condition exhibited significantly stronger implicit bias than those in the Control condition ($P = 0.005$) (mean difference = 0.11, Wilcoxon rank-sum test, two-tailed). The difference in implicit bias between the Image and Text conditions did not reach conventional statistical significance (mean difference = 0.05, $P = 0.09$, Wilcoxon rank-sum test, two-tailed; Wilcoxon equivalence test, $P < 0.05$ for bounds greater than or equal to |0.14|). Across conditions, we find a clear correlation between the strength of participants' self-reported gender associations and the strength of their implicit gender bias, both of which are greater in the Image condition (Fig. 3f; $P < 0.0001$, Jonckheere–Terpstra test = 19,382,281, two-tailed); this result is robust to a range of statistical controls (Supplementary Table 14). Notably, only participants in the Image condition exhibited significantly stronger implicit bias than control participants 3 days after the experiment (Supplementary Table 15), indicating enduring effects.

## Conclusion

The rise of images in popular internet culture may come at a critical social cost. We have found that gender bias online is more prevalent and more psychologically potent in images than text. The growing centrality of visual content in our daily information diets may exacerbate gender bias by magnifying its digital presence and deepening its psychological entrenchment. This problem is expected to affect the well-being of, social status of and economic opportunities for not only women, who are systematically underrepresented in online images, but also men in female-typed categories such as care-oriented occupations[41,42].

Our findings are especially alarming given that image-based social media platforms such as Instagram, Snapchat and TikTok are surging in popularity, accelerating the mass production and circulation of images. In parallel, popular search engines such as Google are increasingly incorporating images into their core functionality, for example, by including images as a default part of text-based searches[43]. Perhaps the apex of these developments is the widespread adoption of text-to-image artificial intelligence (AI) models that allow users to automatically generate images by means of textual prompts, further accelerating the production and circulation of images. Current work identifies salient gender and racial biases in the images that these AI models generate[44], signalling that they may also intensify the large-scale spread of social biases. Consistent with related studies[45], our work suggests that gender biases in multimodal AI may stem in part from the fact that they are trained on public images from platforms such as Google and Wikipedia, which are rife with gender bias according to our measures.

A promising direction for future research is to investigate the social and algorithmic processes contributing to bias in online images, pertaining not only to gender, but also to race and other demographic dimensions. The Google images we examine stem from various sources, with the most common source being personal blogs, followed by business, news and stock photo websites (Supplementary Fig. 24). The gender bias we observe seems to be driven partly by content that internet users choose to display on their blogs, and also by audiences' preferences for which news to consume or images to purchase. Our supplementary results regarding celebrities on IMDb and Wikipedia (Extended Data Fig. 2) reflect extra contributing factors relating to status dynamics and hiring biases in entertainment media. In all cases, the human preference for familiar, prototypical representations of social categories is likely to play a role in perpetuating these biases[46,47]. We further anticipate that the study of online bias will benefit from extending our multimodal framework to analyse other modes of communication, such as audio and video, and to compare human and AI-generated content.

To keep pace with the evolving landscape of bias online, it is important for computational social scientists to expand beyond the analysis of textual data to include other content modalities that offer distinct ways of transmitting cultural information. Indeed, decades of research maintain that images lie at the foundation of human cognition[11,12,25,48] and may have provided the first means of human communication and culture[24,49]. It is therefore difficult to imagine how the science of human culture can be complete without a multimodal framework. Exploring the implications of an image-centric social reality for the evolution of human cognition and culture is a ripe direction for future research. Our study identifies one of many implications of this cultural shift concerning the amplification of social bias, stemming from the salient way in which images present demographic information when depicting social categories. Addressing the societal impact of this ascending visual culture will be essential in building a fair and inclusive future for the internet, and developing a multimodal approach to computational social science is a crucial step in this direction.

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

## Methods

Here we outline the computational and experimental techniques we use to compare gender bias in online images and texts. We begin by describing the methods of data collection and analyses developed for the observational component of our study. Then we detail the study design deployed in our online search experiment. The preregistration for our online experiment is available at https://osf.io/3jhzx. Note that this study is a successful replication of a previous study with a nearly identical design, except the original study did not include a control condition nor several versions of the text condition; the preregistration of the previous study is available at https://osf.io/26kbr.

### Observational methods

**Data collection procedure for online images.** Our crowdsourcing methodology consisted of four steps (Extended Data Fig. 1). First, we gathered all social categories in WordNet, a canonical lexical database of English. WordNet contained 3,495 social categories, including occupations (such as 'physicist') and generic social roles (such as 'colleague'). Second, we collected the images associated with each category from both Google and Wikipedia. Third, we used Python's OpenCV—a popular open-source deep learning framework—to extract the faces from each image; this algorithm automatically isolates each face and extracts a square including the entire face and minimal surrounding context. Using OpenCV to extract faces helped us to ensure that each face in each image was separately classified in a standardized manner, and to avoid subjective biases in coders' decisions for which face to focus on and categorize in each image. Fourth, we hired 6,392 human coders from MTurk to classify the gender of the faces. Following earlier work, each face was classified by three unique annotators[16,17], so that the gender of each face ('male' or 'female') could be identified based on the majority (modal) gender classification across three coders (we also gave coders the option of labelling the gender of faces as 'non-binary', but this option was only chosen in 2% of cases, so we excluded these data from our main analyses and recollected all classifications until each face was associated with three unique coders using either the 'male' or the 'female' label). Although coders were asked to label the gender of the face presented, our measure is agnostic to which features the coders used to determine their gender classifications; they may have used facial features, as well as features relating to the aesthetics of expressed gender such as hair or accessories. Each search was implemented from a fresh Google account with no prior history. Searches were run in August 2020 by ten distinct data servers in New York City. This study was approved by the Institutional Review Board at the University of California, Berkeley, and all participants provided informed consent.

To collect images from Google, we followed earlier work by retrieving the top 100 images that appeared when using each of the 3,495 categories to search for images using the public Google Images search engine[16–18] (Google provides roughly 100 images for its initial search results). To collect images from Wikipedia, we identified the images associated with each social category in the 2021 Wikipedia-based Image Text Dataset (WIT)[27]. WIT maps all images across Wikipedia to textual descriptions on the basis of the title, content and metadata of the active Wikipedia articles in which they appear. WIT contained images associated with 1,523 social categories from WordNet across all English Wikipedia articles (see Supplementary Information section A.1.1 for details on our Wikipedia analysis). The coders identified 18% of images as not containing a human face; these were removed from our analyses. We also asked all annotators to complete an attention check, which involved choosing the correct answer to the common-sense question "What is the opposite of the word 'down'?" from the following options: 'Fish', 'Up', 'Monk' and 'Apple'. We removed the data from all annotators who failed an attention check (15%), and we continued collecting classifications until each image was associated with the judgements of three unique coders, all of whom passed the attention check.

**Collecting human judgements of social categories.** We hired a separate sample of 2,500 human coders from MTurk to complete a survey study in which they were presented with social categories (five categories per task) and asked to evaluate each category by means of the following question (each category was assessed by 20 unique human coders): "Which gender do you most expect to belong to this category?" This was answered as a scalar with a slider ranging from −1 (females) to 1 (males). All MTurkers were prescreened such that only US-based MTurkers who were fluent in English were invited to participate in this task.

**Demographics of human coders.** The human coders were all adults based in the USA who were fluent in English. Supplementary Table 1 indicates that our main results are robust to controlling for the demographic composition of our human coders. Among our coders, 44.2% identified as female, 50.6% as male and 3.2% as non-binary; the remainder preferred not to disclose. In terms of age, 42.6% identified as being 18–24 years, 22.9% as 25–34, 32.5% as 35–54, 1.6% as 55–74 and less than 1% as more than 75. In terms of race, 46.8% identified as Caucasian, 11.6% as African American, 17% as Asian, 9% as Hispanic and 10.3% as Native American; the remainder identified as either mixed race or preferred not to disclose. In terms of political ideology, 37.2% identified as conservative, 33.8% as liberal, 20.3% as independent and 3.9% as other; the remainder preferred not to disclose. In terms of annual income, 14.3% reported making less than US$10,000, 33.4% reported US$10,000–50,000, 22.7% reported US$50,000–75,000, 14.9% reported US$75,000–100,000, 10.5% reported US$100,000–150,000, 2.8% reported US$150,000–250,000 and less than 1% reported more than US$250,000; the remainder preferred not to disclose. In terms of the highest level of education acquired by each annotator, 2.7% selected 'Below High School', 17.5% selected 'High School', 29.2% selected 'Technical/Community College', 34.5% selected 'Undergraduate degree', 14.8% selected 'Master's degree' and less than 1% selected 'Doctorate degree'; the remainder preferred not to disclose.

**Constructing a gender dimension in word embedding space.** Our method for measuring gender associations in text relies on the fact that word embedding models use the frequency of co-occurrence among words in text (for example, whether they occur in the same sentence) to position words in an $n$-dimensional space, such that words that co-occur together more frequently are represented as closer together in this $n$-dimensional space. The 'embedding' for a given word refers to the specific position of this word in the $n$-dimensional space constructed by the model. The cosine distance between word embeddings in this vector space provides a robust measure of semantic similarity that is widely used to unpack the cultural meanings associated with categories[13,22,31]. To construct a gender dimension in word embedding space, we adopt the methodology recently developed by Kozlowski et al.[22]. In their paper, Kozlowski et al.[22] construct a gender dimension in embedding space along which different categories can be positioned (for example, their analysis focuses on types of sport). They start by identifying two clustered regions in word embedding space corresponding to traditional representations of females and males, respectively. Specifically, the female cluster consists of the words 'woman', 'her', 'she', 'female' and 'girl', and the male cluster consists of the words 'man', 'his', 'he', 'male' and 'boy'. Then, for each of the 3,495 social categories in WordNet, we calculated the average cosine distance between this category and both the female and the male clusters. Each category, therefore, was associated with two numbers: its cosine distance with the female cluster (averaged across its cosine distance with each term in the female cluster), and its cosine distance with the male cluster (averaged across its cosine distance with each term in the male cluster). Taking the difference between a category's cosine distance with the female and male clusters allowed each category to be positioned along a −1 (female) to 1 (male) scale in embedding space. The category 'aunt', for instance,

falls close to −1 along this scale, whereas the category 'uncle' falls close to 1 along this scale. Of the categories in WordNet, 2,986 of them were associated with embeddings in the 300-dimensional word2vec model of Google News, and could therefore be positioned along this scale. All of our results are robust to using different terms to construct the poles of this gender dimension (Supplementary Fig. 18). However, our main analyses use the same gender clusters as ref. 22.

To compute distances between the vectors of social categories represented by bigrams (such as 'professional dancer'), we used the Phrases class in the Gensim Python package, which provided a prebuilt function for identifying and calculating distances for bigram embeddings. This method works by identifying an $n$-dimensional vector of middle positions between the vectors corresponding separately to each word in the bigram (for example, 'professional' and 'dancer'). This technique then treats this middle vector as the singular vector corresponding to the bigram 'professional dancer' and is thereby used to calculate distances from other category vectors. This same method was applied to the construction of embeddings for all bigram categories in all models.

To maximize the similarity between our text-based and image-based measures of gender association, we adopted the following three techniques. First, we normalized our textual measure of gender associations using minimum–maximum normalization, which ensured that a compatible range of values was covered by both our text-based and image-based measures of gender association. This is helpful because the distribution of gender associations for the image-based measure stretched to both ends of the −1 to 1 continuum as a result of certain categories being associated with 100% female faces or 100% male faces. By contrast, although the textual measure described above contains a −1 (female) to 1 (male) scale, the most female category in our WordNet sample has a gender association of −0.42 ('chairwoman'), and the most male category has a gender association of 0.33 ('guy'). Normalization ensures that the distribution of gender associations in the image- and text-based measures both equally cover the −1 to 1 continuum, so that paired comparisons between these scales (matched at the category level) can directly examine the relative ranking of a category's gender association in each measure. Minimum–maximum normalization is given by the following equation:

$$\widetilde{x_i} = \frac{(x_i - x_{\min})}{(x_{\max} - x_{\min})} \tag{1}$$

where $x_i$ represents the gender association of category $x_i$ ([−1,1]), $x_{\min}$ represents the category with the lowest gender score, $x_{\max}$ represents the category with the highest gender score, and $\widetilde{x_i}$ represents the normalized gender association of category $x_i$. To preserve the −1 to 1 scale in applying minimum–maximum normalization, we applied this procedure separately for male-skewed categories (that is, all categories with a gender association above 0), such that $x_{\min}$ represents the least male of the male categories and $x_{\max}$ represents the most male of the male categories. We applied this same procedure to the female-skewed categories, except that, because the female scale is −1 to 0, $x_{\min}$ represents the most female of the female categories and $x_{\max}$ represents the least female. For this reason, after the 0–1 female scale was constructed, we multiplied the female scores by −1 so that −1 represented the most female of the female categories and 0 represented the least. We then appended the female-normalized (−1 to 0) and male-normalized (0 to 1) scales. Both the male and female scales before normalization contained categories with values within four decimal points of zero ($|x| < 0.0001$), such that this normalization technique had no effect of arbitrarily pushing certain categories towards 0. Instead, the above technique has the advantage of stretching out the text-based measure of gender association to ensure that a substantial fraction of categories reach all the way to the −1 female region and all the way to the 1 male region of the continuum, similar to the distribution of values for the image-based measure.

## Experimental methods

**Participant pool.** For this experiment, a nationally representative sample of participants ($n = 600$) was recruited from the popular crowd-sourcing platform Prolific, which provides a vetted panel of high-quality human participants for online research. No statistical methods were used to determine this sample size. A total of 575 participants completed the task, exhibiting an attrition rate of 4.2%. We only examine data from participants who completed the experiment. Our main results report the outcomes associated with the Image, Text and Control conditions ($n = 423$); in the Supplementary Information, we report the results of an extra version of the Text condition involving the generic Google search bar ($n = 150$; Supplementary Fig. 26). We only examine data from participants who completed the task. To recruit a nationally representative sample, we used Prolific's prescreening functionality designed to provide a nationally representative sample of the USA along the dimensions of sex, age and ethnicity. Participants were invited to partake in the study only if they were based in the USA, fluent English speakers and aged more than 18 years. A total of 50.8% of participants were female (no participants identified as non-binary). All participants provided informed consent before participating. This experiment was run on 5 March 2022.

**Participant experience.** Extended Data Fig. 2 presents a schematic of the full experimental design. This experiment was approved by the Institutional Review Board at the University of California, Berkeley. In this experiment, participants were randomized to one of four conditions: (1) the Image condition (in which they used the Google Image search engine to retrieve images of occupations), (2) the Google News Text condition (in which they used the Google News search engine, that is, news.google.com, to retrieve textual descriptions of occupations), (3) the Google Neutral Text condition (in which they used the generic Google search engine, that is, google.com, to retrieve textual descriptions of occupations) and (4) the Control condition (in which they were asked at random to use either Google Images or the neutral (standard) Google search engine to retrieve descriptions of random, non-gendered categories, such as 'apple'). Note that, in the main text, we report the experimental results comparing the Image, Control and Google News Text conditions; we present the results concerning the Google Neutral Text condition as a robustness test in the Supplementary Information (Supplementary Fig. 26).

After uploading a description for a given occupation, participants used a −1 (female) to 1 (male) scale to indicate which gender they most associate with this occupation. In this way, the scale participants used to indicate their gender associations was identical to the scale we used to measure gender associations in our observational analyses of online images and text. In the control condition, participants were asked to indicate which gender they associate with a given randomly selected occupation after uploading a description for an unrelated category. Participants in all conditions completed this sequence for 22 unique occupations (randomly sampled from a broader set of 54 occupations). These occupations were selected to include occupations from science, technology, engineering and mathematics, and the liberal arts. Each occupation that was used as a stimulus could also be associated with our observational data concerning the gender associations measured in images from Google Images and the texts of Google News. Here is the full preregistered list of occupations used as stimuli: immunologist, mathematician, harpist, painter, piano player, aeronautical engineer, applied scientist, geneticist, astrophysicist, professional dancer, fashion model, graphic designer, hygienist, educator, intelligence analyst, logician, intelligence agent, financial analyst, chief executive officer, clarinetist, chiropractor, computer expert, intellectual, climatologist, systems analyst, programmer, poet, astronaut, professor, automotive engineer, cardiologist, neurobiologist, English professor, number theorist, marine engineer, bookkeeper, dietician,

model, trained nurse, cosmetic surgeon, fashion designer, nurse practitioner, art teacher, singer, interior decorator, media consultant, art student, dressmaker, English teacher, literary agent, social worker, screen actor, editor-in-chief, schoolteacher. The set of occupations that participants evaluated was identical across conditions.

Once each participant completed this task for 22 occupations, they were then asked to complete an IAT designed to measure the implicit bias towards associating men with science and women with liberal arts[33–35,38]. The IAT was identical across conditions ('Measuring implicit bias using the IAT'). In total, the experiment took participants approximately 35 minutes to complete. Participants were compensated at the rate of US $15 per hour for their participation.

**Measuring implicit bias using the IAT.** The IAT in our experiment was designed using the iatgen tool[33] (https://iatgen.wordpress.com/). The IAT is a psychological research tool for measuring mental associations between target pairs (for example, different races or genders) and a category dimension (for example, positive–negative, science–liberal arts). Rather than measuring what people explicitly believe through self-report, the IAT measures what people mentally associate and how quickly they make these associations. The IAT has the following design (description borrowed from iatgen)[33]: "The IAT consists of seven 'blocks' (sets of trials). In each trial, participants see a stimulus word on the screen. Stimuli represent 'targets' (for example, insects and flowers) or the category (for example, pleasant–unpleasant). When stimuli appear, the participant 'sorts' the stimulus as rapidly as possible by pressing with either their left or right hand on the keyboard (in iatgen, the 'E' and 'I' keys). The sides with which one should press are indicated in the upper left and right corners of the screen. The response speed is measured in milliseconds." For example, in some sections of our study, a participant might press with the left hand for all male + science stimuli and with their right hand for all female + liberal arts stimuli.

The theory behind the IAT is that the participant will be fast at sorting in a manner that is consistent with one's latent associations, which is expected to lead to greater cognitive fluency in one's intuitive reactions. For example, the expectation is that someone will be faster when sorting flowers + pleasant stimuli with one hand and insects + unpleasant with the other, as this is (most likely) consistent with people's implicit mental associations (example borrowed from iatgen). Yet, when the category pairings are flipped, people should have to engage in cognitive work to override their mental associations, and the task should be slower. The degree to which one is faster in one section or the other is a measure of one's implicit bias.

In our study, the target pairs we used were 'male' and 'female' (corresponding to gender), and the category dimension referred to science–liberal arts. To construct the IAT, we followed the design used by Rezaei[38]. For the male words in the pairs, we used the following terms: man, boy, father, male, grandpa, husband, son, uncle. For the female words in the pairs, we used the following terms: woman, girl, mother, female, grandma, wife, daughter, aunt. For the science category, we used the following words: biology, physics, chemistry, math, geology, astronomy, engineering, medicine, computing, artificial intelligence, statistics. For the liberal arts category, we used the following words: philosophy, humanities, arts, literature, English, music, history, poetry, fashion, film. Extended Data Figs. 3–6 illustrate the four main IAT blocks that participants completed (as per standard IAT design, participants were also shown blocks 2, 3 and 4, with the left–right arrangement of targets reversed). Participants completed seven blocks in total, sequentially. The IAT instructions for Extended Data Fig. 3 state, "Place your left and right index fingers on the E and I keys. At the top of the screen are 2 categories. In the task, words and/or images appear in the middle of the screen. When the word/image belongs to the category on the left, press the E key as fast as you can. When it belongs to the category on the right, press the I key as fast as

you can. If you make an error, a red X will appear. Correct errors by hitting the other key. Please try to go as fast as you can while making as few errors as possible. When you are ready, please press the [Space] bar to begin." These instructions are repeated throughout all blocks in the task.

To measure implicit bias based on participants' reaction times during the IAT, we adopted the following standard approach (used by iatgen). We combined the scores across all four blocks (blocks 3, 4, 6 and 7 in iatgen). Some participants are also faster than others, adding statistical 'noise' as a result of variance in overall reaction times. Thus, instead of comparing within-person differences in raw latencies, this difference is standardized at the participant level, dividing the within-person difference by a 'pooled' standard deviation. This pooled standard deviation uses the standard deviation of what are called the practice and critical blocks combined. This yields a $D$ score. In iatgen, a positive $D$ value indicates association in the form of target A + positive, target B + negative, which in our case is male + science, female + liberal arts), whereas a negative $D$ value indicates the opposite bias (target A + negative, target B + positive, which in our case is male + liberal arts, female + science), and a zero score indicates no bias.

Our main experimental results evaluate the relationship between the participants' explicit and implicit gender associations and the strength of gender associations in the Google images and textual descriptions they encountered during the search task. The strength of participants' explicit gender associations is calculated as the absolute value of the number they input using the −1 (female) to 1 (male) scale after each occupation they classified (Extended Data Fig. 2). Participants' implicit bias is measured by the $D$ score of their results on the IAT designed to detect associations between men and science and women and liberal arts. To measure the strength of gender associations in the Google images that participants encountered, we calculated the gender parity of the faces uploaded across all participants who classified a given occupation. For example, we identified the responses of all participants who provided image search results for the occupation 'geneticist', and we constructed the same gender dimensions as described in the main text, such that −1 represents 100% female faces, 0 represents 50% female (male) faces and 1 represents 100% male faces. To identify the gender of the faces of the images that participants uploaded, we recruited a separate panel of MTurk workers ($n = 500$) who classified each face (there were 3,300 images in total). Each face was classified by two unique MTurkers; if they disagreed in their gender assignment, a third MTurk worker was hired to provide a response, and the gender identified by the majority was selected. We adopted an analogous approach to annotating the gender of the textual descriptions that participants uploaded in the text condition. These annotators identified whether each textual or visual description uploaded by participants was female (1), neutral (0) or male (1). Each textual description was coded as male, female or neutral on the basis of whether it used male or female pronouns or names to describe the occupation (for example, referred to a 'doctor' as 'he'); textual descriptions were identified as neutral if they did not ascribe a particular gender to the occupation described. We were then able to calculate the same measure of gender balance in the textual descriptions uploaded for each occupation as we applied in our image analysis.

### Reporting summary
Further information on research design is available in the Nature Portfolio Reporting Summary linked to this article.

### Data availability
All data collected for this study are publicly available at https://github.com/drguilbe/ImgVSText. Preregistration for experiment is available at https://osf.io/3jhzx. Source data are provided with this paper.

## Code availability

All code underlying this study is publicly available at https://github.com/drguilbe/ImgVSText.

**Acknowledgements** We acknowledge members of the MRL (Macro Research Lunch) seminar at the Haas School of Business and the Computational Culture lab, as well as Nina Guilbeault, Nicholas Guilbeault, P. Reginato and R. Lo Sardo for helpful feedback on this project. We thank S. Nanniyur for assistance with data collection. This project was funded by grants from the Fisher Center for Business Analytics; the Center for Equity, Gender and Leadership; and the Barbara and Gerson Bakar Fellowship, each awarded to D.G., and by a grant from the Cora Jane Flood Endowment Fund awarded to S.D., all through the University of California, Berkeley.

**Author contributions** D.G. designed the project. D.G., E.N. and B.S.D. analysed the data. D.G. and S.D. wrote the manuscript. D.G., S.D., T.H., B.S.D., E.N. and M.C. collected the data. D.G., T.H., E.N. and B.S.D. developed the algorithmic methods.

**Competing interests** The authors declare no competing interests.

**Additional information**
**Correspondence and requests for materials** should be addressed to Douglas Guilbeault.

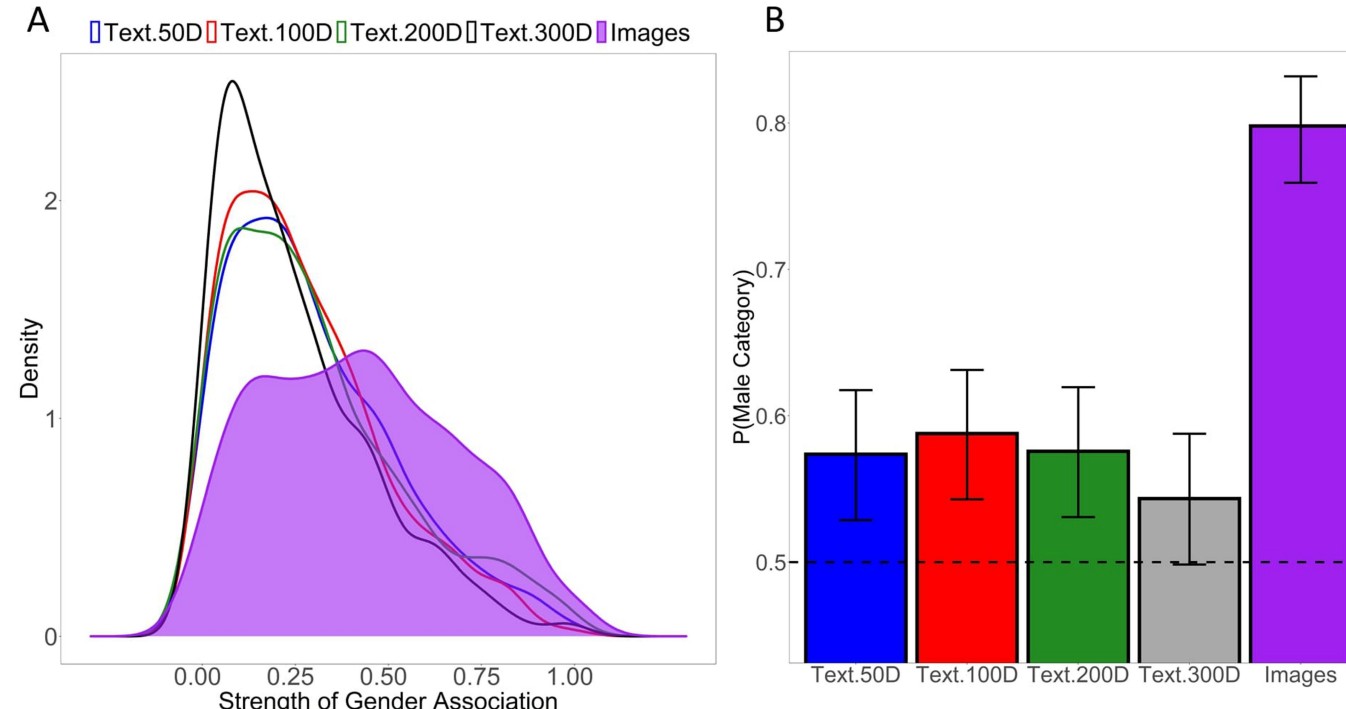

**Extended Data Fig. 1 | Replication using text and image data from Wikipedia.** (A) The absolute strength of gender associations with each category according to textual and visual data from Wikipedia, shown for pre-trained word embedding models of the same Wikipedia dataset trained with different dimensionality[26]. We restrict our analysis to only those categories that (i) were associated with at least 10 faces in the English Wikipedia Image Text (WIT) dataset[27], and (ii) were also present in pre-trained word embedding models of Wikipedia text data[26]. This yielded 495 categories. Results equally replicate if we examine all 1,244 categories that could be matched across these image and text sources (see section A.1.1 in the supplementary appendix). The image-based measure captures the frequency of male and female faces associated with Wikipedia articles on each category (−1 means 100% female; 1 means 100% male); the text-based measure captures the frequency at which each category is associated with male or female terms in Wikipedia articles (−1 means 100% female; 1 means 100% male associations). Panel A shows that the absolute strength of gender associations is significantly higher in images ($\mu = 0.33$), as compared to word embedding models of Wikipedia, regardless of their dimensionality ($p < 0.0001$, Wilcoxon signed-rank test, two-tailed; all word embedding models exhibit an average strength of gender association below 0.1). Panel B shows that 80% of categories are male-skewed according to Wikipedia images ($p < 0.0001$, proportion test, $n = 495$, two-tailed), whereas word embedding models of Wikipedia with different dimensionality show, respectively, 57% (50D), 59% (100D), 57.6% (200D), and 54% (300D) of categories as male, all of which present a significantly weaker male-skew than Wikipedia images at the $p < 0.0001$ level (proportion test, two-tailed). Error bars show 95% confidence intervals calculating using a single-sample proportion test.

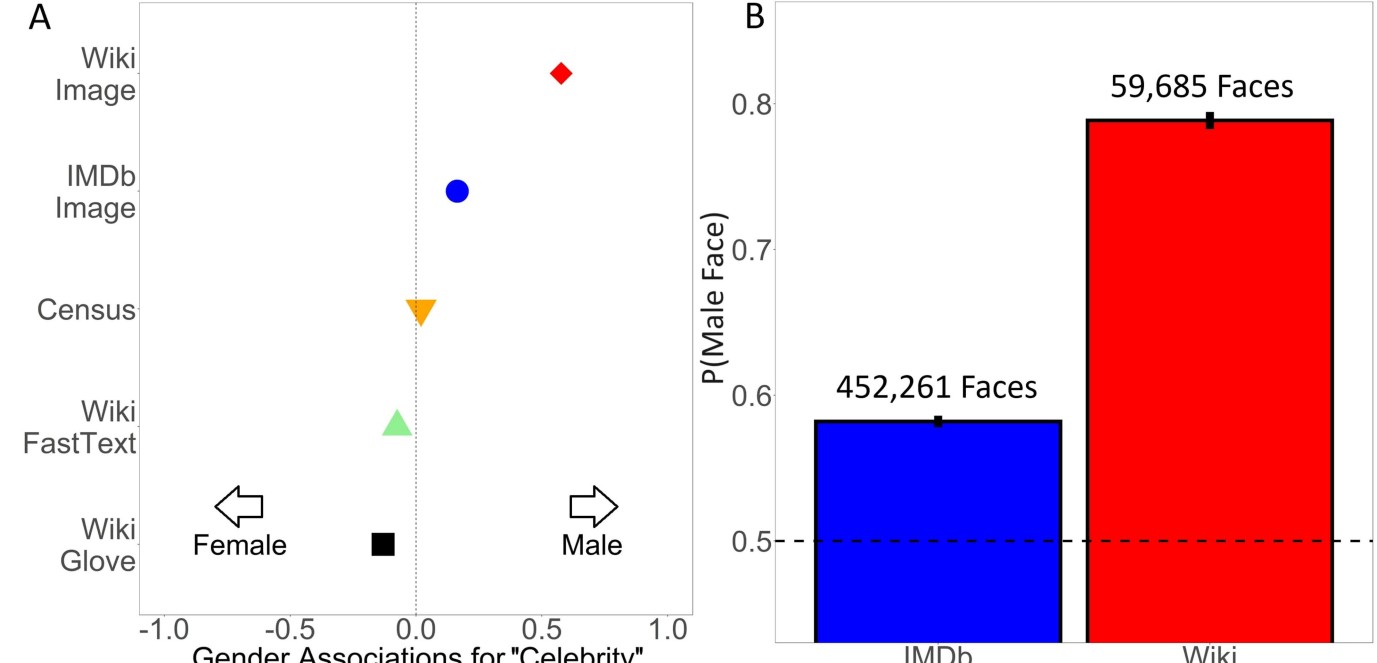

**Extended Data Fig. 2 | Replication using Wikipedia and IMDb images depicting celebrities, where each image is associated with the self-identified gender of the person depicted.** Comparing the gender associations with the social category "celebrity" across four datasets: the Rothe et al.[28] IMDb-Wikipedia Face Image dataset containing[28] (for which the self-reported gender of each face is known), the 2019 census, the FastText word embedding model of gender associations in Wikipedia, and the GloVe word embedding model of gender associations in Wikipedia (see section A.1.2 of the supplementary appendix for details on this analysis). The Rothe et al.[28] dataset contains 511,946 images, where 452,261 IMDb images depict 19,091 celebrities, and 59,685 Wikipedia images depict 58,904 celebrities. Panel A displays the gender associations of "celebrity" across these datasets (−1 means 100% female associations; 1 means 100% male associations; 0 means equally male and female associations). The gender association of "celebrity" is −0.05 according to the FastText model and −0.08 according to the GloVe model (both weakly female-skewed). Meanwhile, the census indicates that 49% of celebrities are women, resulting in a gender association of 0.02 that fails to be significantly skewed toward a particular gender ($p = 0.54$, Student's $t$-test, two-sample, two-tailed). By contrast, the gender association of "celebrity" is 0.57 (0.16) according to Wikipedia (IMDb) images, marking a strong male-skew. Panel B shows the fraction of male faces identified in the IMDb-Wikipedia Face Image dataset, shown separately for Wikipedia and IMDb. 79% (58%) of celebrities depicted over Wikipedia (IMDb) are male, exhibiting a strong male bias in both sources ($p = 2.2 \times 10^{-16}$, Proportion test, two-sample, two-tailed, for both sources). Bars show the proportion of male faces, and error bars show 95% confidence intervals calculated with a single-sample proportion test (two-tailed).

**(A)** Gather all social categories from Wordnet

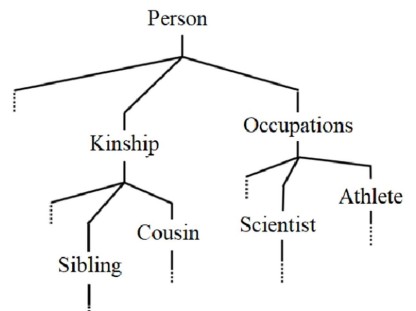

**(B)** Collect top 100 Google images for each social category

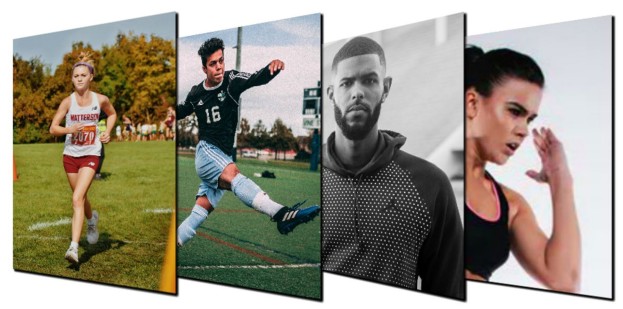

**(C)** Automatically extract all faces in each image

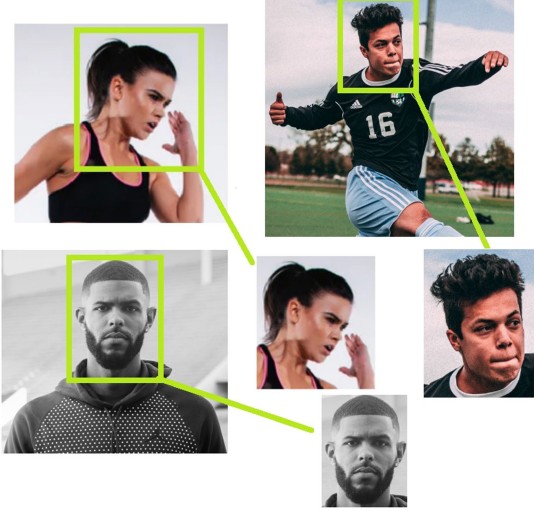

**(D)** Use crowdsourcing to classify each face

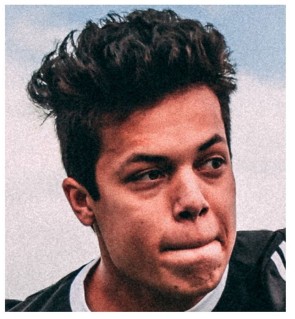

What is the gender of this face?
Male ○  Female ○  Non-binary ○

**Extended Data Fig. 3 | A schematic diagram of our main data collection methodology.** (A) First, we identify all social categories in the lexical ontology Wordnet; (B) second, we use each social category from Wordnet as a search term and automatically collect the top 100 images associated with each search term in Google Images; then (C) we use the OpenCV machine learning application to automatically crop the faces from each of the images collected from Google Images; and finally (D) we automatically upload each extracted face to Amazon's Mechanical Turk where it is classified by three unique human coders. The gender classification of each face is identified as the modal (majority) classification across these three coders. If a coder labeled the gender of a face as non-binary, we remove this classification and hire an additional coder to provide a classification to disambiguate. This allowed all faces in our dataset to be assigned a binary gender classification as "female" or "male".

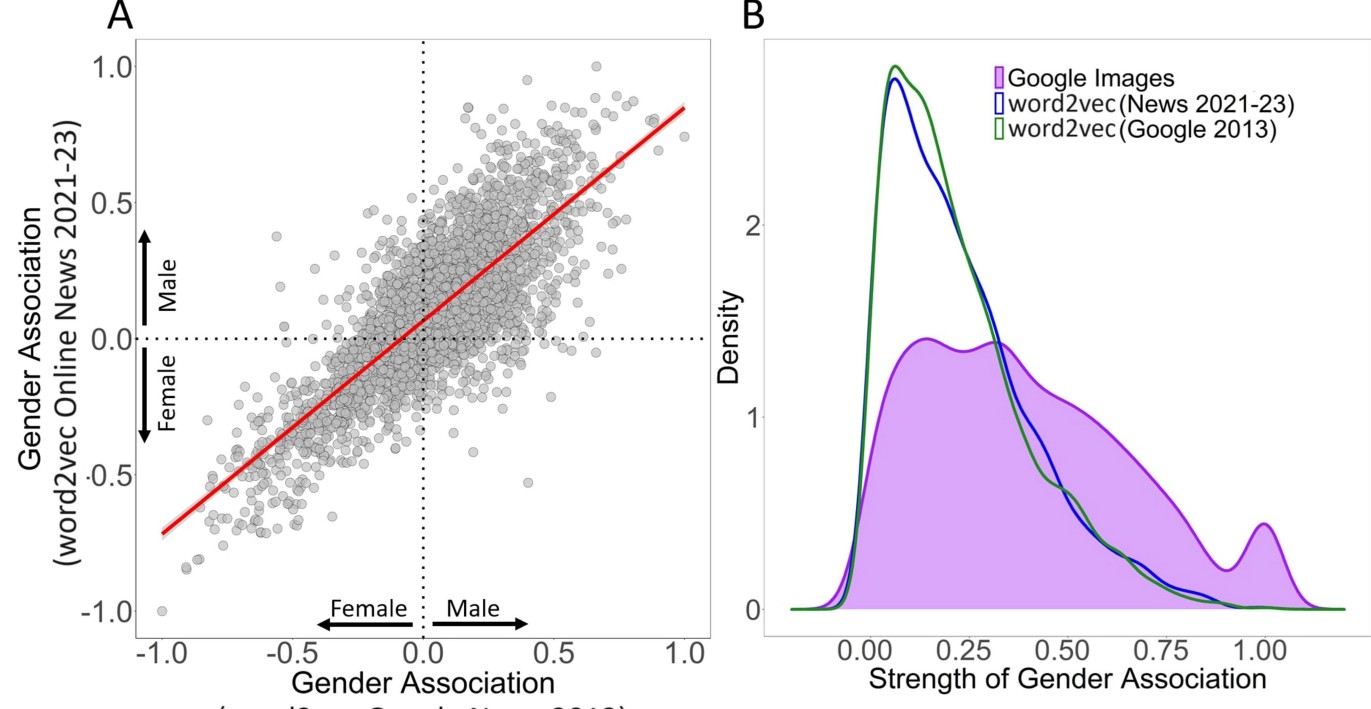

**Extended Data Fig. 4 | Replication comparing gender associations in Google Images to those in a word2vec model trained on a recent sample of online news, published between 2021 and 2023.** This recent sample of online news consists of 2,717,000 randomly sampled news articles published in English across various topics between January 2021 and August 2023. These articles were scraped from the following sources: 1,000,000 articles from the BBC; 500,000 from the Huffington Post; 480,000 from CNBC; 400,000 from Bloomberg; 160,000 articles Time Magazine; 150,000 from Techcrunch; and 27,000 from CNN. (A) Displaying the strong, positive correlation between the gender associations of all social categories in WordNet according to the 2013 Google News word2vec model and our word2vec model retrained using this recent online news data ($r = 0.79$, $p = 2.2 \times 10^{-16}$, Pearson correlation, two-tailed). The trend line reflects the linear correlation between these

models' gender associations, with error bands displaying 95% confidence intervals. Each data point corresponds to a distinct category. 2,992 social categories could be matched across these models. (B) Displaying the absolute strength of gender associations across these same social categories according to each word2vec model as compared to our sample of Google Images collected in 2021. There is no significant difference in the strength of gender associations between the 2013 word2vec model ($\mu = 0.22$) and our 2023 retrained word2vec model ($\mu = 0.22$) ($p = 0.14$, Student's $t$-test, two-tailed). However, the strength of gender associations in the Google Image data ($\mu = 0.39$) is significantly higher than that of both word2vec models at the $p < 0.0001$ level (Student's $t$-test, two-tailed). See section A.1.8 of the supplementary appendix for further details on this analysis.

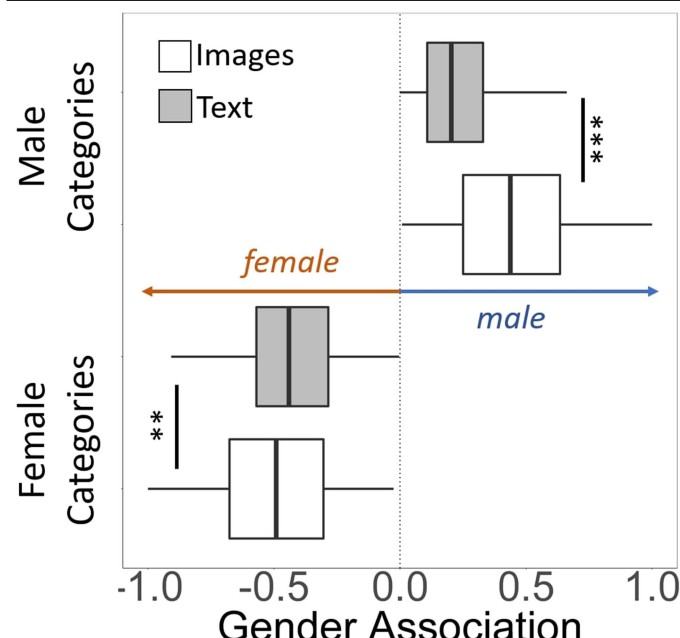

**Extended Data Fig. 5 | Robustness of results to examining only categories that have consistent gender associations across datasets.** This figure shows the strength of gender association across modalities for categories consistently identified as either female- or male-skewed by images from Google images, texts from Google News, and crowdsourced human judgments (*n* = 1,472 categories). See "Collecting Human Judgments of Social Categories" in the Methods section for details; this analysis leverages the average gender rating for each category averaged across 20 unique human coders (2,500 coders in total). 1,281 categories were consistently identified as male-skewed and 191 categories were consistently identified as female-skewed across these sources. The female (male) categories shown along the vertical axis are those that were associated with women (men) in images, text, and human judgments. The horizontal axis displays the gender associations for the same female (male) skewed categories according to images from Google Images and texts from Google News. Box plots show the interquartile range (IQR) +/− 1.5 X IQR. The strength of gender bias is significantly higher in images than text for female-skewed (*p* = 0.005, t = 2.84, MD = 0.05) and male-skewed (*p* = $2.2 \times 10^{-16}$, t = 27.93, MD = 0.22) categories (Student's *t*-test, two-sample, two-tailed, paired at the category level). **, *p* < 0.01; ***, *p* < 0.001.

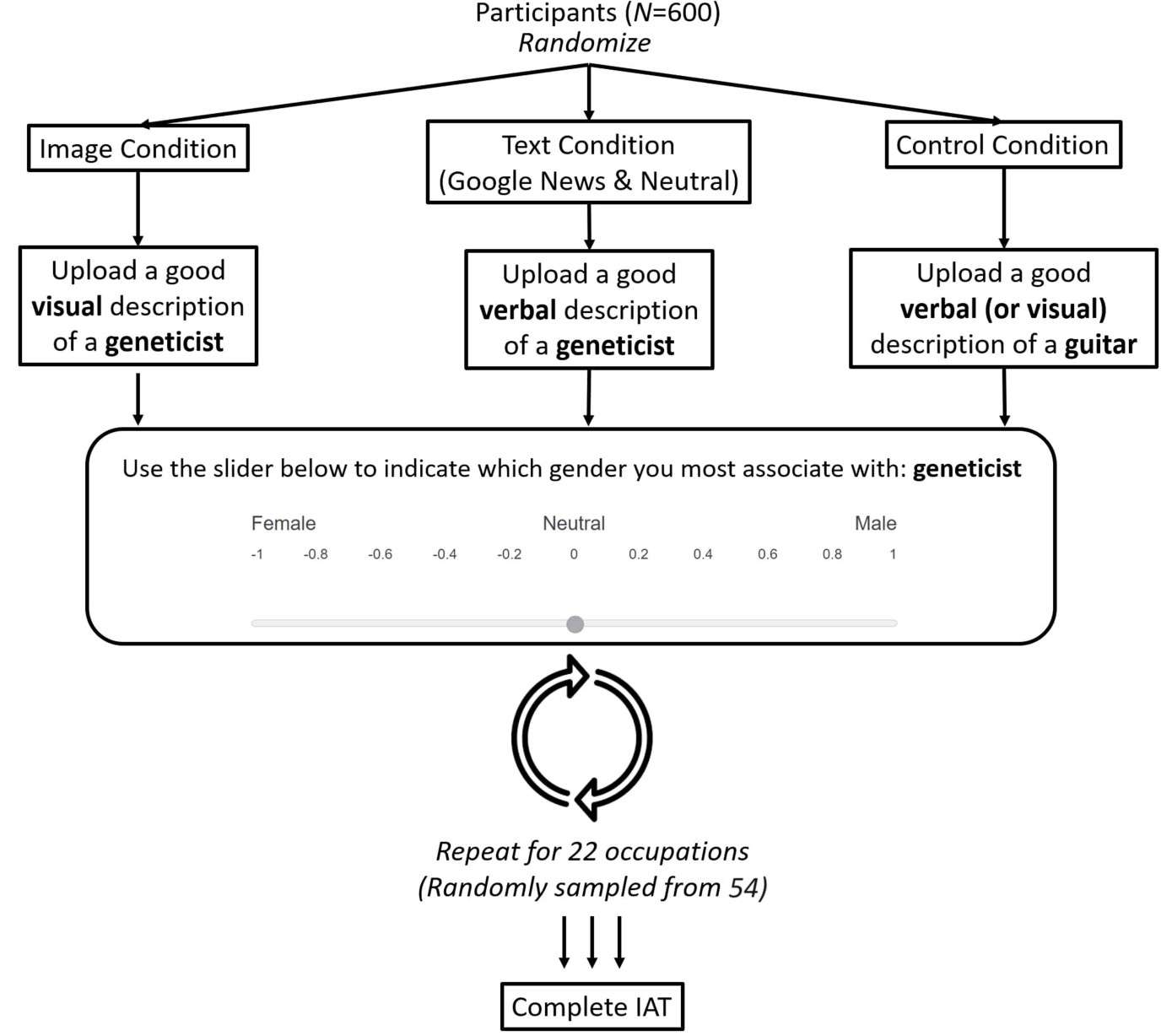

**Participants (*N*=600)**
*Randomize*

**Image Condition**

**Text Condition**
(Google News & Neutral)

**Control Condition**

Upload a good
**visual** description
of a **geneticist**

Upload a good
**verbal** description
of a **geneticist**

Upload a good
**verbal (or visual)**
description of a **guitar**

Use the slider below to indicate which gender you most associate with: **geneticist**

| Female | | | | Neutral | | | | Male |
| -1 | -0.8 | -0.6 | -0.4 | -0.2 | 0 | 0.2 | 0.4 | 0.6 | 0.8 | 1 |

*Repeat for 22 occupations*
*(Randomly sampled from 54)*

Complete IAT

**Extended Data Fig. 6 | Schematic representation of the experimental design.** A nationally representative US sample of participants (*n* = 600) were randomized into one of four conditions: (i) the Google image condition, (ii) the Google News text condition, (iii) the Google Neutral text condition, and (iv) the control condition (in which they were asked at random to use either Google Images or the main Google search engine to retrieve either visual or textual descriptions of random, non-gendered categories, such as *guitar* or *apple*). After uploading a description for a given occupation, participants used a −1 (female) to 1 (male) scale to indicate which gender they most associate with this occupation. In the control condition, participants were asked to indicate which gender they associate with a randomly selected occupation after uploading a description for an unrelated category. Participants in all conditions completed this sequence for 22 unique occupations (randomly sampled from a broader set of 54 occupations). Once each participant completed this task for 22 occupations, they were asked to complete an Implicit Association test (IAT) designed to measure the implicit bias toward associating men with science and women with liberal arts.

A

| Male | Female |
|---|---|

+

Instructions: Place your left and right index fingers on the E and I keys. At the top of the screen are 2 categories. In the task, words and/or images appear in the middle of the screen.

When the word/image belongs to the category on the left, press the **E** key as fast as you can. When it belongs to the category on the right, press the **I** key as fast as you can. If you make an error, a red **X** will appear. Correct errors by hitting the other key.

Please try to go as *fast as you can* while making as few errors as possible.

When you are ready, please press the [Space] bar to begin.

| Male | Female |
|---|---|

Girl

B

| Liberal Arts | Science |
|---|---|

+

Now, the categories have changed, but the rules remain the same. Please try to go as *fast as you can* while making as few errors as possible. Correct errors by hitting the other key.

When you are ready, please press the [Space] bar to begin.

| Liberal Arts | Science |
|---|---|

Engineering

**Extended Data Fig. 7 | Implicit Association Test (IAT) block 1 (A) and 2 (B).** Left panels indicate instructions. Right panels present an example of a word that participants need to assigned to the correct side of the screen. There were seven blocks in total. The instructions displayed are also written in text in the "Measuring Implicit Bias using the IAT" of the Methods.

**A**

| Male | Female |
|---|---|
| or | or |
| **Liberal Arts** | **Science** |

+

Now the four categories you saw separately will appear together. Remember, each word/image fits in only one of the four categories. The label/item colors may help you identify the appropriate category.

Use the **E** key for the two categories on the left and the **I** key for the two categories on the right. Again, try to go as fast as possible without making mistakes. Correct errors by hitting the other key. Practice this combination now.

When you are ready, please press the [Space] bar to begin.

| Male | Female |
|---|---|
| or | or |
| **Liberal Arts** | **Science** |

**History**

**B**

| Female | Male |
|---|---|
| or | or |
| **Liberal Arts** | **Science** |

+

Notice the four categories have been combined again, but in a new configuration. Please practice this combination now, and remember to go as fast as you can while making as few mistakes as possible. Correct errors by hitting the other key.

When you are ready, please press the [Space] bar to begin.

| Female | Male |
|---|---|
| or | or |
| **Liberal Arts** | **Science** |

**History**

**Extended Data Fig. 8 | Implicit Association Test (IAT) block 3 (A) and 4 (B).** Left panels indicate instructions. Right panels present an example of a word that participants need to assigned to the correct side of the screen. There were seven blocks in total. The instructions displayed are also written in text in the "Measuring Implicit Bias using the IAT" of the Methods.

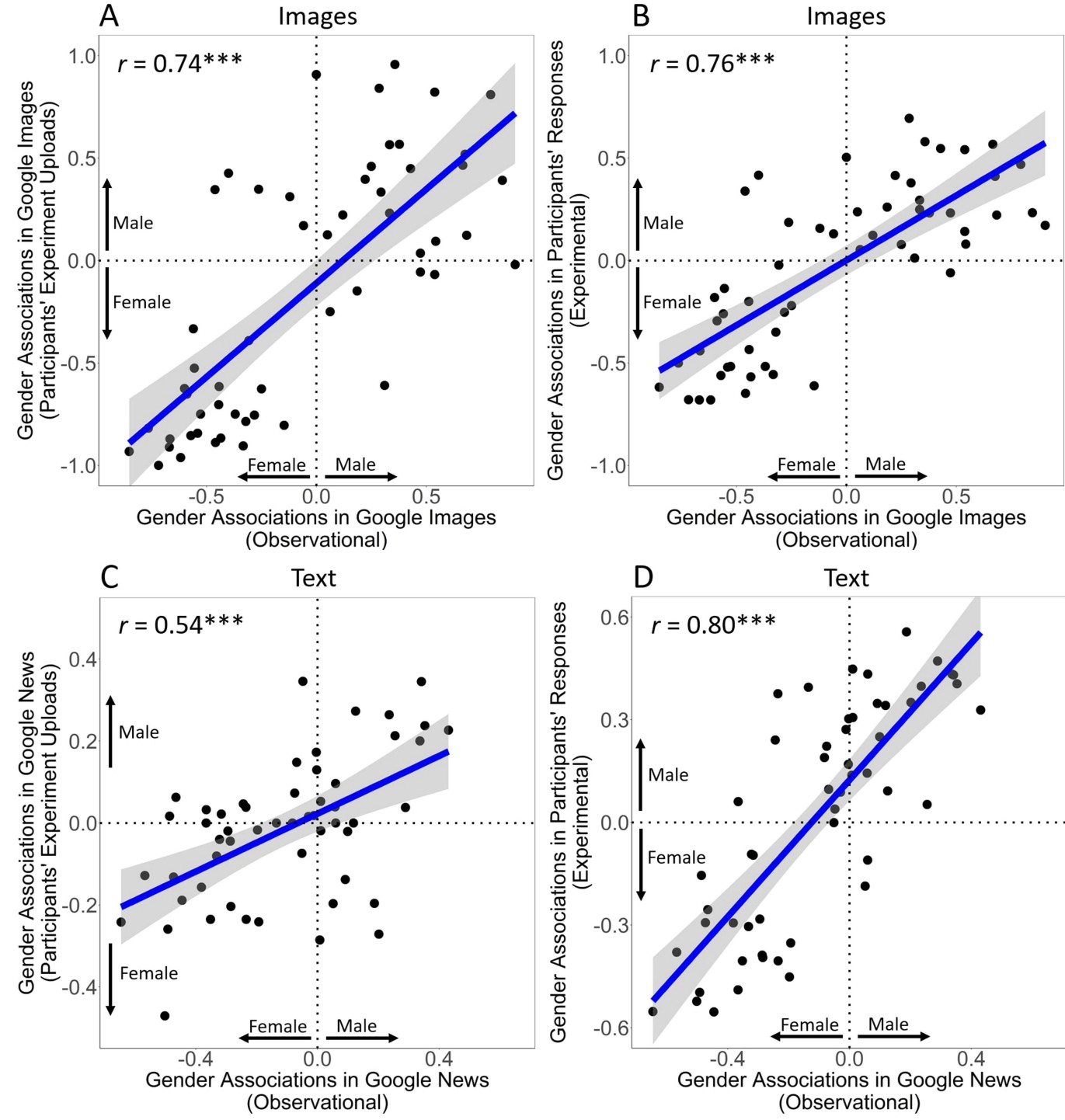

**Extended Data Fig. 9 | Demonstrating that gender associations in our observational data strongly predict gender associations in our experimental data.** The relationship between the gender associations in our observational sample of the top 100 Google images per occupation and (A) the gender associations in the Google images uploaded by participants in our search experiment (***, $p = 1.8 \times 10^{-10}$; $r = 0.73$, Pearson correlation, two-tailed), as well as (B) participants' self-reported gender associations for each occupation, which they provided after uploading an image from Google (***, $p = 1.5 \times 10^{-11}$; $r = 0.76$, Pearson correlation, two-tailed). The relationship between the gender associations in our observational word embedding measures of Google News and (C) the gender associations in the textual descriptions from Google News uploaded by participants (***, $p = 2.5 \times 10^{-5}$; $r = 0.54$, Pearson correlation, two-tailed), as well as (D) participants' self-reported gender associations for each occupation, which they provided after uploading a textual description from Google News (***, $p = 7.6 \times 10^{-13}$; $r = 0.8$, Pearson correlation, two-tailed). All results in all panels are averaged at the occupation level, such that 54 data points (occupations) are shown. In all panels: data points show mean values; lines show a standard OLS model fit to the scattered points using only the variables along the vertical and horizontal axis; error bands show 95% confidence intervals. ***, $p < 0.001$ (Pearson correlation, two-tailed, for all panels). See section A.2.4 of the supplementary appendix for further details on this analysis.

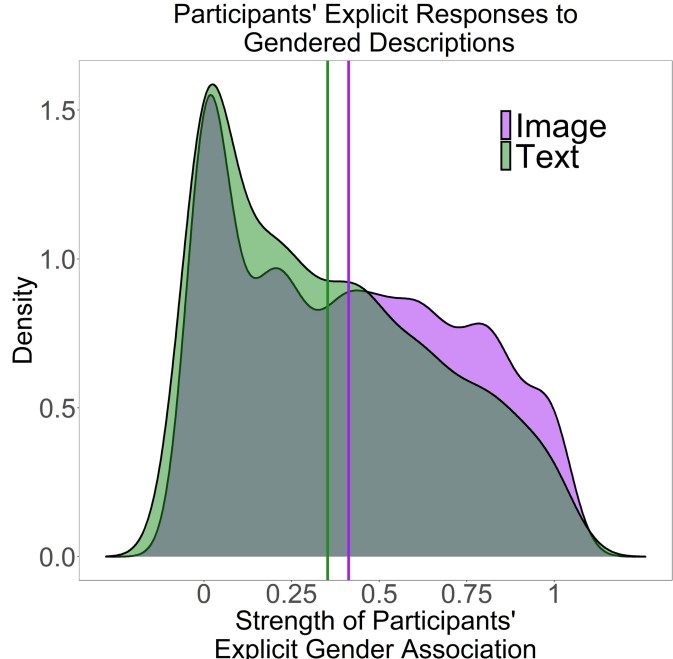

**Extended Data Fig. 10 | Gendered images prime gender bias more strongly than gendered texts, holding constant the occupation being described by these images and texts.** Figure displays the average absolute strength of the gender associations that participants reported for each occupation in each condition, while restricting this analysis to only those descriptions that were explicitly gendered as either male or female in the text and image condition. The green (purple) vertical lines indicate average effects for the text (image) condition. n = 2,775 image descriptions; n = 706 text descriptions. Participants in the image condition exhibited significantly stronger biases in the gender associations they reported for occupations ($p = 5.09 \times 10^{-6}$, $t = 4.58$, MD = 0.06, Student's $t$-test, two-sample, two-tailed), even when participants' in the image and text condition both uploaded a gendered description of the same occupation; this result holds when using a linear regression to control for the specific gender and the specific occupation associated with the uploaded description ($\beta = 0.05$, SE = 0.01, $p = 2.08 \times 10^{-5}$). These findings indicate that even when gender is salient in both text and image, exposure to images leads to stronger biases in people's beliefs. See section A.2.6 of the supplementary appendix for further details on this analysis.

# Reporting Summary

## Statistics

For all statistical analyses, confirm that the following items are present in the figure legend, table legend, main text, or Methods section.

| n/a | Confirmed | |
|---|---|---|
| ☐ | ☒ | The exact sample size (*n*) for each experimental group/condition, given as a discrete number and unit of measurement |
| ☐ | ☒ | A statement on whether measurements were taken from distinct samples or whether the same sample was measured repeatedly |
| ☐ | ☒ | The statistical test(s) used AND whether they are one- or two-sided<br>*Only common tests should be described solely by name; describe more complex techniques in the Methods section.* |
| ☐ | ☒ | A description of all covariates tested |
| ☐ | ☒ | A description of any assumptions or corrections, such as tests of normality and adjustment for multiple comparisons |
| ☐ | ☒ | A full description of the statistical parameters including central tendency (e.g. means) or other basic estimates (e.g. regression coefficient) AND variation (e.g. standard deviation) or associated estimates of uncertainty (e.g. confidence intervals) |
| ☐ | ☒ | For null hypothesis testing, the test statistic (e.g. *F*, *t*, *r*) with confidence intervals, effect sizes, degrees of freedom and *P* value noted<br>*Give P values as exact values whenever suitable.* |
| ☒ | ☐ | For Bayesian analysis, information on the choice of priors and Markov chain Monte Carlo settings |
| ☒ | ☐ | For hierarchical and complex designs, identification of the appropriate level for tests and full reporting of outcomes |
| ☐ | ☒ | Estimates of effect sizes (e.g. Cohen's *d*, Pearson's *r*), indicating how they were calculated |

*Our web collection on statistics for biologists contains articles on many of the points above.*

## Software and code

Policy information about availability of computer code

| | |
|---|---|
| Data collection | All observational data was collected using custom code written in Python by our team (Python version 3.12.0). Experimental data was collected using Qualtrics. The IAT component of the experiment was collected using the iatgen software (version release 1.0), see link here: https://iatgen.wordpress.com/ |
| Data analysis | All data analyses were completed using custom code written in both R and Python by our team. Python version 3.12.0; R version 4.3.2. All data and code relating to data analyses is publicly available at the following github: https://github.com/drguilbe/ImgVSText |

For manuscripts utilizing custom algorithms or software that are central to the research but not yet described in published literature, software must be made available to editors and reviewers. We strongly encourage code deposition in a community repository (e.g. GitHub). See the Nature Portfolio guidelines for submitting code & software for further information.

## Data

Policy information about availability of data

All manuscripts must include a data availability statement. This statement should provide the following information, where applicable:
- Accession codes, unique identifiers, or web links for publicly available datasets
- A description of any restrictions on data availability
- For clinical datasets or third party data, please ensure that the statement adheres to our policy

All data collected in association with this project is publicly available at the following github: https://github.com/drguilbe/ImgVSText

# Field-specific reporting

Please select the one below that is the best fit for your research. If you are not sure, read the appropriate sections before making your selection.

☐ Life sciences    ☒ Behavioural & social sciences    ☐ Ecological, evolutionary & environmental sciences

# Behavioural & social sciences study design

All studies must disclose on these points even when the disclosure is negative.

| | |
|---|---|
| Study description | This study has two components. The first component is observational and provides a quantitative algorithmic audit of gender stereotypes in online texts and images (from Google, Wikipedia, and IMDb). The second component is experimental and compares the effects of googling for images or textual descriptions of occupations on participants' explicit and implicit gender bias (specifically on their implicit bias toward associating men with science and women with liberal arts). |
| Research sample | The research sample for the experimental component of this study consists of a nationally representative sample of the U.S., as curated by the crowdsourcing platform Prolific. The details on Prolific's U.S. nationally representative sample are provided by Prolific at the following link: https://researcher-help.prolific.co/hc/en-gb/articles/360019236753-Representative-samples.<br><br>To create a representative U.S. sample, Prolific takes the intended sample size and stratifies it across three demographics: age, sex and ethnicity. Prolific uses census data from the US Census Bureau to divide the sample into subgroups with the same proportions as the national U.S. population. This means, for example, that a representative sample contains the same proportion of 28-37 year old Asian women as the national population (to the extent possible). Using this representative sample is important for our experiment which does not make any demographic-specific predictions around the effects of internet search modality on gender bias; instead, we aim to identify an effect of internet search modality across demographic groups, and for this reason, using a representative sample enhances our ability to claim that our effect holds in a population whose demographic composition captures the diversity that characterizes the entire U.S. population, and is not, therefore, an artifact of an idiosyncratic demographic distribution in our sample. |
| Sampling strategy | For the experimental component of our study, our sampling strategy was a random sample from Prolific's nationally representative panel (N=600, see "Research sample"). No statistical methods were used to determine sample size prior to data collection. We could not find any prior work that could provide a reasonable estimate of the expected effect size needed to develop robust power calculations. Given that the participants in our sample were statistically independent and our between-condition comparisons used basic WIlcoxon/Student T tests, 150 subjects per condition was deemed to be more than sufficient, since these tests can readily return robust, statistically significant differences between conditions for sample sizes of less than 30 per condition. Moreover, given that participants in our experiment rated 22 occupations, our experiment provided 13,200 raw estimates, such that we expected regression analyses using the raw data -- as well as within-subject analyses -- would be more than sufficiently powered. The experiment presented in this published paper is a replication of a prior experiment which used the same sample size and provided clear results (pre-registration here:  https://osf.io/26kbr), so we maintained this sample size in our successful replication (which differed only by the inclusion of a control condition).<br><br>For the observational component of this study, our sample of Google Images was collected through the following standardized procedure. We started by using each of the social categories in Wordnet to automatically search and retrieve the top 100 images in Google corresponding to each social category in Google Images (Google provides roughly 100 images by default for its initial results on a given search query). Each search was implemented from a fresh Google account with no prior history to avoid the uncontrolled effects of Google's recommendation algorithm, which customizes search results based on browsing history. Searches were run by 10 distinct data servers in New York City. All image data from Google was collected in August 2020. For the sample of Google News articles, we used the pre-trained word2vec embedding models of over 100 billion word tokens from Google News as provided by the gensim Python package: https://code.google.com/archive/p/word2vec/. In total, we compare our image data (including data from Wikipedia) against seven popular word embedding models (see Table S1 in our appendix). For each category, we extracted the publicly available images that Wikipedia provides on the Wikipedia article corresponding to this category. All images from Wikipedia were extracted using WIT ("Wikipedia Image Text Dataset"), the largest multimodal dataset on record (to date). WIT was released in 2021 and can be accessed here: https://github.com/google-research-datasets/wit |
| Data collection | The online textual data is derived from publicly available repositories of online articles from Google News and Wikipedia. The online image data is similarly derived from publicly available image repositories stored via Google Image Search and Wikipedia (see "Sampling strategy"). The experimental data collection was implemented using a survey instrument designed in Qualtrics; for more information. |
| Timing | All of our main image data from Google was collected in August 2020 (see "Sampling strategy"). All textual data from Google news is based on publicly available embedding models via the gensim Python package, which were first released in 2013 and were updated in 2021. All images from Wikipedia were extracted using WIT ("Wikipedia Image Text Dataset"), which was released in 2021. |
| Data exclusions | No data were excluded from observational analyses. For our experimental design, we only examined data associated with recruited participants who successfully joined and completed the task. 575 of the 600 participants recruited to our study completed the task, exhibiting an attrition rate of 4.2%. |
| Non-participation | No participants dropped out of the experiment. |

| Randomization | In the experimental component of our study, participants were randomized to one of four conditions: (1) the Image condition, (2) the Google News Text condition, (3) the Generic Google Search Bar Text condition, and (4) the Control condition. |
|---|---|

# Reporting for specific materials, systems and methods

We require information from authors about some types of materials, experimental systems and methods used in many studies. Here, indicate whether each material, system or method listed is relevant to your study. If you are not sure if a list item applies to your research, read the appropriate section before selecting a response.

## Materials & experimental systems

| n/a | Involved in the study |
|---|---|
| ☒ ☐ | Antibodies |
| ☒ ☐ | Eukaryotic cell lines |
| ☒ ☐ | Palaeontology and archaeology |
| ☒ ☐ | Animals and other organisms |
| ☐ ☒ | Human research participants |
| ☒ ☐ | Clinical data |
| ☒ ☐ | Dual use research of concern |

## Methods

| n/a | Involved in the study |
|---|---|
| ☒ ☐ | ChIP-seq |
| ☒ ☐ | Flow cytometry |
| ☒ ☐ | MRI-based neuroimaging |

## Human research participants

Policy information about studies involving human research participants

| Population characteristics | The research sample for the experimental component of this study consists of a nationally representative sample of the U.S., as curated by the crowdsourcing platform Prolific. The details on Prolific's U.S. nationally representative sample are provided by Prolific at the following link: https://researcher-help.prolific.co/hc/en-gb/articles/360019236753-Representative-samples. The sample population is representative of the national US population along the following demographic variables: age, sex and ethnicity. |
|---|---|
| Recruitment | For the experimental component, our sampling strategy was a random sample from Prolific's nationally representative panel (N=600, see "Research sample"). |
| Ethics oversight | This research was approved by the Institutional Review Board at the University of California, Berkeley, where the study was run. |

Note that full information on the approval of the study protocol must also be provided in the manuscript.

