## [Peer Review File · Nature]

Manuscript Title: Online Images Amplify Gender Bias

Reviewer Comments & Author Rebuttals

Reviewer Reports on the Initial Version:

Referees' comments:

Referee #1 (Remarks to the Author):

A. The paper shows that gender bias is amplified in online images compared to bias in text, public opinion, and occupational gender statistics obtained from U.S. Bureau of Labor Statistics data. In addition, human subject experiments reveal that searching for visual occupation data rather than textual descriptions amplifies implicit associations. The approach consists of methods and tools that have been used in similar studies. Bias in language is measured through pre-trained word2vec and GloVe word embeddings.

B. Although the study is interesting, the components and findings are not novel.

Kay et al. 2015 "experimentally evaluate the effects of bias in image search results on the images people choose to represent those careers and on people's perceptions of the prevalence of men and women in each occupation. We find evidence for both stereotype exaggeration and systematic underrepresentation of women in search results. We also find that people rate search results higher when they are consistent with stereotypes for a career, and shifting the representation of gender in image search results can shift people's perceptions about real-world distributions." Kay et al. use data from the U.S. Bureau of Labor Statistics. Kay et al. also measure perceptions of gender proportions in occupations without influence and after influence. They find that "while a person's original perceptions of an occupation dominated their opinion two weeks later; approximately 7% of a person's subsequent opinion on average was determined by the result set they were exposed to ($p < 0.01$, see Table 3)."

- Kay, M., Matuszek, C. and Munson, S.A., 2015, April. Unequal representation and gender stereotypes in image search results for occupations. In Proceedings of the 33rd annual acm conference on human factors in computing systems (pp. 3819-3828).

Caliskan et al. 2017, show occupation-gender associations with a Pearson's correlation coefficient $r = 0.90$ with $P < 10^{-18}$ in Figure 1, using data released by the U.S. Bureau of Labor Statistics in 2015. The study uses word embeddings (GloVe and word2vec) trained in 2014 to quantify gender bias.

- Caliskan, A., Bryson, J.J. and Narayanan, A., 2017. Semantics derived automatically from language corpora contain human-like biases. *Science*, 356(6334), pp.183-186.

C. There are various limitations and confounders that have not been addressed.

Training corpus details for word2vec and GloVe are missing. For example, are word embeddings trained on corpora collected before 2014 being used to compare with U.S. Bureau of Labor Statistics

from 2019? Sufficient details about the word embeddings are not provided.

Experimental searches are run over servers in New York City. However, as noted in supplementary materials, bias is location dependent. Location bias is not taken into account in the experiments.

Figure 1 contains phrases to represent occupations; however, word embeddings typically contain vector representations for individual words unless phrase embeddings are used. How are the embeddings for phrases obtained? For example, the word "professional" in "professional dancer" is male associated. Methodological details are missing. Polysemy and, whenever possible, word frequency needs to be taken into account.

Ethical concerns: How can Amazon Mechanical Turk workers annotate an individual's race and gender identity in a photo? Non-binary gender is included as a category. How can a third party determine someone else's identity? These need to be self-reported labels.

"the OpenCV module in Python to automatically identify and extract the faces from each of the 100 images associated with each search." There are two issues here. First, running experiments with 100 noisy images might not be sufficient. Second, automated face recognition tools are biased. They perform most accurately on samples from white men.

"Specifically, the female centroid consists of the words woman, her, she, female, and girl, whereas the male centroid consists of the opposite words man, his, he, male, and boy." Using the embeddings of 4 words might not be sufficient to obtain a statistically robust representation of a concept.

D. It is not clear why the dimensionality of the word embeddings is evaluated in this context. Evaluating context window size in training word embeddings on corpora collected in 2019 would be a justifiable machine learning setting. The context window size determines how much syntactic vs. semantic information is being captured.

E. The implicit association tests do not have controlled settings. The magnitude of bias in textual descriptions is not measured. Accordingly, quantifying its influence on subjects' implicit associations does not provide a comprehensive analysis. Moreover, it is unclear if stereotype threat or exposure to a study of gender bias in data is causing amplified implicit gender bias scores. The experimental settings need to account for all possible hypotheses to justify the findings.

F. Please note the exact type of implicit bias in the labels of charts in figure 3.

G. Significant references that show almost identical work are missing. Two of them are noted above. There is vast literature on gender bias in information retrieval, computer vision, socio-technical systems, and human-computer interaction.

H. Precision in language can be improved. "Here, we develop an auditing framework for comparing gender stereotypes across online images and texts for thousands of social categories, which we apply to Google and Wikipedia." What does applying to Google and Wikipedia mean?

Referee #2 (Remarks to the Author):

AUTHORS

This seems like an intriguing paper on the gender gap in online images. Given that we all increasingly rely on online search engines, existing biases might be reproduced if those search engines are structurally biased in some way – i.e., with respect to one or more social categories. What the authors convincingly show is that Google images with regard to a set of pre-defined social categories (e.g., athlete, doctor, and so forth) are highly gendered; that is, women seem to be disproportionately absent in some Google images (e.g., doctors) or highly overrepresented (e.g., nurse) when querying those social categories. The authors compare this with the gender gap with respect to those same social categories in large text corpora and find that the gender gap is stronger in images. They also report on an interesting experiment where they argue that this gendering in pictures reinforces gender stereotypes. I congratulate the authors on this relevant study that is well-carried out and that would likely be very appealing to a broad readership.

In what follows is a list of concerns I have with this paper that led me to not (yet) fully trust the results nor its associated claims.

1. Much of the paper leans on the claim that the gender gap is stronger in images than in text. My most pressing concern is whether or not the authors' claim may rest on two different types of media that are inherently incomparable from the outset. I read and appreciate the supplemental materials that describe well the harmonization between the two metrics for how gendered given social categories are. Yet, does the fact that the textual metrics' range initially is far less "absolute" than the pictures not show that very different data-generating processes are underlying these data sources? Writing about certain social categories might occur in a relatively less biased way by design, whereas the coders are actually pushed into identifying certain genders in the images. The high-dimensional word2vec might capture far more of those nuances compared to coding up the binary genders. So the coding of the images compared to measuring the genderedness of the social categories in text might be incomparable and/or by design differ. That begs the question: how surprising are these results given the media under consideration and given the approaches of the authors? I suggest the authors at the very minimum answer these questions conceptually as I worry that otherwise the paper may overclaim how surprising this finding is. A more fascinating approach would be to actually tackle these questions empirically or theoretically: can the authors devise a way that such design effects are accounted for? Or can they simply theorize and conjecture about the difference a priori?

2. What are the mechanisms underlying the gendered images on Google? I appreciate the approach of the authors that accounts for browsing histories, but without knowing anything about how and why these pictures come to be posted on websites and are then shown on Google it remains quite difficult to comprehend what is going on in the paper's data. Is the same algorithm underlying Google images compared to regular Google queries? So the overarching question here is: does the bias in the Google images underlie some real world data-generating process. The paper claims that it is, but it remains ambiguous what that is. I suggest the authors do more conceptual and theoretical work on these questions so as to support their claims.

3. The authors describe a number of associations with respect to the experiment on images and textual descriptions. This is an intriguing experiment, yet the exact mechanisms underlying the findings are somewhat underdeveloped. What is the micro-level mechanism here? Is there something about visual cues being more important? What does the literature conjecture about that? And do the authors know something about how well respondents read the textual descriptions? If not, the authors are comparing two different treatments that may (again) be inherently incomparable.

4. I think a statement such as “Textual measures of gender stereotypes are at risk of underestimating the prevalence of gender bias online” needs more development. I think the authors mean that when looking at different types of media or expressions online (e.g., avatars, news, images) each have different patterns of gender bias that may in some cases be stronger than in other cases. (Which might be expected.) I suggest the authors develop their premise based on a theoretical conjecture, instead of sketching it as a surprise, that some of these expressions might be stronger in some media compared to others (see point #1). They can then ask (and answer) the question what real-world implications that has in their experiment.

5. My browsing history is probably pretty biased, but when I Googled for “athlete” I found: groups, blurry faces, side by side shots of athletes, book covers, watches, fashion models, laptops, cartoons, ensemble shots of athletes, fashion models, infographics, food supplements, and so forth. Can the authors give a bit more information on the data quality in that respect and how they handled these things? Is the number of actual faces recognized non-randomly distributed over the categories? And what if, say, a non-athlete sport journalist enters into the image results of “athlete”?

6. I found the race question to coders to be somewhat problematic. The coders are asked to label races: “African American, Caucasian, Asian, Hispanic, Native American, two or more races, other”. So what happens if they find a Black non-US athlete? In other words, the coding is done with a US-centered approach, whereas Google image results are likely to provide international results for many social categories for which the categories are not applicable. Can the authors give more information about that?

7. I was unsure whether the gendered social categories were excluded beforehand? I suggest the authors make the analyses on the social categories that exclude the gendered social categories the central analyses of their paper. (And what is the distributions of genderedness in these social categories?) And are there more social categories going to one side compared to other sides (e.g., more husbands, uncles, etc)?

8. I am a bit concerned whether the definition of the “stereotype” that the authors give is one that will match the literature on stereotypes; is overrepresentation in category a stereotype (which might be an empirical finding)? Or is the expectation of certain genders in a category a stereotype? Notice that this is a nuanced but important difference.

9. In what way does the study contribute to the study mentioned under reference #21?

10. I doubt whether the analysis on Google searches for the top-100 images for the gendered

versions (e.g., female athlete/male athlete) is useful to the authors' argument: finding that both versions each retrieve 100 pictures is only logical given how many websites Google can retrieve those pictures from (millions?). This goes back to my point on the data-generating processes with respect to these pictures. So the conclusion that the availability of pictures for the genders is not different seems inaccurate. The second part of that conclusion ("algorithmic differences") may be correct, but the authors need to develop more intuitions about that (see point 2).

Referee #3 (Remarks to the Author):

Nature

2021-10-17102A-Z

May 4, 2022

This paper examines the extent to which the kinds of gender biases that have been documented in large-scale audits of online texts are also found in online images. I really enjoyed reviewing it, and my impression is overall enthusiastic. The main findings are that there is an overrepresentation of men vs. women in online images, that men are more likely to be pictured in stereotypically-male roles whereas women are more likely to be pictured in stereotypically-female roles, and that these patterns are exaggerated in online images when compared to census data or online text. The study also shows that exposure to these gender biases in online images increases implicit gender biases relative to exposure to online text.

The research question is worth investigating. The paper is well-written, the methodology is rigorous and innovative, and the analyses are thorough. I have a few questions, comments, and some suggestions that I hope will be helpful to the authors and contribute to making this paper even more impactful.

1. I really appreciate that the authors not only document the gender bias in online images but also test the consequences of exposure to such biases using experimental methods (also a longitudinal component). However, they only compare the effects of images vs. text, without a true control condition. It would seem informative to include a true control in order to get insights into participants' baseline gender biases and how those may be impacted by exposure to images/text. This is an open question that might be worth addressing in the paper's discussion or future directions.

2. I'm impressed by the scale of the image analysis--298,600 images compared to the vast majority of content analyses, which use only a few hundred or sometimes (rarely) a few thousand. That said, I'm curious as to the inter-coder reliability of the 2500 Mturk coders. Although they were coding only target gender (which is usually pretty straightforward and objective), I also think that it would be informative to have a sense of how reliable these coders were. Usually, content analysis employs 2-5 coders, and reliability metrics for each variable are reported (e.g., Krippendorff, 2004; Neuendorf, 2002). This is not a deal-breaker in my opinion because, again, the only variable coded was target gender; still, if there was any information that the authors could provide as evidence of high

reliability among the human coders, that would increase the credibility of the data.

3. Even with all the precautions taken to gather the images, was there any processing to ensure that each image in the dataset was unique? And, what about non-human images (e.g., icons, drawings, animations)? These are fairly common in Google Images when searching for any kind of occupational category. Were those included or filtered out? Could this point be made clearer in the paper?

4. I was confused about the number $N=5,972$ (last paragraph on p. 3). There were 2,986 social categories in Wordnet with word embeddings in the Google news corpus; why is each category counted twice toward the N ? Figure 1A clearly states that 2,986 categories are included in the analysis. Maybe I'm missing something here, but it would be helpful for the authors to clarify.

5. In Figure 2D, why focus on images of male-dominated occupations only? It would also be informative to see whether the same extreme representations (relative to census and text) is evident for images of female-dominated occupations.

6. I found the authors' definition of stereotypes on p. 1 to be a bit unconventional ("the over-representation of a demographic in the definition of a social category (e.g., female nurses)"). It seems to me like this is the operational definition of "gender stereotype" in the current research, but as a conceptual definition of the construct of stereotype it seems rather limiting.

Within the psychological literature, stereotypes are more commonly defined as: "general expectations about members of particular social groups" (Ellemers, 2017) (where "social group" refers to demographic groups such as men/women but could also refer to many other groups, e.g., "church goers"). Other definitions are more specific about what these "expectations" are about, e.g., "assumptions about traits and behaviors that people in a labeled category are thought to possess" (Kite, Deaux, & Haines, 2008). Gender stereotypes, which are of central interest in the authors' research, are usually defined along the lines of: "consensual beliefs about the attributes of women and men" (Eagly, Nater, Miller, Kaufmann, & Sczesny, 2020).

Similarly, in one of the references the authors cite for their definition of stereotype (Gorman, 2005), the author wrote: "Gender stereotypes are cultural constructs, shared at the societal level, that describe what men and women are "known" to be like (Fiske 1998:377–78)". In sum, stereotypes in general and gender stereotypes in particular are usually defined more broadly than by alluding to the distribution of people from a given demographic group into a social category (which, in the context of this paper, refers to an occupational role or social role such as "cousin"). I would strongly recommend to either revise the definition or clarify that it refers to the operationalization of stereotype within the current research.

7. It was unclear what was meant by "the digital gender gap."

8. It seems like one possible explanation for why online images may show more bias than online text is that online text may be more likely than online images to refer to real people. Even if there are few women in a given male-dominated occupation, something may be written about those women. Images, in contrast, are less likely to refer to real people—in many cases, online images involve

models (e.g., istock photos). Thus, online images are more likely to show a category exemplar or prototype, whereas text may include a wider range of gender representations.

9. I understand the focus in the discussion on the potential negative effects on the well-being and economic opportunities of women. But it may also be worth mentioning that the effects apply to occupational gender segregation at large. For example, these algorithmic biases can also contribute to the low representation of men in care-oriented domains.

Referee #4 (Remarks to the Author):

Signed: Yoav Bar-Anan

The focus of the manuscript is on a comparison of the association between gender and social categories (occupations but also other “social roles”) inferred from online texts versus photos found in Google search or in Wikipedia. The authors’ main conclusion from that comparison is that the associations are stronger for images than for text. The estimate of associations for the images data was computed as how pre-dominantly male or female were the faces in the images within each social role (as opposed to a 50-50 split between the genders). In their caption for Figure 1, the authors explained that the estimate of associations for the textual data “captures the frequency at which each category is associated with male or female terms in the Google News corpus of over 10 billion words”.

Another finding is that for image search (and Wikipedia) more categories were associated with males than with females.

The last finding comes for an experiment that required participants to perform a Google search for images or text, pertaining to 22 occupations, related to science, technology, and art. After that search, participants reported their association of each occupation with gender, and also completed an Implicit Association Test (IAT), that is sensitive to people’s associations between gender and science vs. Liberal Arts. The authors found a correlation of 0.30 between the gender associations found in the participant’s Images search and the IAT score. The authors also found that participants assigned to the image search condition reported stronger associations between occupations and gender (i.e., they more strongly associated a specific gender with a specific occupation).

I am highly impressed by this huge undertaking, and I believe that it reports great studies with a good potential for high impact. I believe that the publication of this research would make a positive contribution to human knowledge. I believe that this paper would attract much interest from various disciplines and the general public.

There were, however, some aspects in the manuscript that the authors might want to consider for a possible revision:

1. My expertise pertains only to the study of implicit social cognition, rather than biases in Internet

media. I am unfamiliar with any study that examined the effect of Google search on self-reported gender stereotypes and performance in the IAT. This is a very nice experiment, with relatively high ecological validity.

There are a bunch of studies about the effect of exposure to information on the IAT, even specifically to the IAT that was designed to measure gender stereotypes (e.g., Asgari et al., 2012; Desgupta & Asgari, 2004; Ramsey & Sekaquaptewa, 2010; Stout et al., 2011). Some of those studies were longitudinal, showing an effect over time. Still, there is something unique in having participants search by themselves, rather than providing them the information directly. The present experiment is a good demonstration of the possible effects of the biases reported by the authors on people's beliefs and stereotypes. In my field, this experiment would be highly interesting and impactful. However, the authors should probably acknowledge that the IAT is not a pure measure of (implicit) stereotypes, and it is sometimes sensitive to factors other than evaluations and beliefs. Critics of the IAT would be quick to argue that the present experiment found an effect of accessible information on IAT performance, which may, or may not, reflect a change in people's attitudes and beliefs.

As for the main part that does not pertain to the IAT, I think the authors might want to emphasize more clearly what the novelty of this research is. It seems that there is already previous work that showed that Google images provides gender-biased results (Kay et al., 2015; Metaxa et al., 2021). Therefore, is the novelty of this research only the comparison between images and text? Or is it perhaps the magnitude of the present research (e.g., more categories)? This point was not clear to me.

2. In general, the methodology, data processing, and analyses seem highly professional and adequate with no serious red flags. However, some aspects were unclear to me, and might require further clarification. First, regarding the experiment, I did not see in the main text or in the supplement document analyses that tested directly the effect of the manipulation on the IAT. The closest I have seen to such a test is in Figure 3D, showing a relation between the measured value of the manipulated variable (Gender ratios in the search performed by the participant) and the IAT score. However, this still leaves a room for an effect of the participant's stereotypes on the predictor (e.g., the participant's choice of images to upload might be influenced by the participant's gender stereotypes). Therefore, it would be best to see analyses including the effect of the manipulation on the IAT score (especially because the analysis in the Figure 3D actually omits the participants from the text condition).

3. One issue that confused me throughout the reading is that most of the variables pertained to gender-bias in absolute terms – how different the score is from a score that reflects a 50-50 presentation (or association) of the social category with male and female. In fact, this was always the case excluding the IAT score (see the range of the y-axis in Figure 3D). That means that the graph in Figure 3D reflected a correlation between general gender-imbalance in Google image search and a specific bias (Men with science, women with Liberal arts) in the IAT. The use of absolute score was surprising because it ignores the question of whether the imbalance in the gender rate reflects the known gender stereotypes. It was not explained in the main text or the supplement why the authors

preferred to look at such scores instead of trying to quantify how much their scores reflect the traditional stereotypes. I can think of good justifications for that, and I suspect that this was probably the correct decision, but it was quite confusing, so it requires an explanation.

4. The overlap between the pre-registration (<https://osf.io/26kbr>) and the reported study does not seem ideal. For example, the pre-registration mentioned using a set of “over 200 occupations” but the main text reported using 54 occupations. More importantly, the pre-registration did not include most of the tests reported in the main text and the supplement, and referred mostly to the comparison between the text and the image search. And, as I mentioned earlier, although the pre-registration includes the analysis of the condition (text vs. image) on the IAT scores, I did not see that test reported in the main text or the supplement.

5. I wonder whether the textual condition in the experiment was the textual equivalent of the image condition. The textual condition seemed less constrained – it was less clear what the participant should do, in comparison to the image condition. For example, I guess that I would have added the word “define” in the textual Google search, and would have just pasted the definition. Or, perhaps I would have added the word “Wikipedia” and would have entered the definition at the beginning of the Wikipedia entry. In both cases, it is not clear whether it is the same as searching in the Google image, which is more clearly related to the question “how does a geneticist look” or “what is a good example for a geneticist” rather than what a geneticist is (for the latter, people would probably not use Google images). Perhaps a more comparable search would have been for the category in Google News.

6. I assume that it is not straightforward to determine whether the measurement of bias in text vs. image were easily comparable. I tried to think of methods to validate the similarity of these two measures using associations that are expected to be the same in both databases, but I could not think of such associations. So, I am raising this issue here, just in case the authors would have an idea.

Author Rebuttals to Initial Comments

Reviewers' comments

Referee #1 (Remarks to the Author):

1. *Although the study is interesting, the components and findings are not novel.*

Kay et al. 2015 "experimentally evaluate the effects of bias in image search results on the images people choose to represent those careers and on people's perceptions of the prevalence of men and women in each occupation. We find evidence for both stereotype exaggeration and systematic underrepresentation of women in search results. We also find that people rate search results higher when they are consistent with stereotypes for a career, and shifting the representation of gender in image search results can shift people's perceptions about real-world distributions." Kay et al. use data from the U.S. Bureau of Labor Statistics. Kay et al. also measure perceptions of gender proportions in occupations without influence and after influence. They find that "while a person's original perceptions of an occupation dominated their opinion two weeks later; approximately 7% of a person's subsequent opinion on average was determined by the result set they were exposed to ($p < 0.01$, see Table 3)."

- Kay, M., Matuszek, C. and Munson, S.A., 2015, April. Unequal representation and gender stereotypes in image search results for occupations. In Proceedings of the 33rd annual acm conference on human factors in computing systems (pp. 3819-3828).

Caliskan et al. 2017, show occupation-gender associations with a Pearson's correlation coefficient $r = 0.90$ with $P < 10^{-18}$ in Figure 1, using data released by the U.S. Bureau of Labor Statistics in 2015. The study uses word embeddings (GloVe and word2vec) trained in 2014 to quantify gender bias.

- Caliskan, A., Bryson, J.J. and Narayanan, A., 2017. Semantics derived automatically from language corpora contain human-like biases. Science, 356(6334), pp.183-186.

We appreciate this thoughtful engagement with our work. We identify four key ways in which our study marks a novel and impactful advance upon these prior studies:

(1) Comparing text to images (observational). The prior studies referenced by Reviewer 1 focus on gender bias either in images (e.g., Kay et al. 2015; Metaxa et al. 2021) or text alone (e.g., Caliskan et al., 2017). We can find no prior work that identifies differences between texts and images in how they encode gender bias. We address this gap by developing computational methods for comparing gender bias across images and text, including a novel experiment to compare the ability of images and text to reinforce gender bias in people's beliefs.

(2) Comparing text to images (experimental). Our experiment, which we have improved and replicated with a control condition, not only reveals that gender bias is more prevalent in online images than text, but also that images more strongly prime gender bias than online texts. This involves a novel integration of prior research on

the “picture superiority effect” and prior work on priming effects in the study of gender bias. Neither Kay et al. (2015) nor all related studies we are familiar with identify this foundational difference between images and text and demonstrate its psychological consequences, which have far-reaching implications for image-based search engines and social media.

(3) Insights into the effects of images on implicit bias in an ecologically-valid experiment. Kay et al. (2015) and Metaxa et al. (2021) examine the ability for gender bias in online images to prime biases in people’s explicit beliefs. We go beyond by measuring the impact of texts and images on both explicit and implicit bias over time. This required integrating the Implicit Association Test (IAT) into a novel online experiment in which participants used Google to search for content in real-time. These efforts reveal that only online images significantly and enduringly increased participants’ implicit gender bias. See our reply to (4.) below for an explanation of the robustness of our IAT. Another novel feature of our experiment is that it required participants to leverage Google to search for information in a realistic manner, whereas Kay et al. (2015) and Metaxa et al. (2021) test the effects of exposing people to artificially created image sets without engaging them in a search task. This lends our design considerable ecological validity, which is highlighted by Reviewer 4 as another impactful contribution of our experiment.

(4) Scale. Kay et al. (2015) and Metaxa et al. (2021) only investigated roughly 5100 images mapped to 51 occupations. Our dataset is almost 100 times larger. We study gender bias in the images associated with nearly 700 occupations, and 3,495 social categories in total, not only from Google images, but also Wikipedia; altogether, our sample consists of over 600,000 images. Furthermore, our revised SOM tests our theory using a dataset of over 500k images from IMDb and Wikipedia. Combined, our entire image dataset comprises over one million images, and is mapped to the full known taxonomy of social categories in the English language, enabling a range of analyses infeasible for prior research.

2. There are various limitations and confounders that have not been addressed.

Training corpus details for word2vec and GloVe are missing. For example, are word embeddings trained on corpora collected before 2014 being used to compare with U.S. Bureau of Labor Statistics from 2019? Sufficient details about the word embeddings are not provided.

Thank you for identifying this important need for clarification. As Reviewer 1 notes, our main analyses leverage the canonical word2vec model, pre-trained on Google News in 2013. Our SOM now demonstrates the robustness of our results while comparing against six popular embedding models, each with different algorithmic specifications and with training data collected as recently as 2020. Table S1 (provided below) specifies the training corpora and algorithmic details for each embedding model we analyze. These supplementary revisions are copied below. Furthermore, as we elaborate in response to comment 8 from Reviewer 2, we also show how the gender associations detected by the canonical 2013 Google News pre-trained model

are highly predictive of the gender associations detected in a sample of Google News descriptions of occupations which we collected in 2022.

Credit: AGI, data derived from the U.S. Census Bureau and U.S. Bureau of Labor Statistics

Memo Fig. 1. The distribution of women across occupations (by industry) from 2005 to 2019, according to data from the U.S. Census Bureau and the U.S. Bureau of Labor Statistics.

Regarding the validity of comparing word embedding models trained on data from different time periods to census data from 2019, we wanted to highlight that the distribution of women across occupations has remained stable over the last decade. This is the focus of recent work highlighting the stalling of gender equality across industries (see England et al. 2021). This result can also be seen in the above figure based on U.S. census data (AGI, 2020), which uses U.S. census data to show almost no change in the distribution of women across occupations for nearly all industries reported from 2013 to the present.

Memo references:

American Geosciences Institute (AGI). 2020. Diversity in the Geosciences. https://www.americangeosciences.org/sites/default/files/DB_2020-023-DiversityInTheGeosciences.pdf
 Paula England, Andrew Levine, and Emma Mishel. Progress toward gender equality in the United States has slowed or stalled. *Proceedings of the National Academy of Sciences*, 117(13):6990–6997, 2020.

SOM section specifying all word embedding models examined (section A.2.3):

“Here, we show that our main results equally hold when comparing our Google Image data against a wide range of state-of-the-art embedding models. These models vary along a rich array of methodological parameters, including window size, data sources, the date of the data collection (ranging from 2013 to 2020), along with the algorithms they use to generate word vectors based on the distribution of words and characters. Table S1 describes each embedding model we compare against. All models are trained on online textual corpora that contain many billions of word tokens.

Model	Paper	Year	Window Size	Dimensions	Datasets	Size
word2vec-googlenews	Mikolov et al. (2013)	2013	10	300	-Google News (2013)	100 billion word tokens.
glove-gigaword	Pennington et al. (2014)	2014	10	50, 100, 200, 300*	-Wikipedia (2014) -Gigaword News Archive (2003)	5.6 billion word token.
glove-twitter	Pennington et al. (2014)	2014	10	50, 100, 200, 300*	-Twitter (2014)	27 billion word tokens.
fasttext-wiki-news-subwords	Mikolov et al. (2018) *Facebook	2017	5	300	-Wikipedia (2017) -UMBC -Statmt.org	16 billion word tokens.
Conceptnet 5.5	Speer et al. (2017)	2017	Ensemble of wordnet, glove, and conceptnet	Ensemble	-ConceptNet Graph, -Google News (2013, word2vec) -Wikipedia (2014, glove) -OpenSubtitles 2016	Full corpora for 10 languages.
CharacterBERT	Boukkouri et al. (2020) *Google	2020	NA	768	-Wikipedia (2020) -Brown Corpus -OpenWebText	2.14 billion word tokens from Wikipedia and 1.2 billion tokens from crawled website HTMLs.
GPT3	Brown et al. (2020) *OpenAI	2020	NA	Estimated at 12,288	-Common Crawl -WebText2 -Books1 -Books2 -Wikipedia	Common Crawl: 410 billion tokens WebText2: 19 billion tokens Books1: 12 billion tokens Books2: 55 billion tokens Wikipedia: 3 billion tokens

Table S1. Summary of the word embeddings models examined in this study.

Table S1 reveals the rich diversity of model designs leveraged in our analysis. For example, we examine Word2vec [10] and GloVe models [6] which use the same window size but which vary in the online sources of text they examine (including Google News, Wikipedia, and Twitter). We also evaluate our results using Facebook’s FastText embeddings [9], which vary from prior models by using a window size of 5, while also relying on a more recent sample of Wikipedia from 2017. We further examine our results using embeddings from ConceptNet [11], which is an ensemble approach to building word embeddings that combines Word2vec and Glove, along with various sources of human crowdsourcing, manual lexical ontology construction across 300 unique languages, and a large 2016 sample of online news. We also evaluate our theory using a fundamentally different kind of model – CharacterBERT [12] – which generates word embeddings by examining co-occurrence patterns at the level of characters as well as words (n-grams); this model tracks full character sequences at the level of both words and sentences in a contextually-dynamic manner, such that window size varies at the word and sentence level and is therefore not a fixed contextual window like the other models. What is more, CharacterBERT is trained on a more recent 2020 Wikipedia dataset, as well as a comprehensive 2020 sample of scraped texts from public websites all over the open web. Lastly, we compare our results against a recent, super-large language model, GPT-3 [13], which uses a novel approach (“Generative Pre-trained Embeddings”) and which possesses substantially more dimensions compared to other models, as well as a flexible and large window size; this method relies on textual data from a combination of Wikipedia, online books, and a representative random sample of websites known as the common crawl.

For all word embedding models, the gender associations of each category were calculated using exactly the same methodology we applied to the Google News corpus in the main text [5]: (1) a one-dimensional gender vector is constructed consisting of two poles, the ‘male’ and ‘female’ pole; the male pole consists of words in vector space relating the men, such as *male, man, he, and his*, and the female pole consists of the mirror feminine terms, such as *female, woman, her, and hers*; (2) then, to identify the gender association of each category of interest, one calculates the average pairwise cosine distance between this category and each gender pole, with the effect of locating this category along this gender dimension, ranging from -1 (female) to 1 (male). Where a social category falls along this gender dimension captures the frequency with which this category co-occurs with words relating to either women or men. To maximize the correspondence between our text-based and image-based measures, we apply min-max normalization separately to each text-based model of gender associations (i.e., in Wikipedia, Twitter, and Google News), so that -1 and 1 represent the most female and male categories respectively according to each measure.

In what follows, we show that our core results equally hold across all of these diverse models. In this way, we show that our results are robust to varying the core aspects of word embedding models, including window size, data source, time of data collection, and algorithmic method for generating embeddings.

Figure S5: (A) The distribution of gender associations for all categories, according to all models. (B) The overall strength of gender association for all categories, according to all models. See Table S1 for a full description of each model.

Fig. S5A shows that the gender association in Google images are significantly more male than all language models compared against (all comparisons between our image data and each word embedding model are significant at the $p < .0001$ level; Wilcoxon Signed-Rank Test, Two-sided). In fact, both GPT-3 and BERT

exhibit significant biases toward female associations, in contrast to male over-representation in online images. Fig. S5B further shows that the absolute strength of the gender associations in Google images is also significantly stronger than all language models (all comparisons between our image data and each word embedding model are significant at the $p < .0001$ level; Wilcoxon Signed-Rank Test, Two-sided). Both of these findings are even stronger if we compare our image data against the non-normalized distribution of gender associations in each word embedding model, especially for GPT-3 and BERT which have the highest dimensionality and exhibit weak differentiation of categories along the gender dimension in their raw gender associations prior to normalization.”

3. Experimental searches are run over servers in New York City. However, as noted in supplementary materials, bias is location dependent. Location bias is not taken into account in the experiments.

This is an excellent point. We have added a new robustness test to our supplementary material that replicates our pipeline for 300 categories across five additional IPs corresponding to distinct countries: (i) Amsterdam, Holland; (ii) Bangalore, India; (iii) Frankfurt, Germany; (iv) Singapore, Republic of Singapore; and (v) Toronto, Canada. We show not only that male over-representation persists across all four additional locations, but also that the gender distribution for each category is highly correlated across these IPs (including New York). This is particularly striking since we also show that the images that appeared for the same searches across these locations were distinct and minimally overlapped, consistent with location-specific effects on search results. The supplementary analyses corresponding to this robustness test are copied below, from pg. 19-22 of the SOM:

“Google is known to tailor search results to IP location [14]. Here, we show that our results hold when collecting Google images from 5 additional IPs from around the world. We tested the generality of our Google results by analyzing the top 100 images associated with 300 categories when searching from six distinct locations: Singapore (the Republic of Singapore), Frankfurt (Germany), Bangalore (India), Toronto (Canada), and Amsterdam (the Netherlands). These IP locations were selected based on those available through DigitalOcean, a public VPN provider. All searches were run during the same time window. The 300 categories comprised 256 occupations that could be mapped between our WordNet dataset and the US census data, along with a random selection of additional categories (N=44). The gender of images in this replication were classified using the same procedure in our main study, whereby the final gender classification associated with each face was determined by the modal (majority) gender classification across three unique coders. We hired 1,223 annotators to categorize these images. Again, all coders were based in the US and were fluent English speakers.

Figure S7: Comparing the Google images across search locations (IPs) from distinct countries, namely: (i) New York (USA), (ii) Singapore (the Republic of Singapore), (iii) Frankfurt (Germany), (iv) Bangalore (India), (v) Toronto (Canada), and (vi) Amsterdam (the Netherlands). This figure shows the average overlap in the images that appeared for the same search queries across all search locations. Overlap is measured via the Jaccard Index, defined as the proportion of common elements between two sets compared to the total number of unique elements in both sets.

First, we examine the extent to which the image search results across IPs were similar for the same searches. We conducted this analysis by using image metadata and pixel-level comparisons to identify whether an image repeated across searches. Consistent with location-specific search results, figure S7 estimates that on average, only 21% of the image search results were the same across locations. Any consistency in the patterns of gender bias across these sources cannot, therefore, be attributed to stability in Google’s search results across IPs. Instead, these results indicate that changing the geolocation of one’s IP produces qualitatively distinct image search results in Google.

Figure S8: (A) The distribution of gender associations for 300 social categories in Google Images collected via online searches from five distinct locations (IPs) from distinct

countries, namely: (i) Singapore (the Republic of Singapore), (ii) Frankfurt (Germany), (iii) Bangalore (India), (iv) Toronto (Canada), and (v) Amsterdam (the Netherlands). The image-based measure captures the frequency of male and female faces associated with each category in Google Image search results (-1 means 100% female; 1 means 100% male). Vertical lines display the average gender associations across categories for each IP; the dotted vertical line indicates perfect gender balance (0 along the gender dimension). (B) The pairwise correlations in the gender associations of categories as they appear in Google images collected via online searches from these distinct IPs.

Despite the differences in images returned by Google searches from different IPs, we nevertheless find strikingly similar patterns in the gender associations of categories in these distinct image sets. Fig. S8 shows that, for all IPs, the categories examined skew significantly toward male representation in Google images ($p < .001$ for all IPs, Wilcoxon Signed-Rank Test, Two-tailed). Moreover, the gender bias displayed in the Google Image search results across these IPs is significantly stronger than the measure of gender bias according to word embedding models of Google News, as reported in the main text ($p < .001$ for all IPs, Wilcoxon Signed-Rank Test, Two-tailed). Panel B of Figure S8 displays the pairwise correlation in the gender associations of categories as they appear in Google images from these distinct IPs. The gender association by categories is highly and significantly correlated across IPs ($p < .001$ for all pairwise comparisons measured using either Pearson or Spearman correlation).”

4. Figure 1 contains phrases to represent occupations; however, word embeddings typically contain vector representations for individual words unless phrase embeddings are used. How are the embeddings for phrases obtained? For example, the word "professional" in "professional dancer" is male associated. Methodological details are missing. Polysemy and, whenever possible, word frequency needs to be taken into account.

We break our response into two parts: (1) phrasal categories and (2) polysemy.

(1) Phrasal Categories

Our dataset consisted only of unigram (one word) and bigram (two word) categories. We obtained word embeddings for all bigrams across all models in the gensim Python package by using the ‘Phrases’ gensim class (a prebuilt function within gensim for obtaining word embeddings for bigram phrases). We have added a sentence clarifying this technique on pg. 5 of the SOM:

“To compute distances between the vectors of social categories represented by bigrams (such as ‘professional dancer’), we used the Phrases class within the gensim Python package, which provided a pre-built function for identifying and calculating distances for bigram embeddings. This method works by identifying an n-dimensional vector of middle positions between the vectors corresponding separately to each word in the bigram (e.g., ‘professional’ and ‘dancer’). This technique then treats this middle vector as the singular vector corresponding to the

bigram ‘professional dancer’ and is thereby used to calculate distances from other category vectors. This same method was applied to the construction of embeddings for all bigram categories in all models.”

We undertook an additional robustness test to show that all of our main results hold if we only examine unigram categories, leaving 1,906 categories in the Google Image data set (none of the categories examined in our Wikipedia analysis were bigrams). Removing bigram categories has no impact on our main results. This robustness test is copied here:

“A.2.11 Robustness to Controlling for Linguistic Properties of Social Categories.

In this section, we show that our main results hold when controlling for a range of linguistic features of social categories, namely: number of ngrams, category ambiguity, the frequency of a category in everyday language use, and whether categories contain explicit gender connotations.

Figure S14: The distribution of gender associations in images from Google Images and texts from Google News for 1,906 social categories (excluding all categories from our main analysis that are not unigrams, e.g., “professional dancer”). The image-based measure captures the frequency of male and female faces associated with each category in Google Image search results (-1 means 100% female; 1 means 100% male); the text-based measure captures the frequency at which each category is associated with male or female terms in the Google News corpus of over 100 billion words (-1 means 100% female; 1 means 100% male). Solid green (purple) line indicates the average gender association according to text (images).

Figure S14 shows that our main results hold when analyzing only social categories in Wordnet that consist of a single term (i.e., unigrams). Figure S14 compares gender associations across images and text using only the 1,902 unigram categories in WordNet. The results show that the Google Images for these categories skew significantly toward male representation, both in general

($p < .0001$) and in comparison to text ($p < .0001$) (Wilcoxon Signed-Rank Test). In this way, we show that our results are not driven by idiosyncrasies introduced by bigram categories. Note, the same trend equally holds for bigram categories ($p < .0001$).”

(2) Polysemy

Our appendix now includes an additional statistical model which shows that our main results hold when controlling for the polysemy and word frequency of each category. Polysemy was measured using WordNet by quantifying the number of distinct definitions associated with each category in WordNet’s architecture (see Iliev & Axelrod, 2017, below). Word frequency was measured by matching each social category to the Exquisite Corpus by Luminoso (made available through the python package *wordfreq*), which identifies the frequency of English words and phrases across Google News, Google Books, Wikipedia, Twitter, and Reddit (based on data collected as recently as 2021). These statistical models, and the prose describing them, are provided below (copied from pg. 37 of the SOM):

“Next, we use a regression approach to evaluate our results while controlling for category polysemy, which refers to when categories have multiple definitions. The category *doctor*, for instance, can refer to a doctoral graduate student or more typically to a clinician, such that *doctor* is more polysemous than the category *plumber*, which likely only refers to one possible group. We measured polysemy using WordNet, which lists the conventional definitions associated with each category; to measure polysemy, we counted the number of unique definitions associated with each category in WordNet [23]. In the same model, we also control for the frequency of each word in terms of how commonly it is used in daily language. Word frequency was measured by matching each social category to the Exquisite Corpus by Luminoso (made available through the python package *wordfreq*), which identifies the frequency of English words and phrases across Google News, Google Books, Wikipedia, Twitter, and Reddit.

Predictors	Strength of Gender Bias		
	Estimates	CI	p
(Intercept)	0.39	0.38 – 0.40	< 0.001
Measure [Text]	-0.17	-0.18 – -0.16	< 0.001
Word.Frequency.Scaled	0.01	0.01 – 0.02	0.007
Polysemy	0.00	-0.00 – 0.00	0.611
Observations	6411		
R ² / R ² adjusted	0.124 / 0.124		

Table S4: An OLS regression predicting a category’s overall strength of gender associations, as a function of (i) the data source (either images from Google Images or

texts from Google News), (ii) the frequency of each category's use in online texts, including Google News, Google Books, Twitter, and Reddit (scaled by standard deviation), and (iii) the polysemy of each category according to WordNet (where polysemy is measured as the number of different definitions associated with each category). Data are shown for all 3,495 categories. Standard errors are clustered at the category level.

Table S4 presents the results of an OLS regression predicting a category's overall strength of gender associations (from 0 to 1), as a function of (i) the data source (either images from Google Images or texts from Google News), (ii) the frequency of each category's use in online texts, including Google News, Google Books, Twitter, and Reddit, and (iii) the polysemy of each category according to WordNet (where polysemy is measured as the number of different definitions associated with each category). Table S4 shows that textual measures of Google News are associated with significantly weaker gender associations than Google Images ($\beta = -0.17$, $CI = [-0.18, -0.16]$, $p < .001$), controlling for the polysemy and the frequency of each category in the English language. Additionally, table S4 shows that the polysemy of each category fails to predict the strength of gender associations in either data source ($\beta < .01$, $p = .61$); similarly, word frequency is weakly associated with the strength of gender association with each category ($\beta = .01$, $p = .007$), suggesting that more frequent categories are associated with slightly stronger gender biases, controlling for whether the measurement of bias is made via text or images."

Memo References:

Iliev, Rumen, and Robert Axelrod. "The Paradox of Abstraction: Precision Versus Concreteness." *Journal of Psycholinguistic Research* 46, no. 3 (June 2017): 715–29.
<https://doi.org/10.1007/s10936-016-9459-6>.

5. Ethical concerns: How can Amazon Mechanical Turk workers annotate an individual's race and gender identity in a photo? Non-binary gender is included as a category. How can a third party determine someone else's identity? These need to be self-reported labels.

We are greatly sympathetic to these concerns. We do not mean to imply that our Mturkers' classifications are correct representations of people's self-identified gender and race. Our main interest is in the gender that internet users perceive in the faces they encounter online, rather than the "true" self-identified gender of the faces depicted online, which internet users typically do not have access to. We have added footnote 1 to our manuscript to clarify this point (copied below). Despite our focus on perceived gender, we nevertheless see the importance of testing whether our theory holds in contexts of online images where the self-identified gender of the people depicted is known. For this reason, we have added a new robustness test to our supplementary appendix, which confirms our hypotheses using canonical image datasets of celebrities depicted over IMDb and Wikipedia, where each image in these datasets is associated with the self-identified gender of the celebrity depicted (see Rothe et al. 2018). We also copy this supplementary analysis below.

Footnote 1:

“Our focus is on the gender that internet users perceive in the faces online, since it is infeasible to obtain the self-identified gender of most people depicted via Google Images. In supplementary analyses, we replicate our findings using a canonical image dataset of 72,214 celebrities depicted over IMDb and Wikipedia (yielding 511,946 images), where each image in this dataset is associated with the self-identified gender of the person depicted (Fig. S12). This analysis demonstrates that our theory holds in a setting in which the ground truth gender of online faces can be identified.”

From the *SOM* (starting on pg. 11):

“A.2.2 Robustness to Analyzing Ground Truth Measures of Gender in Online Images from IMDb and Wikipedia.

In this section, we validate our theory using the IMDb-Wiki dataset [8], which contains over half a million online images of celebrity faces for which the ground truth gender of the face is known via the celebrity’s public biographical page on either IMDb or Wikipedia. This dataset contains 460,723 faces of celebrities from IMDb and 62,328 faces of celebrities from Wikipedia (covering 72,214 celebrities in total); each of these faces is associated with the self-identified gender of the celebrity in the image. As described by Rothe et al. (2018) [8], this dataset was compiled by, first, identifying the top 100,000 celebrities according to IMDb (this dataset includes actors, as well as producers, directors, public figures, athletes, and more, such that “celebrity” is the most appropriate category of reference for this data, as the authors describe). Then, the authors automatically crawled from IMDb the name, age, and gender of the celebrities from this list, as well as all images associated with each celebrity over the IMDb website. The authors additionally crawled all profile images from pages of these celebrities from Wikipedia. They removed the images without a timestamp (the date when the photo was taken). In total, they obtained 460,723 face images from 20,284 celebrities from IMDb and 62,328 from Wikipedia, thus 523,051 in total. As an additional pre-processing, all faces are removed that are missing an associated gender identification. This resulted in 452,261 face images from 19,091 celebrities from IMDb and 59,685 face images from 58,904 celebrities from Wikipedia, and thus 511,946 faces in total.

The IMDb-Wiki dataset allows us to test our theory using true gender assignments independent of our human coders’ judgments. For this analysis, we used two different word embedding models of the Wikipedia corpus: the 300-dimensional 2014 GloVe model [6] and the 300-dimensional 2017 FastText model from Facebook [9]. In each model, we determine the gender association of the word “celebrity” using our standard method of constructing a gender dimension in word embedding space. Moreover, to identify the gender representation with the social category of ‘celebrity’ in the census, we used the aggregated gender

representation in the census' CPS category of occupations relating to "arts, design, entertainment, sports, and media occupations," which includes the occupations "actors", "directors", "musicians", and "entertainers/performers."

Figure S4 below presents the results of comparing the gender associations with the social category "celebrity" across these five datasets: the IMDb-Wikipedia Face Image dataset (for which the true gender of each face is known), the census (for which the true empirical distribution of gender is known), the gender associations in the 2014 GloVe word embedding model of Wikipedia text, and the gender associations in the 2017 FastText word embedding model of Wikipedia text.

Figure S4: Comparing the gender associations with the social category "celebrity" across four datasets: the IMDb-Wikipedia Face Image dataset (for which the true gender of each face is known), the census (for which the true empirical distribution of gender is known), the FastText word embedding model of gender associations in the Wikipedia dataset, and the GloVe word embedding model of gender associations in the Wikipedia dataset. Panel A shows the gender associations across these datasets (-1 means 100% female associations; 1 means 100% male associations; 0 means equally male and female associations). Panel B shows the overall fraction of male faces identified in the IMDb-Wikipedia Face Image dataset, shown separately for IMDb and Wikipedia. Error bars show 95% confidence intervals.

As panel A of Fig. S4 demonstrates, online texts encode more female associations for the category "celebrity" as compared to the census. Specifically, the gender association of the category "celebrity" is -0.05 according to the FastText model of Wikipedia texts and -0.08 according to the GloVe model of Wikipedia texts. Meanwhile, the census indicates that 49% of people in occupations relating to celebrities are women, resulting in a gender association score of 0.02 that fails to be significantly skewed toward a particular gender ($p = 0.54$). By contrast, online images from Wikipedia encode substantially greater male representation compared to both the census and Wikipedia texts. The gender association of the

category “celebrity” is 0.57 according to Wikipedia images. The same data is shown for IMDb images of celebrities, which are also significantly more biased toward male representation (0.16 along the gender scale) compared to these textual measures and the census. Panel B of Fig. S4 indicates that this skew toward male representation in online images is highly significant; over IMDb, 58% of celebrity images are of men ($p < .001$, Proportion Test, $N=482,261$), and over Wikipedia, 79% of celebrity images are of men ($p < .001$, Proportion Test, $N=59,681$). These results strongly support our theory using online images that are associated with verified ground truth gender classifications, indicating that our findings hold independently of the subjective gender judgments of human coders.”

Cited Reference

Rothe, Rasmus, et al. "Deep expectation of real and apparent age from a single image without facial landmarks." *International Journal of Computer Vision* 126.2-4 (2018): 144-157.

6. *"the OpenCV module in Python to automatically identify and extract the faces from each of the 100 images associated with each search." There are two issues here. First, running experiments with 100 noisy images might not be sufficient. Second, automated face recognition tools are biased. They perform most accurately on samples from white men.*

Section A.2.18 of the SOM shows that all of our results hold when collecting up to 500 images associated with each search category. We adopt four additional approaches to demonstrating the validity of our results while accounting for the limitations of facial recognition software, described below:

(1) Robustness to human classifications of uncropped images (section A.2.15, SOM). We recruited an additional 1,024 human annotators to classify the faces in the original, non-cropped images for a subset of 300 categories. We asked each annotator to classify the focal face in each image belonging to the specific category associated with this image in Google search (e.g., “doctor”). Consistent with our main results, we find that uncropped Google images present significantly more male bias, and significantly stronger gender associations overall, compared to textual measures of gender bias for the same categories (Fig. S23). This indicates that our findings are not an artifact of biases in our cropping algorithm, since the same findings hold in the absence of any cropping technique.

(2) Robustness to machine learning gender classifications of both cropped and uncropped images (section A.2.9 SOM). We used OpenCV to automatically classify the gender of the faces in our full dataset of cropped Google Images ($N=586,214$) and our full dataset of uncropped Google Images ($N=417,882$) (these analyses include images associated with gender-explicit searches, section A.2.10). Consistent with prior literature highlighted by Reviewer 2’s comment (e.g., Buolamwini 2017 and Nagpal et al. 2019), we show that OpenCV exhibits significantly higher levels of classification confidence when labeling faces as “male” as compared to “female,” exposing gender-related biases in its performance (Fig. S9). Given these biases, we

prefer to rely on human annotator judgments for our main results. Nevertheless, we present the results based on automated classifications in our *SOM* because they allow us to test whether our main results hold in the absence of subjective human judgments. Indeed, figure S10 shows that men are significantly over-represented in our dataset according to OpenCV's classifications of both our cropped and uncropped datasets, and importantly, figure S11 further shows that these results equally hold when controlling for OpenCV's confidence in its classifications. The gender distribution according to OpenCV's classifications is highly stable regardless of whether we examine the cropped or uncropped images.

(3) Validation of OpenCV (section A.2.16, *SOM*). We conduct a series of analyses to investigate the reliability and quality of OpenCV's facial cropping algorithm. We show that its false negative rate - i.e., the rate at which it fails to successfully identify a face -- exhibits no significant gender-related biases in our dataset.

(4) Robustness to ground truth datasets for which the true underlying gender of the faces depicted is known (section A.2.2 *SOM*). In response to comment 5 from Reviewer 1, we show that our main results hold when analyzing a canonical dataset of over 500k images of celebrities depicted over IMDb and Wikipedia, for which each image is associated with the self-identified gender of each celebrity according to their public IMDb and Wikipedia biographical page (Fig. S4). Since this analysis does not rely on cropping but rather ground truth gender labels, it shows that our findings generalize to settings that lack the use of a cropping algorithm.

Memo references:

Buolamwini, Joy A. *Gender shades: intersectional phenotypic and demographic evaluation of face datasets and gender classifiers*. PhD thesis, Massachusetts Institute of Technology, 2017.

Nagpal, Shruti, Maneet Singh, Richa Singh, and Mayank Vatsa. Deep learning for face recognition: Pride or prejudiced? *arXiv preprint arXiv:1904.01219*, 2019.

7. *"Specifically, the female centroid consists of the words woman, her, she, female, and girl, whereas the male centroid consists of the opposite words man, his, he, male, and boy." Using the embeddings of 4 words might not be sufficient to obtain a statistically robust representation of a concept.*

We thank Reviewer 1 for recommending another valuable robustness test. Our main analysis originally replicated the Kozlowksi et al.'s (2018, ref. below) method for defining gendered centroids using five gendered word pairs - that is, *woman, her, she, female,* and *girl* for the female centroid, and *man, him, he, male,* and *boy* for the male centroid. All of our main results equally hold both with (i) using fewer gendered terms to define the gender centroids, and (ii) when using many more gendered terms to define the centroids. The new supplementary analyses are provided for your review, below (starting on pg. 39 of the *SOM*).

"Here, we test the robustness of our main results to different methods for defining the female and male centroids used to construct the gender dimension in embedding space. Our main analyses replicate the Kozlowksi et al. (2018) [5]

method for defining gendered centroids using five gendered word pairs - that is, *woman*, *her*, *she*, *female*, and *girl* for the female centroid, and *man*, *him*, *he*, *male*, and *boy* for the male centroid. Here, we show that all of our main results equally hold while (i) using fewer gendered terms to define the gender centroids, and (ii) when using up to three times more gendered terms to define the centroids. Specifically, we compare the results from our main method (version 1) to three alternative versions of the gendered centroids:

Version 2, where the female (male) centroid is defined only by *woman* (*man*), *female* (*male*), and *girl* (*boy*);

Version 3, where the female (male) centroid is defined only by *woman* (*man*), *her* (*him*), *she* (*he*), *female* (*male*), *girl* (*boy*), *feminine* (*masculine*), and *womanly* (*manly*); and

Version 4, where the female (male) centroid is defined only by *woman* (*man*), *her* (*him*), *she* (*he*), *female* (*male*), *girl* (*boy*), *feminine* (*masculine*), *womanly* (*manly*); *mother* (*father*), *sister* (*brother*), *daughter* (*son*), *grandmother* (*grandfather*), *granddaughter* (*grandson*), *aunt* (*uncle*), *girlfriend* (*boyfriend*), and *wife* (*husband*).

Figure S16: The gender associations for all social categories in WordNet (N=3,434) according to textual word2vec embeddings of Google News, under different representations of the gender dimension. Correlating the gender associations between the main gender dimension in our study and (A) version 2, (B) version 3, and (C) version 4 of the gender dimension. Error bars display 95% confidence intervals.

Figure S16 examines the correlation of the gender associations between the representation of the textual gender dimension used in our main study and three alternative versions of the textual gender dimension. All panels of Figure S16 illustrate that, across all 3,434 social categories, the gender association of categories is highly correlated across these alternative gender dimensions ($p < .0001$ for all comparisons), indicating that reasonable changes in the gender dimension have minimal impact on the gender associations of categories.

Figure S17: The distribution of gender associations for 2,986 social categories in images from Google Images and texts from Google News, while varying the construction of the gender dimension used to score the gender association of categories in the word embedding model of Google News. The image-based measure captures the frequency of male and female faces associated with each category in Google Image search results (-1 means 100% female; 1 means 100% male); the text-based measures capture the frequency at which each category is associated with male or female terms in the Google News corpus of over 100 billion words (-1 means 100% female; 1 means 100% male associations).

Figure S17 goes further by confirming that our main results comparing gender associations across images and text equally hold under alternative representations of the gender dimension. Men are significantly over-represented in Google Images compared to Google News, regardless of how the gender dimension is constructed ($p < .0001$ for all comparisons, Wilcoxon Signed-Rank Test); and the overall magnitude of the gender associations, in either the male or female direction, are similarly stronger in Google Images as compared to Google News, regardless of how the gender dimension is constructed ($p < .0001$ for all comparisons, Wilcoxon Signed-Rank Test).”

8. It is not clear why the dimensionality of the word embeddings is evaluated in this context. Evaluating context window size in training word embeddings on corpora collected in 2019 would be a justifiable machine learning setting. The context window size determines how much syntactic vs. semantic information is being captured.

We examine the dimensionality of embedding models to control for the level of semantic nuance captured in the embeddings, which helps to rule out the concern that textual models exhibit weaker gender bias because they detect significantly more semantic nuance in the meaning of categories. To the contrary, we show that the dimensionality of word embedding models hold no relation to the strength of its gender associations as detected by our method. We appreciate Reviewer 1’s recommendations for how to further improve our machine learning setting. We take two approaches to implementing these recommendations:

(1) Robustness to comparing against a range of popular word embedding models. First, as specified in response to comment 2 by Reviewer 1, our results hold when comparing against a wide array of word embedding methods (Fig. S5), which vary in terms of window size, data source, time of data collection (ranging from 2013 to 2020), and their algorithmic techniques for generating word vectors. This approach to demonstrating robustness was preferred given the resource constraints of collecting and training a new word embedding model of Google news records from 2019. Training new word embedding models at the appropriate scale for our analysis takes an enormous amount of computing power, which is why social scientists and much of the machine learning community standardly harness pre-trained models released by companies such as Google, Facebook, and OpenAI, who can afford to spend millions of dollars to train these models; and even with their resources, training these models can take weeks to converge and evaluate, often leading to months of refinement by teams of engineers. Indeed, it has become a popular talking point in the AI community that the percentage of large-scale AI results from academia has plummeted in recent years. Consider the following results from recent studies evaluating the role of academia in contemporary AI research.

Memo Fig. 2. The fraction of large-scale AI results produced by academia between the 1960s to today; data and figure are from Benaich & Hogarth (2022; link here).

Memo Fig. 3. The percent of the 10 biggest AI models that are from industry, from 2002 to 2021; data and figure are from Ahmed et al. (2023, ref. below).

Benaich & Hogarth (2022) argue that the fraction of large-scale AI results produced from academia has sharply declined from 80% in the 1960s and even 60% in 2010 to now 0% in 2020 (Memo Figure 2, copied from their review). This is echoed by a recent paper in *Science* entitled “The Growing Influence of Industry in AI Research” (Ahmed et al. 2023), which shows that almost 100% of the top 10 AI models in recent years have come from industry (Memo Figure 3). We share these references not as definitive evidence of the empirical distribution of large-scale AI results across industry and academia, but rather as a clear indication of the extent to which the AI community has documented the inability for academics to scale AI projects to keep pace with the financial and computing resources in industry.

Thankfully, by comparing against the full breadth of cutting-edge pre-trained embedding models, we can show that our results are highly robust to varying the parameters highlighted by Reviewer 1. What social scientists offer the study of AI is critical theoretical insights and analyses that advance our sociological and psychological understanding of biases in the models and datasets released by companies for public consumption and use in research communities. We believe our contributions in this direction warrant publication.

Memo Reference

Ahmed, Nur, Muntasir Wahed, and Neil C. Thompson. “The Growing Influence of Industry in AI Research.” *Science* 379, no. 6635 (March 3, 2023): 884–86.

(2) Robustness to an alternative method for comparing images and texts from 2022 that does not rely on word embedding methods. In our revised experiment (see reply to comment 9 from Reviewer 1 for elaboration), participants were asked to gather either images or textual descriptions of each occupation from Google and then to upload these descriptions via our online survey. We hired a team of annotators to code whether each image or textual description uploaded by participants was female (1), neutral (non-gendered) (0), or male (1). Each textual description was coded as either male, female, or neutral based on whether it used male or female pronouns or names to describe the occupation (e.g., a description that referred to a doctor as “he” would be coded as male). Textual descriptions were identified as neutral if they did not ascribe a particular gender to the occupation described. Similarly, the gender of each image was identified by classifying the gender of the focal face in each image; faces were coded as neutral if they did not contain a face or if the face presented was indecipherable. Then for each occupation in each condition, we calculate the average gender association across the descriptions provided by separate participants, such that the measure of gender association is identical for the image and text condition. Since the number of descriptions per occupation in each condition is constant (via randomization), and the outcome measure of gender association is similarly constant (such that each description is associated with a single gender label), this provides an apples-to-apples comparison of the strength of gender associations between the textual and image modalities.

Another key strength of this approach is that the image and textual descriptions being compared were encountered via Google and uploaded by participants at the exact same time, in May 2022 (pre-registration: <https://osf.io/3jhzx>). For this reason, temporal differences in when these images and texts were downloaded cannot account for the differences observed in their gender associations. This robustness test also shows that the differences observed in the gender associations between images and text are not driven by technical differences in how gender associations are represented in word embedding models as compared to our simple count-based approach to detecting such associations in images. This alternative method uses a simple count-based approach, whereby annotators were required to code both image and textual descriptions in terms of a simple binary representation of gender at the description-level.

Crucially, using this coarse-grained approach, we continue to find clear statistical evidence that gender associations are significantly stronger in images than text, in terms of the frequency with which each modality presents overtly gendered descriptions of occupations in Google search results. Reassuringly, the gender associations detected by this simpler, count-based method are highly correlated with the gender associations detected by our word embedding method and our observational sample of online images, suggesting that both methods capture fundamental differences in the gender distributions captured by images and text (see figure S30, copied below). This is particularly impressive since the word embedding model of Google News that we compare against was trained on 2013 data, whereas these experimental descriptions were downloaded in 2022.

Here, we copy this revised analysis in the main text, along with a supplementary section which shows that the measures of gender associations provided by this alternative method highly correlate with our original measures.

Revisions to pg. 8 in the main text:

“Consistent with our observational results, panel A of Figure 3 shows that the descriptions participants uploaded were significantly more gendered in the image condition than the text condition ($p < .0001$, Wilcoxon Signed-rank Test, Two-sided; the gender associations for each occupation are highly correlated with those detected for the same occupations in our main observational analyses of Google Images and Google News, Fig. S30). Panel B of Figure 3 shows that exposure to more gendered stimuli in the image condition led participants to report significantly stronger gender associations as compared to participants in both the text ($p < .001$) and the control ($p < .001$) condition, while there was no significant difference between those in the text and control condition ($p = .56$) (Wilcoxon Signed-rank Test, Two-sided). These results hold when controlling for the number of online sources that participants encountered, and the amount of time they spent evaluating online descriptions (Fig. S31; Table S9).”

Figure 3: Differential effects of using Google to search for images rather than textual descriptions of occupations on participants' explicit and implicit gender biases, as compared to a control condition (N=450). The green (purple) vertical lines indicate average effects for the text (image) condition. (A) The average absolute strength of the gender associations in participants' uploads for each category in both the text and image condition. (B) The average absolute strength of the gender associations that participants reported for each occupation in each condition. (C) The average absolute strength of the implicit bias (dscore) that participants exhibited in each condition. (D) The correlation between the average strength of participants' self-reported gender associations and their implicit bias toward associating women with liberal arts and men with science. (E) The correlation between the gender associations of the descriptions that participants uploaded and the gender associations they reported for each occupation, colored by condition. (F) The correlation between the average strength of the gender associations of the descriptions that participants uploaded and average strength of the gender associations they reported for each occupation, colored by condition. *** , $p < .001$. All error bands and error bars display 95% confidence intervals.

Corresponding revisions in the SOM (starting on pg. 67):

Figure S30: The relationship between the distribution of gender in our observational sample of the top 100 Google images per occupation and (A) the distribution of gender associations in the images from Google uploaded by participants, and (B) participants' self-reported gender associations for each occupation, which they provided after uploading an image from Google. The relationship between the distribution of gender in our observational word embedding measures of Google News and (C) the distribution of gender associations in the textual descriptions from Google News uploaded by participants, and (D) participants' self-reported gender associations for each occupation, which they provided after uploading a textual description from Google News. All results are averaged at the occupation level, such that 58 data points (occupations) are shown. Error bands show 95% confidence intervals.

“A.4.1 Using our Observational Measures of Gender Bias to Predict Gender Bias in Participants' Uploaded Descriptions and Self-report Beliefs.

Here we confirm that our observational measures of gender bias in images from Google Images and texts from Google News are predictive of gender bias in the image/textual descriptions that participants uploaded in our experiment, as well as

in their self-reported gender associations with each occupation. This analysis is important not only for demonstrating the ability for our observational measures to predict ecologically valid patterns of content exposure and response bias among human participants using google, but also to confirm the correspondence between our alternative methods of measuring gender bias in text, namely the word embedding technique and the coarser method of counting the frequency of gendered descriptions in our experimental data.

The results of this analysis are presented in Figure S30. Panel A of Figure S30 shows that the distribution of gender in our observational sample of the top 100 Google images per occupation is highly correlated with the distribution of gender in the images uploaded by participants in our experiment ($r = 0.58$, $p < .00001$, $N = 58$ occupations). Similarly, panel B of Figure S30 shows that the distribution of gender in our observational sample of the top 100 Google images per occupation is highly correlated with participants' self-reported gender associations for each occupation, which they provided after uploading an image from Google ($r = 0.76$, $p < .00001$, $N = 58$ occupations). The same results hold for the text condition. Panel C of Figure S30 shows that the distribution of gender in our observational word embedding measures of Google News is highly correlated with the distribution of gender in the textual descriptions uploaded by participants in our experiment ($r = 0.54$, $p < .00001$, $N = 58$ occupations). This finding is particularly striking not only because gender bias is measured differently by our word embedding model and experimental analysis, but also because the Google News data underlying the word embedding model was collected in 2013, whereas our experimental data involves descriptions from Google News downloaded in 2022. Lastly, panel D of Figure S30 shows that the distribution of gender in our observational word embedding measures of Google News is highly correlated with participants' self-reported gender associations for each occupation, which they provided after uploading a textual description from Google News ($r = 0.79$, $p < .00001$, $N = 58$ occupations)."

9. The implicit association tests do not have controlled settings. The magnitude of bias in textual descriptions is not measured. Accordingly, quantifying its influence on subjects' implicit associations does not provide a comprehensive analysis. Moreover, it is unclear if stereotype threat or exposure to a study of gender bias in data is causing amplified implicit gender bias scores. The experimental settings need to account for all possible hypotheses to justify the findings.

We agree that greater clarity is needed in the presentation of our experimental results. We now quantify the magnitude of bias in textual descriptions as well as image descriptions, and we show (in Figure 3A) not only that bias is significantly stronger in image than text descriptions, but also that this difference is correlated with higher levels of explicit (Figure 3B) and implicit bias (Figure 3C) in the image as compared to the text condition. These results are situated within an improved replication of our experimental design that now includes a control condition, along with a modified version of the text condition which requires participants to search for occupational descriptions using the Google News search engine specifically. This updated design

-- and our revised presentation of these experimental results -- helps address this important comment from Reviewer 1. We provide the corresponding revisions to our manuscript and *SOM* below.

In general, we agree with Reviewer 1 that our experimental results may be compatible with a number of more specific mechanisms concerning the nuances of subjects' psychological processes – such as stereotype threat – but for our purposes, the theoretical resolution at which our theory is defined and tested is sufficient for supporting our core argument, namely that gender biases are more prevalent in online images than online texts, and this stronger bias has significant impacts on the explicit and implicit biases of internet users. We believe that this marks a significant contribution that paves the way toward future work which can explore the myriad of psychological processes that produce the stable, higher-level outcome with which the present study is concerned. We provide the corresponding revisions to the manuscript and appendix below. This includes the supplementary section revealing the robustness of our results in the text condition to having participants use google.com or news.google.com.

Main experimental results (starting on pg. 8 of the manuscript):

“We begin by examining the extent of gender bias in the descriptions uploaded by participants. A team of annotators labeled each textual description as either female, male, or neutral based on whether it used female or male pronouns or names to describe the occupation (e.g., a description referring to a “doctor” as “he” would be coded as “male”); textual descriptions were identified as neutral if they did not ascribe a particular gender to the occupation (*SOM*). Similarly, a team of annotators labeled the gender of the focal face in each uploaded image as either female, male, or neutral; faces were coded as neutral if they contained no face or an indecipherable face. Then, for each occupation, we calculated the gender balance of the descriptions provided by participants by computing the average gender association across all descriptions. A key feature of this analysis is that it provides another method for comparing gender associations across images and texts that does not rely on word embedding models, while also ensuring that the images and texts being compared were collected by users during the same time period.

Fig. 3. Differential effects of googling for visual rather than textual descriptions of occupations on participants' explicit and implicit gender biases, as compared to a neutral control condition ($N=450$). (A) The average absolute strength of the gender associations that participants uploaded for each category in both the text and image condition. (B) The average absolute strength of the gender associations that participants reported for each occupation in each condition. (C) The average absolute strength of the implicit bias (dscore) that participants exhibited in each condition. (D) The correlation between the average strength of participants' self-reported gender associations and their implicit bias toward associating men with science and women with liberal arts. (E) The correlation between the gender associations of the descriptions that participants uploaded and the gender associations they reported for each occupation, split by condition. M., Male; F., Female. (F) The correlation between the average strength of the gender associations of the descriptions that participants uploaded and average strength of the gender associations they reported for each occupation, split by condition. All error bands and error bars display 95% confidence intervals.

Consistent with our observational results, panel A of Figure 3 shows that the descriptions participants uploaded were significantly more gendered in the image condition than the text condition ($p < .0001$, Wilcoxon Signed-rank Test, Two-sided; the gender associations for each occupation are highly correlated with those detected for the same occupations in our main observational analyses of Google Images and Google News, Fig. S30). Panel B of Figure 3 shows that exposure to more gendered stimuli in the image condition led participants to report significantly

stronger gender associations as compared to participants in both the text ($p < .001$) and the control ($p < .001$) condition, while there was no significant difference between those in the text and control condition ($p = .56$) (Wilcoxon Signed-rank Test, Two-sided). These results hold when controlling for the number of online sources that participants encountered, and the amount of time they spent evaluating online descriptions (Fig. S31; Table S9).

Concerningly, Panel C of Figure 3 shows that exposure to more gendered descriptions in the image condition also led participants to exhibit significantly stronger implicit bias toward associating women with arts and men with science. There was no significant difference between participants' implicit bias in the text and control condition ($p = .19$), yet participants in the image condition exhibited significantly stronger implicit bias than those in the control condition ($p < .001$) (Wilcoxon Signed-rank Test, Two-sided). Indeed, panel D of Figure 3 identifies a robust correlation between the strength of participants' self-reported gender associations and the strength of their implicit bias, both of which are amplified in the image condition ($p < .0001$, Jonckheere-Terpsta Test). Together, these findings suggest that exposure to more gendered descriptions in the image condition more strongly primed participants' explicit and implicit gender associations.

This priming mechanism is further supported by panel E of Figure 3, which shows a strong correlation between the gender associations in the descriptions that participants uploaded and the gender associations in their own explicit responses across occupations ($\rho = .8$, $p < .0001$); as well as by panel F of Figure 3, which shows a strong correlation between the overall strength of gender associations in the descriptions that participants uploaded and the overall strength of the average gender associations they reported across occupations ($\rho = .57$, $p < .0001$). These results hold for both the image and text condition, and across occupations (Table S10).

Importantly, we find that images differ from text not only in the prevalence of gender bias they contain, but also in their ability to prime gender bias in people's beliefs. Participants who uploaded gendered images reported significantly stronger gender bias compared to participants who uploaded gendered textual descriptions of the same occupations ($p < .0001$, Student T-test, Two-tailed; Fig. S32). Thus, even when gender is salient in both text and image, exposure to images leads to stronger bias in people's beliefs. Most strikingly, only participants in the image condition exhibited significantly stronger implicit bias three days after the experiment (Table S11), indicating that these outcomes go beyond ephemeral priming effects and may contribute to enduring biases."

Supplementary analysis comparing alternative versions of the text condition (starting on pg. 60 of the SOM):

"A.3.3 Robustness to Participants Searching in Google Instead of Google News

In our main experiment, participants in the text condition were guided to the Google News search bar and were asked to identify and upload a description of an occupation from this context. This design choice was made to maximize the similarity of the experimental context to our observational comparisons between Google Images and Google News. However, participants in our experiment (N=150) were also randomized to a different version of the text condition, in which they were guided to the general Google search engine (i.e., simply google.com) and asked to perform the same task. Thus, in total, our experiment included 600 unique participants.

Figure S25: Differential effects of googling for images rather than textual descriptions of occupations on participants' explicit and implicit gender biases, as compared to a neutral control condition, while including a version of the text condition in which participants were directed to search for textual descriptions in the general Google search engine, rather than in Google News specifically (N = 600); the data shown for all other conditions are the same as those presented in our main results (see Fig. 3 in the main text). (A) The average absolute strength of the gender associations that participants reported for each occupation in each condition. (B) The average absolute strength of the implicit bias (dscore) that participants exhibited in each condition. The results are shown in solid purple (image condition), solid green (Google News text condition), dotted black (control condition), and dotted green (General Google text condition).

Building on the presentation of our main experimental results, Figure S25 compares the outcomes of each condition to an alternative version of the text condition in which participants were directed to retrieve textual descriptions of each occupation by searching via the general Google search engine (referred to as the “text neutral” condition) rather than Google News specifically. Panel A of Figure S25 shows that there was no significant difference between the neutral text condition and the Google News text and control condition in terms of the strength of participants' self-reported gender associations for each occupation. Yet, again, we find that participants in the image condition produced significantly stronger gender associations compared to participants in the neutral text condition ($p < .001$,

Wilcoxon Signed Rank Test, Two-tailed), along with all other conditions (as discussed in the main text). Panel B of Figure S25 shows that this result replicates when analyzing the overall strength of participants' implicit bias toward associated men with science and women with liberal arts; that is, we find no significant difference between participants' implicit bias in the text neutral and control condition, nor between the text neutral condition and main text condition; yet, participants in the image condition exhibited significantly stronger implicit biases than those in the text neutral condition ($p < .001$, Wilcoxon Rank Sum Test, Two-tailed). These results show that our main findings are not an artifact of comparing against Google News search results in particular, but instead generalize to a different means of acquiring textual descriptions of social categories via the Google search engine.”

Supplementary analysis demonstrating the stronger priming effect of images compared to textual descriptions:

Figure S32: The average absolute strength of the gender associations that participants reported for each occupation in each condition, while excluding this analysis to only those descriptions that were explicitly gendered as either male or female in the text and image condition. The green (purple) vertical lines indicate average effects for the text (image) condition. $N=2,570$ image descriptions; $N=673$ text descriptions.

“A.4.4 Comparing the Strength of Gender Priming between Text and Images

In our main text, we focus on comparing the image and text condition in terms of the overall prevalence of gender biases in the descriptions that participants uploaded and the associated effects this has on the prevalence of gender biases in participants' explicit and implicit beliefs. Here, we evaluate the additional prediction that images are stronger at priming gender biases in people's beliefs, even when the images and texts being compared are explicitly gendered. Figure S32 shows that when comparing only participants who provided explicitly gendered descriptions in the image and text condition, participants in the image condition reported significantly stronger gender biases in their associations with

occupations ($p < .0001$, Student T-test), even when using a linear regression to control for the specific gender and the specific occupation associated with the uploaded description ($\beta = .06$, $SE = .01$, $p < .0001$). These findings indicate that even when gender is salient in both text and image, exposure to images leads to stronger biases in people's beliefs."

10. Please note the exact type of implicit bias in the labels of charts in figure 3.

Panel C and D of our revised figure 3 (copied in response to comment 9, above) now explicitly indicate that our measure of implicit bias corresponds to the IAT dscore; the meaning of this measure is further elaborated in the caption to the figure. These revisions are provided on page 9 of the manuscript.

11. Significant references that show almost identical work are missing. Two of them are noted above. There is vast literature **on** gender bias in information retrieval, computer vision, socio-technical systems, and human-computer interaction.

Thank you for this suggestion. We agree that there is a large body of work on gender bias across fields and subdisciplines. In keeping with the succinct style of *Nature* papers, our revised manuscript references the most relevant papers that are closest to our study -- namely, Kay et al. 2015, Metaxa et al. 2021, and Vlasceanu & Amodio 2022 -- and in addition, we also reference two books that summarize this enormous literature, in particular Noble's *Algorithms of Oppression* and Christian's *The Alignment Problem*. We further include a number of related references in our manuscript and supplementary material that reflect a broader coverage of studies in this space. We have copied the newly added citations below. We are happy to add any additional references recommended by Reviewer 1.

- Jason J Jones, Mohammad Ruhul Amin, Jessica Kim, and Steven Skiena. Stereotypical gender associations in language have decreased over time. *Sociological Science*, 7:1–35, 2020.
- Aylin Caliskan, Joanna J Bryson, and Arvind Narayanan. Semantics derived automatically from language corpora contain human-like biases. *Science*, 356(6334):183–186, 2017.
- Madalina Vlasceanu and David M Amodio. Propagation of societal gender inequality by internet search algorithms. *PNAS*, 119(29):e2204529119, 2022.
- Joy Adowaa Buolamwini. *Gender shades: intersectional phenotypic and demographic evaluation of face datasets and gender classifiers*. PhD thesis, MIT, 2017.
- Shruti Nagpal, Maneet Singh, Richa Singh, and Mayank Vatsa. Deep learning for face recognition: Pride or prejudiced? *arXiv preprint arXiv:1904.01219*, 2019.
- Abeba Birhane, et al. Auditing saliency cropping algorithms. In *Proceedings of the IEEE/CVF Winter Conference on Applications of Computer Vision*, pages 4051– 4059, 2022.
- Christian, Brian. *The Alignment Problem: Machine Learning and Human Values*. 1st edition. New York, NY: W. W. Norton & Company, 2020.

12. Precision in language can be improved. "Here, we develop an auditing framework for comparing gender stereotypes across online images and texts for thousands of social categories, which we apply to Google and Wikipedia." What does applying to Google and Wikipedia mean?

We thank Reviewer 1 for this helpful suggestion for how to improve our language. We have decided to remove this sentence. In its place, we now write (pg. 1):

“Here, we address this gap by developing computational techniques for comparing gender bias in the representation of social categories across images and texts, which we apply to (i) a novel dataset comprising over one million images from Google, Wikipedia, and IMDb, mapped to 3,495 social categories, as well as to (ii) textual representations of these categories, using word embedding models trained on billions of words from these platforms.”

Referee #2 (Remarks to the Author):

1. This seems like an intriguing paper on the gender gap in online images. Given that we all increasingly rely on online search engines, existing biases might be reproduced if those search engines are structurally biased in some way – i.e., with respect to one or more social categories. What the authors convincingly show is that Google images with regard to a set of pre-defined social categories (e.g., athlete, doctor, and so forth) are highly gendered; that is, women seem to be disproportionately absent in some Google images (e.g., doctors) or highly overrepresented (e.g., nurse) when querying those social categories. The authors compare this with the gender gap with respect to those same social categories in large text corpora and find that the gender gap is stronger in images. They also report on an interesting experiment where they argue that this gendering in pictures reinforces gender stereotypes. I congratulate the authors on this relevant study that is well-carried out and that would likely be very appealing to a broad readership.

Thank you!

In what follows is a list of concerns I have with this paper that led me to not (yet) fully trust the results nor its associated claims.

2. Much of the paper leans on the claim that the gender gap is stronger in images than in text. My most pressing concern is whether or not the authors' claim may rest on two different types of media that are inherently incomparable from the outset. I read and appreciate the supplemental materials that describe well the harmonization between the two metrics for how gendered given social categories are. Yet, does the fact that the textual metrics' range initially is far less “absolute” than the pictures not show that very different data-generating processes are underlying these data sources? Writing about certain social categories might occur in a relatively less biased way by design, whereas the coders are actually pushed into identifying certain genders in the images. The high-dimensional word2vec might capture far more of those nuances compared to coding up the binary genders. So the coding of the images compared to measuring the genderedness of the social categories in text might be incomparable and/or by design differ. That begs the question: how surprising are these results given the media under consideration and given the approaches of the authors? I suggest the authors at the very minimum answer these questions conceptually as I worry that otherwise the paper

may overclaim how surprising this finding is. A more fascinating approach would be to actually tackle these questions empirically or theoretically: can the authors devise a way that such design effects are accounted for? Or can they simply theorize and conjecture about the difference a priori?

These are very important points, which have led us to significantly improve our measurements and findings. We focus on addressing Reviewer 2's core concern that *"writing about certain social categories might occur in a relatively less biased way by design, whereas the coders are actually pushed into identifying certain genders in the images. The high-dimensional word2vec might capture far more of those nuances compared to coding up the binary genders."*

To address this concern, we develop an alternative method that pushes coders to identify the gender indicated by images and textual descriptions of social categories using the same counting procedure and coding scheme. This approach overcomes design differences in the data-generating process underlying the measurement of gender associations in images and word embedding models. We apply this method to the descriptions participants uploaded in our experiment, as we describe below.

In our revised experiment (see reply to comment 4 from Reviewer 2 for direct discussion of this improved design), participants were asked to upload either images or textual descriptions of each occupation (in this revised experiment, participants searched for textual descriptions via the news.google.com engine). We hired a team of annotators to code whether each image or textual description uploaded by participants was female (-1), neutral (0), or male (1). Each textual description was coded as either male, female, or neutral based on whether it used male or female pronouns or names to describe the occupation (e.g., a description referring to a "doctor" as "he" would be coded as "male"); textual descriptions were identified as neutral if they did not ascribe a particular gender to the occupation described. Similarly, the gender of each image description was identified by coding the gender of the focal face in each image; faces were coded as neutral if they did not contain a face or the face was indecipherable. Then for each occupation in each experimental condition, we calculated the average gender association across the descriptions provided by separate participants.

This coarse-grained method levels the playing field between images and text. Via this method, descriptions of occupations in either images or texts are equally compressed into a simple binary coding of gender at the description level. Since the number of descriptions per occupation is constant across the image and text condition, and the binary outcome classification of gender association is also constant across conditions, this provides a direct apples-to-apples comparison of the strength of gender associations between images and text. In other words, the data-generating process between the text and image modality is standardized: participants are asked to use Google to extract single, bounded descriptions of each occupation (in either textual or image form) and, despite the distinct patterns of nuance captured by each modality

respectively, each description is collapsed to either a -1 (female), 0 (neutral), or 1 (male) classification by annotators.

Using this coarse-grained approach, we continue to find clear statistical evidence that gender associations are significantly stronger in images than text, in terms of the frequency with which each modality presents overtly gendered descriptions of occupations in Google search results (note, our findings are identical regardless of whether participants searched for textual descriptions in Google News specifically or the generic Google search bar, fig. S25). Importantly, the gender associations detected for each occupation by this simpler count-based method are highly correlated with the gender associations detected for the same occupations by our word embedding method and our observational sample of online images, suggesting that both methods capture fundamental differences in the gender distributions reflected by images and text (see figure S30, copied below). Impressively, this is despite the fact that the word embedding model of Google News that we compare against was trained on 2013 text data, whereas these experimental descriptions were downloaded and annotated in 2022.

Here, we copy this revised analysis in the main text, along with a supplementary section which shows that the measures of gender associations provided by this alternative method highly correlate with our original observational measures.

Revisions to pg. 8 in the main text:

“Consistent with our observational results, panel A of Figure 3 shows that the descriptions participants uploaded were significantly more gendered in the image condition than the text condition ($p < .0001$, Wilcoxon Signed-rank Test, Two-sided; the gender associations for each occupation are highly correlated with those detected for the same occupations in our main observational analyses of Google Images and Google News, Fig. S30). Panel B of Figure 3 shows that exposure to more gendered stimuli in the image condition led participants to report significantly stronger gender associations as compared to participants in both the text ($p < .001$) and the control ($p < .001$) condition, while there was no significant difference between those in the text and control condition ($p = .56$) (Wilcoxon Signed-rank Test, Two-sided). These results hold when controlling for the number of online sources that participants encountered, and the amount of time they spent evaluating online descriptions (Fig. S31; Table S9).”

Figure 3: Differential effects of using Google to search for images rather than textual descriptions of occupations on participants' explicit and implicit gender biases, as compared to a control condition (N=450). The green (purple) vertical lines indicate average effects for the text (image) condition. (A) The average absolute strength of the gender associations in participants' uploads for each category in both the text and image condition. (B) The average absolute strength of the gender associations that participants reported for each occupation in each condition. (C) The average absolute strength of the implicit bias (dscore) that participants exhibited in each condition. (D) The correlation between the average strength of participants' self-reported gender associations and their implicit bias toward associating women with liberal arts and men with science. (E) The correlation between the gender associations of the descriptions that participants uploaded and the gender associations they reported for each occupation, colored by condition. (F) The correlation between the average strength of the gender associations of the descriptions that participants uploaded and average strength of the gender associations they reported for each occupation, colored by condition. ***, $p < .001$. All error bands and error bars display 95% confidence intervals.

“A.4.1 Using our Observational Measures of Gender Bias to Predict Gender Bias in Participants' Uploaded Descriptions and Self-report Beliefs.

Here we confirm that our observational measures of gender bias in images from Google Images and texts from Google News are predictive of gender bias in the image/textual descriptions that participants uploaded in our experiment, as well as

in their self-reported gender associations with each occupation. This analysis is important not only for demonstrating the ability for our observational measures to predict ecologically valid patterns of content exposure and response bias among human participants using google, but also to confirm the correspondence between our alternative methods of measuring gender bias in text, namely the word embedding technique and the coarser method of counting the frequency of gendered descriptions in our experimental data.

Figure S30: The relationship between the distribution of gender in our observational sample of the top 100 Google images per occupation and (A) the distribution of gender associations in the images from Google uploaded by participants, and (B) participants' self-reported gender associations for each occupation, which they provided after uploading an image from Google. The relationship between the distribution of gender in our observational word embedding measures of Google News and (C) the distribution of gender associations in the textual descriptions from Google News uploaded by participants, and (D) participants' self-reported gender associations for each occupation, which they provided after uploading a textual description from Google News. All results are averaged at the occupation level, such that 58 data points (occupations) are shown. Error bands show 95% confidence intervals.

The results of this analysis are presented in Figure S30. Panel A of Figure S30 shows that the distribution of gender in our observational sample of the top 100 Google images per occupation is highly correlated with the distribution of gender in the images uploaded by participants in our experiment ($r = 0.58$, $p < .00001$, $N=58$ occupations). Similarly, panel B of Figure S30 shows that the distribution of gender in our observational sample of the top 100 Google images per occupation is highly correlated with participants' self-reported gender associations for each occupation, which they provided after uploading an image from Google ($r = 0.76$, $p < .00001$, $N=58$ occupations). The same results hold for the text condition. Panel C of Figure S30 shows that the distribution of gender in our observational word embedding measures of Google News is highly correlated with the distribution of gender in the textual descriptions uploaded by participants in our experiment ($r = 0.54$, $p < .00001$, $N=58$ occupations). This finding is particularly striking not only because gender bias is measured differently by our word embedding model and experimental analysis, but also because the Google News data underlying the word embedding model was collected in 2013, whereas our experimental data involves descriptions from Google News downloaded in 2022. Lastly, panel D of Figure S30 shows that the distribution of gender in our observational word embedding measures of Google News is highly correlated with participants' self-reported gender associations for each occupation, which they provided after uploading a textual description from Google News ($r = 0.79$, $p < .00001$, $N=58$ occupations)."

3. What are the mechanisms underlying the gendered images on Google? I appreciate the approach of the authors that accounts for browsing histories, but without knowing anything about how and why these pictures come to be posted on websites and are then shown on Google it remains quite difficult to comprehend what is going on in the paper's data. Is the same algorithm underlying Google images compared to regular Google queries? So the overarching question here is: does the bias in the Google images underlie some real world data-generating process. The paper claims that it is, but it remains ambiguous what that is. I suggest the authors do more conceptual and theoretical work on these questions so as to support their claims.

These are very important concerns, which we address as two separate points:

(1) ...without knowing anything about how and why these pictures come to be posted on websites and are then shown on Google it remains quite difficult to comprehend what is going on in the paper's data.

The question about "how and why these pictures come to be posted on websites" is indeed very interesting, and we have worked to provide data-driven insights into this challenging question (described below). But before we provide these insights, we would like to first clarify that our primary interest is in the gender bias conveyed by the content that people encounter online via Google search and Wikipedia, regardless of the underlying processes leading to the curation and distribution of this content (which

differ across these platforms). That is, we are interested in how differences in the prevalence of gender bias in online images and texts from these platforms can have differential effects on priming gender bias in the beliefs of people using these platforms. We believe these questions can be explored by examining the distribution of gender bias in the images and texts that regular internet users encounter via these platforms through normal usage behavior, which often operates independently of any understanding of how the underlying algorithms shape the curation and ranking of this content.

Nevertheless, we agree that this question is fascinating, and we have collected additional data to provide preliminary insights into the sources of the images returned by Google search, which we believe suffices to demonstrate that our images do reflect a myriad of real-world data-generating processes that are consistent with our overarching claims.

To do this, we leveraged data from our experiment, which required subjects to document the website from which they downloaded each description after they initially encountered this image via Google Images. We hired human annotators to classify the type of website from which participants downloaded each image (see section A.4.6 in the *SOM*, copied below).

“A.4.6 Examining the Sources of Online Images

In this supplementary analysis, we examine the online sources from which the images in our sample primarily derive, since the Google Image search engine serves mainly as an intermediary for routing internet users to websites containing particular images. For this purpose, we leveraged data from our main experiment, which required participants to report the website that they downloaded each image from after they initially searched for this image on Google. A trained team of five undergraduates then classified the type of website from which participants downloaded each image; all final classifications used were the result of consensus reached among these annotators. These online sources are classified into nine classes: blogs, entertainment (e.g., IMDb), Governmental, Social Media (e.g., Facebook and Twitter), News, Stock Photos, Business, Educational (e.g., Wikipedia), and unknown.

Figure S33: The distribution of online sources of the images that participants downloaded and uploaded as part of our main experiment.

Figure S33 presents the distribution of online sources for the images that participants downloaded and uploaded in our main experiment. We find that these images were extracted from a variety of distinct sources; no single type of website supplied the majority of images. The most common source – personal blogs – comprised 34% of images. This suggests that many images in our sample derived from personal websites, where images are typically selected and uploaded at the discretion of internet users, reflecting a real-world data generation process (i.e., reflecting images that real people elect to use on their personal websites). A related selection process is reflected in the second most popular source of images – business websites – where images are often designed and uploaded as part of marketing materials. News platforms are the fourth most common source (16%), followed by stock photo websites (10%), suggesting that industries characterized by image-editing processes contribute to the data. Based on these results, it is reasonable to suggest that the Google images we examine do, indeed, reflect a variety of real-world data generating processes, which are mediated not only by algorithms, but also by psychological factors motivating the human choice of which content to curate and upload. In other words, despite its limitations, this analysis quite clearly suggests that no single source of images (e.g., “stock photos”) appears to be artificially driving our results; that of course would be concerning, since it would suggest that our study is really a study of the market dynamics of the stock photo industry as compared to regular Google text search results. However, we find no evidence to support this concern. Stock photo websites were only identified as the fifth most common source of images. This needs to be taken with a grain of salt, since it is possible for people to purchase and copy stock photo images and reuse them on their personal and professional websites. However, we think this is unlikely to bias our results since less than 5% of images in our experimental data repeated across sources. The same is true in our observational data, where we find that only 11% of images repeat within or across searches, and all of our results are robust to the exclusion of repeat images. Altogether, these

findings suggest that the images in our sample derive from a variety of sources that engage a range of distinct, real-world data generating processes, suggesting that the gender bias we observe likely reflects a general bias that seeps into online images through multiple channels and mechanisms.”

We acknowledge the importance of these concerns and the implications of these supplementary findings in our discussion (from pg. 10 of the manuscript):

“A promising direction for future research is the social and algorithmic processes contributing to bias in online images, pertaining not only to gender, but also to race and other demographic dimensions. **The Google images we examine stem from various sources, with the most common source being personal blogs, followed by business, news, and stock photo websites (Fig. S33). This suggests that the bias we observe is driven partly by content that internet users elect to display on their blogs, and also by audiences’ preferences for which news to consume or images to purchase. Our supplementary results regarding celebrities on IMDb and Wikipedia (Fig. S4) reflect additional contributing factors relating to status dynamics and hiring biases in entertainment media.** In all cases, the human preference for familiar, stereotypical representations likely plays a role in perpetuating bias in online images [29]. We anticipate that the study of online bias will benefit from extending our multimodal approach across additional modes of communication, such as audio and video, and also by comparing both human and AI-generated content.”

Cited Reference:

Winkielman, P., Halberstadt, J., Fazendeiro, T. & Catty, S. Prototypes are attractive because they are easy on the mind. *Psychol Sci* 17, 799–806 (2006).

(2) Is the same algorithm underlying Google images compared to regular Google queries?

Reviewer 2 is correct that there are differences in the recommendation algorithms underlying Google’s text-based and image-based search engines. While research indicates that Google’s text and image-based search engines both rely on the pageRank algorithm (Yushi & Baluja 2008; Langville & Meyer, 2012), there is also work suggesting that biases relating to advertising and news influence Google’s text-based search results (Noble 2018; Fischer & Lelkes 2020), and it is unclear whether (or how) such factors influence Google’s image search results. We also know that Google’s image-based recommendation algorithm leverages image features to make its recommendations. As documented by Yushi & Baluja (2008) and Langville & Meyer (2012), the Google Images algorithm first extracts images that are relevant to the searched keyword through the aid of the metadata, including image taglines (this is similar to the text-based recommendation algorithm). The next step for the image algorithm is to calculate pairwise similarity scores among the images retrieved via the first step, such that the most similar set of images is extracted. As the final step, the ranking of images is informed by user behavior, such as which images they are most

expected to click on based on previous click-through behavior (this final step is also common to both image and text algorithms). We are happy to add this information to our supplementary materials.

While we are unable to directly observe and control for these algorithmic differences, we do not think they pose an issue for our findings, since we are interested in the distribution of perceived gender in online texts and images as they are encountered by regular internet users. Moreover, two important aspects of our study help to rule out the concern that differences in Google's recommendation algorithm can account for our findings regarding the differences between text and images.

First, in our revised experiment, our results equally replicate when asking participants to use either (i) the generic Google search engine (google.com) or (ii) to the Google News specific search engine (news.google.com; figure S25). This helps to rule out the concern that our results may be driven by idiosyncrasies in the recommendation algorithm underlying Google's generic search bar, since we observe essentially the same outcomes when examining the descriptions that participants upload from the Google News search engine specifically.

Second, we show that our main findings concerning the differences of gender association between images and text equally hold when analyzing data from Wikipedia (see fig. S2), including data associated with the ground truth self-identified genders of the people depicted (see fig. S3). This shows that our findings are not dependent on differences in the recommendation algorithm associated with textual and image data, since Wikipedia does not use a recommendation algorithm.

Memo References

- Fischer, Sean, Kokil Jaidka, and Yphtach Lelkes. "Auditing Local News Presence on Google News." *Nature Human Behaviour* 4, no. 12 (December 2020): 1236–44.
- Jing, Yushi, and Shumeet Baluja. "Pagerank for Product Image Search." In *Proceedings of the 17th International Conference on World Wide Web*, 307–16. WWW '08. New York, NY, USA: Association for Computing Machinery, 2008.
- Langville, Amy N., and Carl D. Meyer. *Google's PageRank and Beyond: The Science of Search Engine Rankings*. Princeton University Press, 2012.
- Noble, Safiya Umoja. *Algorithms of Oppression: How Search Engines Reinforce Racism*. 1 edition. New York: NYU Press, 2018.

4. The authors describe a number of associations with respect to the experiment on images and textual descriptions. This is an intriguing experiment, yet the exact mechanisms underlying the findings are somewhat underdeveloped. What is the micro-level mechanism here? Is there something about visual cues being more important? What does the literature conjecture about that? And do the authors know something about how well respondents read the textual descriptions? If not, the authors are comparing two different treatments that may (again) be inherently incomparable.

Our revised manuscript includes an improved replication of our experiment (pre-registration here <https://osf.io/3jhzx>) that more clearly identifies the micro-level

mechanism. We use this experiment (described below) to successfully identify the presence of two micro-level mechanisms: (1) that gender associations are more frequent in images than text, such that participants are more frequently primed with gender bias in the image condition; and (2) that, even when both text and images are explicitly gendered, images more strongly prime gender biases in people's beliefs than texts, consistent with prior work on embodied cognition and the picture superiority effect (references 13-17 in the manuscript). Our improved experimental design also allows us to evaluate these mechanisms by comparing both the text and the image condition to a neutral control condition (described below), while also validating our outcomes across two distinct versions of the text condition, one in which participants used the generic google.com and one in which participants used news.google.com (Fig. S25). We also evaluate the robustness of our outcomes while accounting for the statistical controls recommended by reviewer 2: we show that participants spent a reasonable amount of time evaluating descriptions in both conditions, and we further show that our results hold when controlling for the amount of time participants spent evaluating descriptions in each condition, as well as the overall number of descriptions that participants encountered in each condition. In what follows, we describe this improved experimental design and analysis strategy.

Our new experimental design includes: (i) a control condition, in which participants were asked to use Google to search for and upload either images or text for random categories (e.g., "apple" and "guitar") prior to rating the gender they most associated with each occupation; and (ii) a refined text condition, in which subjects were asked to search for textual descriptions from Google News specifically (all results replicate if subjects are asked to search in the general Google search engine).

A team of human annotators coded the gender association of all the textual and visual descriptions uploaded by participants (our response to comment 2 from Reviewer 2 details our method for labeling the gender association of each description). This allowed us to test and confirm our first hypothesized mechanism, namely that gender associations will be more prevalent in images than text, leading them to more consistently prime gender biases in people's beliefs (see Fig. 3ABC). Secondly, we test whether images are stronger than texts at priming gender biases when both modalities present a gendered description of the same occupation. Again, the results support our theory (see Fig. S32). Our main experimental results are provided in response to comment 2 by Reviewer 2. Here we provided the additional supplementary analyses discussed above.

(1) SOM section demonstrating the stronger priming effect of images compared to texts:

"A.4.4 Comparing the Strength of Gender Priming between Text and Images

In our main text, we focus on comparing the image and text condition in terms of the overall prevalence of gender biases in the descriptions that participants uploaded and the associated effects this has on the prevalence of gender biases

in participants' explicit and implicit beliefs. Here, we evaluate the additional prediction that images are stronger at priming gender biases in people's beliefs, even when the images and texts being compared are explicitly gendered. Figure S32 shows that when comparing only participants who provided explicitly gendered descriptions in the image and text condition, participants in the image condition reported significantly stronger gender biases in their associations with occupations ($p < .0001$, Student T-test), even when using a linear regression to control for the specific gender and the specific occupation associated with the uploaded description ($\beta = .06$, $SE = .01$, $p < .0001$). These findings indicate that even when gender is salient in both text and image, exposure to images leads to stronger biases in people's beliefs."

Figure S32: The average absolute strength of the gender associations that participants reported for each occupation in each condition, while excluding this analysis to only those descriptions that were explicitly gendered as either male or female in the text and image condition. The green (purple) vertical lines indicate average effects for the text (image) condition. $N = 2,570$ image descriptions; $N = 673$ text descriptions.

(2) SOM section demonstrating the robustness of our results to controlling for time and the number of sources participants encountered (pg. 69):

“A.4.2 Robustness of Experimental Results to Controlling for Time and the Number of Sources.

Here, we demonstrate that our experimental results are robust to controlling for how much time participants spent browsing and downloading descriptions (textual or visual) for each occupation, as well as controlling for the number of content sources they evaluated before selecting a description for each occupation. We begin by presenting the relevant descriptive statistics for our main Google News text condition and Google image condition. On average, participants in the Google News text condition spent 140 seconds (2.33 minutes) (min=15.4 seconds; max=1,740 seconds) browsing descriptions for each occupation (Panel A of Figure S31), which was distributed across an average of 3.5 online sources (min=1

source; max=65 sources) (Panel B of Figure S31), such that participants in this condition spent 56 seconds on average evaluating each potential online description. Participants in the Google image condition spent 81 seconds (1.35 minutes) (min=12.5 seconds; max=1,395 seconds) browsing images of each occupation (Panel A of Figure S31), which was distributed across an average of 9 online sources (min=1 source; max=100 sources) (Panel B of Figure S31), such that participants in this condition spent 21 seconds on average evaluating each potential online description. These descriptive statistics indicate that participants spent a considerable amount of time both browsing and evaluating online descriptions before making their content selections for each occupation, thus satisfying an important precondition for our theory, which assumes that people are paying sufficient attention to the descriptions they encounter online to permit these descriptions to prime (i.e., bias) their gender judgments.

Figure S31: (A) The number of seconds that each participant spent browsing and downloading descriptions for each occupation, split by the Google News text condition and the Google image condition; (B) the number of content sources each participant evaluated before selecting a description for each occupation split by the Google News text condition and the Google image condition; (C) The average number of seconds that each participant spent evaluating each online description before selecting a description for each occupation, split by the Google News text condition and the Google image condition.

Table S9 shows that participants in the text condition spent significantly more time browsing for online descriptions than those in the image condition ($p < .001$, $t = 19.33$ Student T-test, Two-tailed), while participants in the image condition reported evaluating significantly more online sources than those in the text condition. These differences are likely due to the fact that, cognitively speaking, people tend to be faster at processing visual information than verbal information [17]. Importantly, however, we show that these differences do not confound our main experimental results. Table S9 presents the results of an OLS regression predicting the absolute strength of participants' gender associations as a function of experimental condition (the Google image vs. Google News text condition), while controlling for (i) the number of seconds participants spent browsing content for each occupation, (ii) the number of online sources participants encountered when identifying a description for each occupation, and (iii) the number of seconds that participants

spent per online source for when seeking descriptions for each occupation. This model also includes fixed effects by occupation, as well as robust standard errors clustered at the participant level. We find that neither the time spent evaluating each occupation, nor the number of sources encountered for each occupation are significantly predictive of the absolute strength of participants' gender associations for each occupation. Crucially, we find that while holding time and the number of sources constant, participants in the image condition reported significantly stronger gender associations for each occupation than those in the text condition ($p < .01$, $\beta = .06$, $CI = [.05 - .08]$).

	Coefficients	Confidence Interval
Experimental Condition [Image]	0.06** (0.02)	0.05 - 0.08
Seconds per Occupation	0.00 (0.00)	0.00 - 0.00
Sources per Occupation	-0.00 (0.00)	0.00 - 0.00
Seconds per Source	-0.00 (0.00)	0.00 - 0.00
Occupation Fixed Effects	Included	
Constant	0.42*** (0.03)	0.37 - 0.48
N	6,140	
R ²	0.16	

Standard errors in parentheses (clustered by participant)

* $p < 0.05$, ** $p < 0.01$, *** $p < 0.001$

Table S9: An OLS regression predicting the absolute strength of participants' gender associations as a function of experimental condition (the Google Image vs. Google News text condition), while controlling for (i) the number of seconds participants spent browsing content for each occupation, (ii) the number of online sources participants encountered when identifying a description for each occupation, and (iii) the number of seconds that participants spent per online source for when seeking descriptions for each occupation. This model includes fixed effects for each occupation, and its standard errors are clustered at the participant level."

5. I think a statement such as "Textual measures of gender stereotypes are at risk of underestimating the prevalence of gender bias online" needs more development. I think the authors mean that when looking at different types of media or expressions online (e.g., avatars, news, images) each have different patterns of gender bias that may in some cases be stronger than in other cases. (Which might be expected.) I suggest the authors develop their premise based on a theoretical conjecture, instead of sketching it as a surprise, that some of these expressions might be stronger in some media compared to others (see point #1). They can then ask (and answer) the question what real-world implications that has in their experiment.

We agree. We have revised our introductory framing to focus on the contrast between textual and image communication, and to develop clear predictions for why we expect gender biases to be more prevalent (and stronger primes overall) in online images compared to texts. To maintain our focus on the contrast between text and images,

we have chosen to avoid mentioning the implications of our theory for other modes of communication, such as avatars or video, in the introduction, however we highlight this as a promising area for future research in our revised discussion. We have also revised our discussion section to elaborate on how our theory and findings advance a longstanding inquiry concerning the sociocultural and psychological effects of images dating at least as far back as Frederick Douglass’s prescient lecture on “Pictures and Progress.” We have copied these revisions below from the introduction and discussion.

From the introduction of the manuscript (pg. 1):

“Recent analyses of textual data from online sources, including social media and popular news outlets, report a steady reduction of gender bias over the last several decades [1, 2]. Yet, empirical studies show that progress toward gender equality has slowed or stalled throughout social life [3], including hiring practices [4], household management [5], and patterns of recognition and specialization in academia [6, 7]. **There is a puzzling disconnect between the textual representation of gender online and the current socioeconomic realities of gender inequality. This apparent disconnect may be driven by aspects of textual communication that do not generalize to online content as a whole. For example, while text can minimize bias by leveraging gender-neutral terminology, online images often concretely visualize people, such that demographic information is inevitably conveyed. Images may thus provide a clearer window into large-scale patterns of gender bias today.** However, most prior studies of online gender bias focus exclusively on text [1, 2, 8, 9], with only a few recent studies examining gender bias in a small sample of Google images [10–12]. The lack of multimodal methods and sufficiently rich image datasets has prevented prior work from systematically comparing gender bias across images and texts. Here, we address this gap by developing computational techniques for comparing gender bias in the representation of social categories across images and texts, which we apply to (i) a novel dataset comprising over one million images from Google, Wikipedia, and IMDb, mapped to 3,495 social categories, as well as to (ii) textual representations of these categories, using word embedding models trained on billions of word tokens from these platforms. We find that women are systematically under-represented in online images compared to texts, and that social categories are more strongly gendered in online images than online texts, public opinion, and US census data. Using a pre-registered experiment with a nationally representative US sample from Prolific ($N=450$), we show that googling for images rather than textual descriptions of occupations amplifies people’s implicit bias toward associating women with arts and men with science, a bias linked to pervasive inequalities in academia and industry [6, 7]. Strikingly, exposure to online images increased implicit gender bias for several days after our experiment.

Advances in natural language processing have spurred an avalanche of research focusing on the ways in which human cognitive biases are reflected in online texts

[1, 2, 8, 9]. Images, however, are equally foundational to human cognition [13–15]. Numerous studies have replicated the “picture superiority effect,” which shows that images are often more memorable and emotionally evocative than words [16, 17]. Based on this work, we hypothesize that images are more effective than texts at priming (i.e., psychologically activating) gender associations in people’s beliefs, providing a more powerful medium for the digital spread of gender bias, though this has yet to be shown empirically. While a few studies find that the images people encounter via Google predict gender biases in their beliefs about occupations, these studies focus on only 50 occupations or less using only a few thousand images, without the capacity to compare these biases to those in online texts from the same platforms [10–12]. It thus remains unclear whether gender bias is indeed prevalent online, especially given the recent evidence of decreasing gender bias both in online texts [1, 2] and public opinion [18].”

From the discussion of the manuscript (pg. 10):

“In this study, we show not only that gender bias pervades the images of the world’s most widely used search engine, Google, but also that this bias is stronger than those in online texts, public opinion, and the US census. The same findings hold when comparing images and texts from Wikipedia (Fig. S3), in contrast with its perceived status as a neutral source of information. The scale of these findings is especially alarming in light of mounting evidence that images from these sources are not only widely used by people [26], but also by machine learning algorithms [27, 28], with the potential to amplify biases in each. Indeed, our experiments show that images are stronger than texts at amplifying explicit and implicit gender bias in people’s beliefs. These findings draw attention to the potential for online images to negatively impact the well-being, social status, and economic opportunities for those who identify as women, with related implications for men in female-stereotyped categories, such as care-oriented occupations [6, 7, 23]. Thus, while recent work has reported improvements in the prevalence of gender bias in everyday language [1, 2], we must evaluate these narratives of progress with caution. The fight against inequality lags behind in the realm of images, which are ubiquitous and impactful.

[...]

At the dawn of photography, political philosopher Frederick Douglass forewarned of the potential for images to either subvert or reinforce stereotypes at large, writing in his 1861 lecture “Pictures and Progress” that “the great cheapness and universality of pictures must exert a powerful though silent influence upon the ideas and sentiment of present and future generations” [30]. Nearly two centuries later, despite the rapid proliferation of images across popular media, the discriminatory impact of online pictures remains largely silent. We have aimed in this study to break this silence.”

6. *My browsing history is probably pretty biased, but when I Googled for “athlete” I found: groups, blurry faces, side by side shots of athletes, book covers, watches, fashion models, laptops, cartoons, ensemble shots of athletes, fashion models, infographics, food supplements, and so forth. Can the authors give a bit more information on the data quality in that respect and how they handled these things? Is the number of actual faces recognized non-randomly distributed over the categories? And what if, say, a non-athlete sport journalist enters into the image results of “athlete”?*

These are excellent points. You are correct that a source of noise in our main image analyses stems from the inability to identify which Google search results correctly depict the intended category. A similar source of noise impacts word embedding measures of gender associations, since not all co-occurrence patterns between gender markers and social categories indicate the gender of the member of the category; for example, in the following sentence, ‘his’ does not refer to the ‘doctor’ (whose gender remains unspecified): “He gave the chart to his doctor.” Another important source of heterogeneity, as you note, is variation across categories in terms of the number of faces associated with these categories in Google Image search results. We have added two sections to our SOM to evaluate these sources of heterogeneity and to show the robustness of our results to controlling for these sources of heterogeneity. We describe these below.

(1) Robustness to human classifications of uncropped images (section A.2.15, SOM). For a subset of 300 categories, we recruited an additional 1,024 annotators to manually classify the faces in the original, non-cropped images. Crucially, we asked each annotator to classify the focal face in each image belonging to the specific category associated with this image in Google search (e.g., “doctor”) to minimize noise stemming from background faces unrelated to target the category being coded as associated with this category. As shown in figure S23, we find that uncropped Google images present significantly more male bias, and significantly stronger gender associations overall, compared to textual measures of gender bias for the same categories. The gender association for the same categories in our cropped and uncropped images are highly correlated ($r = 0.7$, $p < .00001$).

(2) Robustness to the number of images associated with particular categories. Here, we copy the revised section A.2.14, which evaluates heterogeneity across categories in terms of the number of faces they retrieve via Google Images, while also demonstrating the robustness of our results to controlling for this heterogeneity.

“A.2.14 Robustness to the Number and Ranking of Images in Google Image Search Results.

Here, we evaluate the robustness of our results while controlling for heterogeneity across social categories in terms of the number of faces associated with each category in Google Image search results. On average, each social category was associated with 48 faces in Google Images. To gain insight into what accounts for this heterogeneity, we find that the number of faces associated with each category

in Google Images was positively and significantly predicted by the frequency with which the social category was searched in Google Images across the US ($r = 0.08$, $p < .0001$; see section A.2.13); as well, the number of faces associated with each category in Google Images was also positively and significantly predicted by the frequency of each social category in the English language ($r = 0.13$, $p < .0001$; see section A.2.11). These analyses suggest that the number of faces associated with a social category is non-randomly distributed across categories as a function of the overall frequency of categories in general language use and in Google search activity online.

Predictors	Strength of Gender Bias			
	Estimates	std. Error	CI	p
(Intercept)	0.22	0.00	0.21–0.23	< 0.001
Measure [Images]	0.17	0.00	0.16–0.18	< 0.001
Number of Faces	0.00	0.00	0.00–0.00	0.08
Google Img. US Search Freq	0.09	0.03	0.04–0.16	0.003
Word Frequency Scaled	0.00	0.00	0.00–0.02	0.07
Observations	6852			
R ² / R ² adjusted	0.125 / 0.125			

Table S6: An OLS predicting the absolute strength of gender bias for each category, according to either our main image (Google Images) or text (Google News) measure, as a function of the number of faces associated with each category, the frequency with which each category is searched across the US in Google Images, and the frequency with which each category is used in the English language. Standard errors are clustered at the category level.

Importantly, Table S6 shows that the absolute strength of gender bias associated with each category is significantly higher in images than text ($\beta = 0.17$, $p < .001$), while controlling for the number of faces associated with each category, as well as the overall frequency of each category both in general language use and in Google Image search activity. Moreover, Table S6 shows that the number of faces associated with a category is statistically unrelated to the absolute strength of its associated gender bias ($\beta = 0.00$, $p = .08$), subject to these controls.

Figure S21: The vertical axis shows the difference in the strength of gender bias associated with each category according to our image measure (Google Images) and text (Google News text) measure. This difference is displayed while only examining categories associated with a minimum number of faces in Google images; the minimum number of faces used to subset the data is shown on the horizontal axis. Results are shown separately for male-skewed and female-skewed categories. Error bars display 95% confidence intervals.

These results are corroborated by figure S21, which displays the difference in the strength of gender bias associated with each category according to our image and text measure, as a function of the number of faces associated with each category in Google images. For this analysis, we only examine categories associated with a minimum number of faces in Google Images. The minimum number of faces associated with each category is shown along the horizontal axis. For example, where the minimum number of faces is 25, this means that we subset our data to only compare categories across images and text where these categories are associated with at least 25 faces in Google Images. The results show that, across all cutoff points for the minimum number of faces considered, images continue to display significantly stronger gender bias than text at the $p < .05$ level for both male- and female-skewed categories. These results show that heterogeneity in the number of faces associated with social categories in Google Images is unlikely to serve as a confound driving our results.”

7. If found the race question to coders to be somewhat problematic. The coders are asked to label races: “African American, Caucasian, Asian, Hispanic, Native American, two or more races, other”. So what happens if they find a Black non-US athlete? In other words, the coding is done with a US-centered approach, whereas Google image results are likely to provide international results for many social categories for which the categories are not applicable. Can the authors give more information about that?

We agree that racial categorizations introduce a number of subtleties along these lines. Since neither our main nor supplementary analyses examine the racial dimension of faces in our data, we believe that these concerns are beyond the scope of the current study and provide fertile ground for future studies. We have added a note to this effect in our discussion (copied below). Regarding the question of international results, our *SOM* now includes analyses confirming that our results replicate across Google search results from five countries (see section A.2.5).

From the discussion (pg. 10, most relevant edits in bold):

“A promising direction for future research is the social and algorithmic processes contributing to bias in online images, **pertaining not only to gender, but also to race and other demographic dimensions**. The Google images we examine stem from various sources, with the most common source being personal blogs, followed by business, news, and stock photo websites (Fig. S33). This suggests that the bias we observe is driven partly by content that internet users elect to display on their blogs, and also by audiences’ preferences for which news to consume or images to purchase. Our supplementary results regarding celebrities on IMDb and Wikipedia (Fig. S4) reflect additional contributing factors relating to status dynamics and hiring biases in entertainment media. In all cases, the human preference for familiar, stereotypical representations likely plays a role in perpetuating bias in online images [29]. We anticipate that the study of online bias will benefit from extending our multimodal approach across additional modes of communication, such as audio and video, and also by comparing both human and AI-generated content.”

8. I was unsure whether the gendered social categories were excluded beforehand? I suggest the authors make the analyses on the social categories that exclude the gendered social categories the central analyses of their paper. (And what is the distributions of genderedness in these social categories?) And are there more social categories going to one side compared to other sides (e.g., more husbands, uncles, etc)?

Given our goal of providing a comprehensive analysis of nearly all common social categories according to WordNet, we have chosen to maintain the use of all categories in our core analyses. To address this important concern, our revised supplementary appendix includes an additional analysis showing that our main results hold – and in fact, are slightly stronger – when excluding gendered categories. Moreover, WordNet’s coverage of gendered categories was relatively even (52% were male). We have copied this analysis below (pg. 38, *SOM*).

Figure S15: The gender associations for all 2,922 non-gendered social categories in WordNet according to Google Images (purple) and textual embeddings of Google News (green). Solid purple (green) vertical lines indicate the average gender association according to images (text).

“Finally, we show that our main results are highly robust to the exclusion of explicitly gendered categories, such as brother, sister, uncle, and aunt. There was no significant gender bias in the gendered categories that were excluded (52% were male). Excluding these categories left 2,922 non-gendered social categories in WordNet for analysis. Figure S15 compares gender associations across images and text using only the 2,922 non-gendered categories in WordNet. We show that Google Images of non-gendered categories skew significantly toward male representation, both in general ($p < .0001$, average gender association 0.16) and in comparison to text ($p < .0001$, average gender association 0.05) (Wilcoxon Signed-Rank Test, Two-tailed). Similarly, when examining only non-gendered categories, the strength of gender associations remains significantly higher in images than text for categories with male-skewed ($p < .0001$) and female-skewed ($p < .0001$) associations (Wilcoxon Signed-Rank Test, Two-tailed). Thus, these results show that our main findings are robust to controlling for whether categories are explicitly linguistically gendered.”

9. *I am a bit concerned whether the definition of the “stereotype” that the authors give is one that will match the literature on stereotypes; is overrepresentation in category a stereotype (which might be an empirical finding)? Or is the expectation of certain genders in a category a stereotype? Notice that this is a nuanced but important difference.*

Thank you for raising this interesting and nuanced question. Initially, building on the operationalization of Kay et al. (2015) and Metaxa et al. (2021), we viewed our approach as consistent with Ellemers’ (2018, ref. below) statistical view of gender stereotypes as “general expectations about members of particular social groups,” on the assumption that such associations reflect statistical expectations for the gender of those who belong to these categories. However, further research into the literature on stereotypes revealed several competing definitions, leading us to worry that using the

term “stereotype” may cause confusion when presenting our research. For this reason, in light of *Nature’s* succinct format and focus on cross-disciplinary accessibility, we have decided to remove all mentions of stereotypes from our article, and instead refine the description of our study as focusing on “gender bias” as a form of statistical bias in terms of both overrepresentation and the statistical strength of gender associations for occupations and related categories, as compared to the empirical distribution of gender in people’s beliefs and the U.S. census. We have added the following paragraph to our manuscript to clarify this focus.

From pg. 3 of the manuscript:

“Using these measures, we quantify gender bias as a form of statistical bias along three key dimensions. First, we examine the extent to which social categories are associated with a specific gender in images and texts. Second, we examine the overall extent to which women are represented compared to men across social categories in images and texts. Third, we compare the gender associations in our image and text data to the empirical representation of women and men in public opinion and US census data on occupations. These analyses allow us to test not only whether gender bias is statistically stronger in images than texts, but also whether this bias reflects a distorted representation of the underlying distribution of gender in society.”

Memo reference:

Ellemers, Naomi. "Gender stereotypes." *Annual review of psychology* 69 (2018): 275-298.

10. *In what way does the study contribute to the study mentioned under reference #21?*

We maintain that there are four key ways in which our study marks a substantial and impactful advance upon the Kay et al. (2015) conference paper, and indeed all recent studies on the topic that we are familiar with:

(1) Comparing text to images (observational). The prior studies referenced by the Reviewers focus on gender bias either in images (e.g., Kay et al. 2015; Metaxa et al. 2021) or text alone (e.g., Caliskan et al., 2017). We can find no prior work that identifies differences between texts and images in how they encode gender bias. We address this gap by developing computational methods for comparing gender bias across images and text, while also using a novel experiment to compare the ability of images and text to reinforce gender bias in people’s beliefs.

(2) Comparing text to images (experimental). Our experiment, which we have improved and replicated with a control condition, not only reveals that gender bias is more prevalent in online images than text, but also that images more strongly prime gender bias than online texts. This involves a novel integration of prior research on the “picture superiority effect” and prior work on priming effects in the study of gender bias. Neither Kay et al. (2015) nor all related studies we are familiar with identify this foundational difference between images and text and demonstrate its psychological

consequences, which have far-reaching implications for image-based search engines and social media.

(3) Insights into the effects of images on implicit bias in an ecologically-valid experiment. Kay et al. (2015) and Metaxa et al. (2021) examine the ability for gender bias in online images to prime biases in people's explicit beliefs. We go beyond by measuring the impact of texts and images on both explicit and implicit bias over time. This required integrating the Implicit Association Test (IAT) into a novel online experiment in which participants used Google to search for content in real-time. These efforts reveal that only online images significantly and enduringly increased participants' implicit gender bias. See our reply to (4.) below for an explanation of the robustness of our IAT. Another novel feature of our experiment is that it required participants to leverage Google to search for information in a realistic manner, whereas Kay et al. (2015) and Metaxa et al. (2021) test the effects of exposing people to artificially created image sets without engaging them in a search task. This lends our design considerable ecological validity, which is highlighted by Reviewer 4 as another impactful contribution of our experiment.

(4) Scale. Kay et al. (2015) and Metaxa et al. (2021) only investigated approximately 5100 images mapped to 51 occupations. The dataset collected and analyzed in our study is almost 100 times larger. We study gender bias in the images associated with nearly 700 occupations, and 3,495 social categories in total, not only from Google images, but also Wikipedia; altogether, our sample consists of over 600,000 images. Furthermore, our revised *SOM* also tests our theory using a dataset of over 500k images from IMDb and Wikipedia. Combined, our entire image dataset comprises over one million images, and is mapped to the full known taxonomy of social categories in the English language, enabling a range of analyses infeasible for prior research.

11. I doubt whether the analysis on Google searches for the top-100 images for the gendered versions (e.g., female athlete/male athlete) is useful to the authors' argument: finding that both versions each retrieve 100 pictures is only logical given how many websites Google can retrieve those pictures from (millions?). This goes back to my point on the data-generating processes with respect to these pictures. So the conclusion that the availability of pictures for the genders is not different seems inaccurate. The second part of that conclusion ("algorithmic differences") may be correct, but the authors need to develop more intuitions about that (see point 2).

Reviewer 2 has convinced us that our gender-specific searches are not a satisfactory way to control for the overall availability of images associated with each gender online. We also agree that the intuition underlying our prior efforts to examine algorithmic differences required further development. In the end, we decided that these analyses of algorithmic effects were distracting from the main results of this paper. For this reason, we have decided to include the results of our gender-specific searches in a simpler form that more directly follows from our main analyses. In our *SOM*, we now show that men are significantly overrepresented in female-specific searches compared to the representation of women in male-specific searches. This is a kind of

error analysis. We ask: when searching for female-specific searches, what is the likelihood that Google mistakenly returns a male face, and how does this error rate compare to the likelihood that Google mistakenly returns a female face when making male-specific searches? Indeed, we find that when making gender-specific searches, Google's 'errors' are significantly more likely to favor male-representation. We frame this not as an indication of biases in Google's algorithm *per se*, but rather as another indication of the extent of over male-representation in online images. We provide the relevant revisions from the manuscript and the appendix below:

From pg. 5 of the manuscript:

“Critically, we also find that women are systematically under-represented in images compared to texts. Figure 2A shows that texts from Google News exhibit a weak bias toward male representation ($p < .0001$), whereas this bias is over four times stronger in images from Google Images ($p < .0001$), (Wilcoxon Signed-Rank Test, Two-sided). While 56% of categories are male-skewed according to Google News, 62% are male-skewed according to Google Images ($p < .0001$, Proportion Test, Two-sided). Supplementary analyses show that the under-representation of women is even more pronounced when using a deep learning algorithm to classify gender in the same online images (Figs. S9-S11). **Supplementary analyses further show that men continue to be over-represented even when searching for “female” and “male” images of each category in Google Images, yielding 491,169 additional images (Figs. S12 and S13).**”

From pg. 33 the SOM:

“Using this data, we examine whether the Google search engine is more successful at retrieving male rather than female examples of the same social category when explicitly asked to do so. Specifically, for each category, we calculate the probability that a male-specific search returns a male face, and from this we subtract the probability that a female-specific search with the same category returns a female face. Positive values indicate that male-specific searches were more effective at returning male faces than female-specific searches were at returning female faces, for the same category.

Figure S13: (A) The distribution of gender associations for 3,096 categories in Google Images, shown separately for female- and male-specific searches of the same category (e.g., searching “male doctor” and “female doctor” separately). (B) The bias score for each category is determined by subtracting the probability that a female-specific search returns female faces from the probability that a male-specific search returns male faces, for the same category (positive values indicate a bias toward male representation). ***, $p < .0001$.

Panel A of Figure S13 confirms, as expected, that female-specific searches yielded significantly more female search results, while male-specific searches for the same categories yielded significantly more male search results. However, panel B of Figure S13 shows that the ability for Google to successfully provide male faces in response to male-specific searches is significantly better than its ability to provide female faces in response to female-specific searches ($p < .0001$, Wilcoxon Signed Rank Test, Two-tailed). Indeed, for 58% of categories, Google was more successful at retrieving male than female faces for gender-specific searches, exhibiting significant male bias ($p < .001$, Proportion Test, Two-tailed). Thus, even when requesting equally female and male search results, we observe a significant bias toward the over-representation of men in Google Image search results.

Consistent with these findings, we also show that when Google provides the wrong gender for a gendered query – e.g., by showing a female when searching for a “male doctor” or showing a male when searching for a “female doctor” – such errors are significantly more likely to favor male representation. Men are significantly more likely to mistakenly appear in female-specific searches (27%) than women are to appear in male-specific searches (23%), ($p < .001$, $N = 2,960$ categories, Student’s T-test, Two-tailed). This error rate significantly favors male representation for 58% of categories ($p < .001$, Proportion Test, Two-sample).

Referee #3 (Remarks to the Author):

1. The research question is worth investigating. The paper is well-written, the methodology is rigorous and innovative, and the analyses are thorough. I have a few

questions, comments, and some suggestions that I hope will be helpful to the authors and contribute to making this paper even more impactful.

Thank you!

2. I really appreciate that the authors not only document the gender bias in online images but also test the consequences of exposure to such biases using experimental methods (also a longitudinal component). However, they only compare the effects of images vs. text, without a true control condition. It would seem informative to include a true control in order to get insights into participants' baseline gender biases and how those may be impacted by exposure to images/text. This is an open question that might be worth addressing in the paper's discussion or future directions.

This is an excellent suggestion. To address your comment and directly build on your suggestion, we improved the design by adding a proper control condition. To maximize thoroughness, we pre-registered a full replication of our experiment, while including a true control condition. The link to this pre-registration can be found here (<https://osf.io/3jhzx>); it contains the precise details regarding the number of subjects and categories these subjects were asked to search for and evaluate (along with the specific categories used in this task). This improved design also addresses another Reviewer's comment (#5 by Reviewer 4) by including a novel version of the text condition where participants were directed to search and upload textual descriptions directly from Google News. Our revised manuscript describes the design and results of this experiment on page 9, while focusing on the contrasts between the control, image, and the Google News version of the text condition (which we believe is the most appropriate version of the text condition to compare against). We have copied these revisions here for your review:

“What consequences do these biases in online images have on internet users? To conclude, we report the results of a pre-registered experiment² designed to test the impact of online images on people's explicit and implicit gender bias. In this experiment, we recruited a nationally representative sample of US participants from Prolific (N = 450) who were tasked with using Google to search for descriptions of occupations relating to science, technology, and the arts (Fig. S24). Each participant used Google to retrieve descriptions of 22 randomly selected occupations from a set of 54 (SOM). For the experimental treatments, participants were randomized into either (i) the text condition, in which participants used Google News to search for and upload textual descriptions of these occupations or (ii) the image condition, in which participants used Google Images to search for and upload images of occupations. After uploading the description for each occupation, each participant was asked to rate which gender they most associate with the occupation being described, using a -1 (female) to 1 (male) scale (SOM). To evaluate these experimental effects, participants were also randomized into a control condition that used the same task design, except that participants used Google to search for and upload either images or textual descriptions of random categories (e.g., “apple” and “guitar”) prior to rating the gender they associate with

each occupation. Asking participants to search for textual descriptions in the generic Google search bar instead of the Google News search bar yields indistinguishable outcomes (Fig. S25).

After completing the search task for all occupations, participants undertook an implicit association test (IAT) [22], a standard method in psychology for detecting implicit biases. We adopt an IAT designed to detect the bias toward associating women with liberal arts and men with science (Figs. S26-29), since prior work demonstrates the robustness of this IAT [18] and its ability to predict judgments and behaviors [23] relating to a stark pattern of inequality in industry and academia [3, 6, 7]³. We administered the IAT to participants immediately after the experiment, as well as three days later. Participants' implicit bias was measured using the standard IAT dscore [22], which tracks whether participants are faster at associating women with liberal arts and men with science (positive dscores indicate a bias toward associating women with liberal arts and men with science, *SOM*).

We begin by examining the extent of gender bias in the descriptions uploaded by participants. A team of annotators labeled each textual description as either female, male, or neutral based on whether it used female or male pronouns or names to describe the occupation (e.g., a description referring to a “doctor” as “he” would be coded as “male”); textual descriptions were identified as neutral if they did not ascribe a particular gender to the occupation (*SOM*). Similarly, a team of annotators labeled the gender of the focal face in each uploaded image as either female, male, or neutral; faces were coded as neutral if they contained no face or an indecipherable face. Then, for each occupation, we calculated the gender balance of the descriptions provided by participants by computing the average gender association across all descriptions. A key feature of this analysis is that it provides another method for comparing gender associations across images and texts that does not rely on word embedding models, while also ensuring that the images and texts being compared were collected by users during the same time period.

Fig. 3. Differential effects of googling for visual rather than textual descriptions of occupations on participants' explicit and implicit gender biases, as compared to a neutral control condition ($N=450$). (A) The average absolute strength of the gender associations that participants uploaded for each category in both the text and image condition. (B) The average absolute strength of the gender associations that participants reported for each occupation in each condition. (C) The average absolute strength of the implicit bias (dscore) that participants exhibited in each condition. (D) The correlation between the average strength of participants' self-reported gender associations and their implicit bias toward associating men with science and women with liberal arts. (E) The correlation between the gender associations of the descriptions that participants uploaded and the gender associations they reported for each occupation, split by condition. M., Male; F., Female. (F) The correlation between the average strength of the gender associations of the descriptions that participants uploaded and average strength of the gender associations they reported for each occupation, split by condition. All error bands and error bars display 95% confidence intervals.

Consistent with our observational results, panel A of Figure 3 shows that the descriptions participants uploaded were significantly more gendered in the image condition than the text condition ($p < .0001$, Wilcoxon Signed-rank Test, Two-sided; the gender associations for each occupation are highly correlated with those detected for the same occupations in our main observational analyses of Google Images and Google News, Fig. S30). Panel B of Figure 3 shows that exposure to more gendered stimuli in the image condition led participants to report significantly

stronger gender associations as compared to participants in both the text ($p < .001$) and the control ($p < .001$) condition, while there was no significant difference between those in the text and control condition ($p = .56$) (Wilcoxon Signed-rank Test, Two-sided). These results hold when controlling for the number of online sources that participants encountered, and the amount of time they spent evaluating online descriptions (Fig. S31; Table S9).

Concerningly, Panel C of Figure 3 shows that exposure to more gendered descriptions in the image condition also led participants to exhibit significantly stronger implicit bias toward associating women with arts and men with science. There was no significant difference between participants' implicit bias in the text and control condition ($p = .19$), yet participants in the image condition exhibited significantly stronger implicit bias than those in the control condition ($p < .001$) (Wilcoxon Signed-rank Test, Two-sided). Indeed, panel D of Figure 3 identifies a robust correlation between the strength of participants' self-reported gender associations and the strength of their implicit bias, both of which are amplified in the image condition ($p < .0001$, Jonckheere-Terpsta Test). Together, these findings suggest that exposure to more gendered descriptions in the image condition more strongly primed participants' explicit and implicit gender associations.

This priming mechanism is further supported by panel E of Figure 3, which shows a strong correlation between the gender associations in the descriptions that participants uploaded and the gender associations in their own explicit responses across occupations ($\rho = .8$, $p < .0001$); as well as by panel F of Figure 3, which shows a strong correlation between the overall strength of gender associations in the descriptions that participants uploaded and the overall strength of the average gender associations they reported across occupations ($\rho = .57$, $p < .0001$). These results hold for both the image and text condition, and across occupations (Table S10).

Importantly, we find that images differ from text not only in the prevalence of gender bias they contain, but also in their ability to prime gender bias in people's beliefs. Participants who uploaded gendered images reported significantly stronger gender bias compared to participants who uploaded gendered textual descriptions of the same occupations ($p < .0001$, Student T-test, Two-tailed; Fig. S32). Thus, even when gender is salient in both text and image, exposure to images leads to stronger bias in people's beliefs. Most strikingly, only participants in the image condition exhibited significantly stronger implicit bias three days after the experiment (Table S11), indicating that these outcomes go beyond ephemeral priming effects and may contribute to enduring biases."

3. I'm impressed by the scale of the image analysis--298,600 images compared to the vast majority of content analyses, which use only a few hundred or sometimes (rarely) a few thousand. That said, I'm curious as to the inter-coder reliability of the 2500 Mturk coders. Although they were coding only target gender (which is usually pretty straightforward and objective), I also think that it would be informative to have a sense of

how reliable these coders were. Usually, content analysis employs 2-5 coders, and reliability metrics for each variable are reported (e.g., Krippendorff, 2004; Neuendorf, 2002). This is not a deal-breaker in my opinion because, again, the only variable coded was target gender; still, if there was any information that the authors could provide as evidence of high reliability among the human coders, that would increase the credibility of the data.

While Reviewer 3 maintains that this issue is not a ‘deal-breaker’, we felt that addressing this issue with additional data collection and robustness tests would make a substantial contribution toward strengthening our results, while also helping us to address several other concerns from the Reviewers. Our response to this point involves four key advances:

(1) Replicating data collection with three unique coders per image and demonstrating strong intercoder reliability. We replicated our full data collection pipeline so that three unique coders labeled each image, such that we determined the gender of each face by identifying the modal classification across these three coders. This involved nearly tripling our sample size of annotators to 6,392 unique coders. In SOM section A.2.8, we show that our coders agreed 91% of the time in their gender classifications, constituting high intercoder reliability, and that our main results strongly hold when using a regression to control for the rate of intercoder agreement and the demographics of annotators.

(2) Validating the accuracy of our coders in a ground truth dataset. For a random sample of 250 coders, we invited them to complete an additional validation task in which they classified the gender of faces for which the self-identified “ground truth” gender of each person is known (see section A.2.7 in the SOM). These coders identified the “correct” self-identified gender 97% of the time, with stable accuracy rates across all age groups.

(3) Successfully testing our theory in a ground truth dataset of images for which the true gender of the faces depicted is known. We test and confirm our theory by analyzing the distribution of gender in the canonical IMDb-Wikipedia data (Rothe et al. 2018), which consists of over 500k images of celebrities for which the true self-identified gender of the celebrities is known (see section A.2.2). We show that online images of celebrities from both IMDb and Wikipedia encode significantly stronger gender associations -- and significantly more male gender associations - as compared to the U.S. census and textual embeddings from Wikipedia.

(4) We replicate our findings using machine learning classifications of both the cropped and uncropped data. As an additional test of whether our results hold while accounting for annotator subjectivity, we show that our results replicate when using a popular deep learning classifier from OpenCV to classify the gender of faces in the cropped and uncropped versions of our image data (Fig. S10 & S11). We prefer to present our main results based on human classification judgments, since deep learning models often perform more accurately at classifying the gender of men (Buolamwini 2017; Nagpal et al. 2019); and indeed, we show that OpenCV exhibits significantly higher levels of classification confidence when labeling faces as “male”,

exposing gender-related biases in its performance (Fig. S9). As well, our primary interest is in the impact that online images have on the opinions of internet users, such that the “perceived gender” by people is our central target, consistent with the primary outcome of interest in our online search experiment (Fig. 3). Nevertheless, we present the results based on automated classifications in our supplementary materials because they allow us to test whether our main results hold in the absence of subjective human judgments. Indeed, figure S10 shows that men are significantly over-represented in our dataset according to OpenCV’s classifications of our cropped and uncropped datasets, and importantly, figure S11 shows that these results hold when controlling for OpenCV’s confidence in its classifications. The gender distribution according to OpenCV’s classifications is highly stable regardless of whether we examine the cropped or uncropped images.

Memo references:

- Buolamwini, Joy A. *Gender shades: intersectional phenotypic and demographic evaluation of face datasets and gender classifiers*. PhD thesis, Massachusetts Institute of Technology, 2017.
- Nagpal, Shruti, Maneet Singh, Richa Singh, and Mayank Vatsa. Deep learning for face recognition: Pride or prejudiced? *arXiv preprint arXiv:1904.01219*, 2019.
- Rothe, Rasmus, Radu Timofte, and Luc Van Gool. “Deep Expectation of Real and Apparent Age from a Single Image Without Facial Landmarks.” *International Journal of Computer Vision* 126, no. 2 (April 1, 2018): 144–57. <https://doi.org/10.1007/s11263-016-0940-3>.

4. Even with all the precautions taken to gather the images, was there any processing to ensure that each image in the dataset was unique? And, what about non-human images (e.g., icons, drawings, animations)? These are fairly common in Google Images when searching for any kind of occupational category. Were those included or filtered out? Could this point be made clearer in the paper?

Thank you for these helpful suggestions. We were able to assign each image a unique identifier by combining its metadata, including its source website, with pixel-level analysis and manual inspection. Using this information, we have added a table S8 to our supplementary appendix which shows that our results hold while controlling for whether images are repeated within and across searches.

Regarding the question of avatars, Reviewer 3’s comment inspired us to incorporate a question about whether faces are cartoon or avatars when recollecting annotators’ classifications of our data to demonstrate the robustness of our results to controlling for intercoder reliability. Indeed, 8% of images were flagged as avatars. Table S7 also shows that all of our results continue to hold while controlling for whether faces were identified as avatars. We refer to this robustness test on pg. 5 of the manuscript. We have chosen to keep all faces in our main dataset because we believe that both photographic and illustrated representations of people are important for understanding the gender associations of a particular category.

5. I was confused about the number $N=5,972$ (last paragraph on p. 3). There were 2,986 social categories in Wordnet with word embeddings in the Google news corpus;

why is each category counted twice toward the N? Figure 1A clearly states that 2,986 categories are included in the analysis. Maybe I'm missing something here, but it would be helpful for the authors to clarify.

Thank you for pointing out this miscommunication. The original N=5,972 simply referred to the total number of data points being compared, which in this case consisted of 2,986 categories across two modalities (image and text). To clarify this ambiguity, we have chosen to state the N specifically as N=2,986 categories. This revision is reflected in all of our statistical reporting.

6. In Figure 2D, why focus on images of male-dominated occupations only? It would also be informative to see whether the same extreme representations (relative to census and text) is evident for images of female-dominated occupations.

We are grateful to Reviewer 3 for this helpful suggestion. We have revised Figure 2 so that former panel D (now panel C) compares our image, text, and census measures in terms of their average association along the -1 to 1 gender dimension, which does not prioritize examining male-dominant categories. The results directly support our theory.

“While these analyses show that gender associations are more strongly biased in images than texts, an open question is whether images present a biased representation of the underlying empirical distribution of gender in society. We show that online images exhibit significantly stronger gender bias compared to public opinion and US census data on occupations.

To compare our results to public opinion, we hired a separate panel of 2,500 annotators from Mturk who used the same -1 (female) to 1 (male) scale to provide their opinions about the gender they most associate with each category in our dataset (SOM). While both our image and text measures are highly predictive of gender associations in public opinion, panel B of Figure 2 shows that texts significantly *underestimate* male bias in public opinion ($p < .0001$), whereas images significantly *overestimate* it ($p < .0001$), (Wilcoxon Signed-Rank Test, Two-sided).

Figure 2: (A) The distribution of gender associations for 2,986 social categories in images from Google Images and texts from Google News. The image-based measure captures the frequency of female and male faces associated with each category in Google Image search results (-1 means 100% female; 1 means 100% male); the text-based measure captures the frequency at which each category is associated with male or female terms in the Google News corpus (-1 means 100% female; 1 means 100% male associations). Solid green (purple) line indicates the average gender association according to text (images). (B) The correlation of gender associations, paired at the category level, as measured by these online images and texts, as well as internet users' ($N=2,500$) judgments of each category. Human coders indicated their beliefs about the gender representation of each category by moving a slider along the same -1 to 1 scale (horizontal axis displays the average human judgment across evenly spaced bins). (C) The gender association of 687 occupations according to (i) textual patterns in Google News (green), (ii) the empirical distribution of gender in the 2019 US census Bureau of Labor Statistics (grey), and (iii) Google Images (purple). Error bars display 95% confidence intervals.

We also compare our measures to the frequency of genders across occupations according to the 2019 US Bureau of Labor Statistics (687 occupations could be matched between our data and the census). Panel C of Figure 2 shows that, according to texts from Google News, the gender association of these occupations is neutral and significantly less male than the census ($p<.001$) and Google images ($p<.001$) (Wilcoxon Signed-Rank Test, Two-sided). By contrast, while these occupations are male-skewed in both the census and Google images, the same occupations are significantly more biased toward male representation in Google images ($p<.001$, Wilcoxon Signed-Rank Test, Two-sided)."

7. I found the authors' definition of stereotypes on p. 1 to be a bit unconventional ("the over-representation of a demographic in the definition of a social category (e.g., female

nurses)"). *It seems to me like this is the operational definition of "gender stereotype" in the current research, but as a conceptual definition of the construct of stereotype it seems rather limiting.*

Within the psychological literature, stereotypes are more commonly defined as: "general expectations about members of particular social groups" (Ellemers, 2017) (where "social group" refers to demographic groups such as men/women but could also refer to many other groups, e.g., "church goers"). Other definitions are more specific about what these "expectations" are about, e.g., "assumptions about traits and behaviors that people in a labeled category are thought to possess" (Kite, Deaux, & Haines, 2008). Gender stereotypes, which are of central interest in the authors' research, are usually defined along the lines of: "consensual beliefs about the attributes of women and men" (Eagly, Nater, Miller, Kaufmann, & Sczesny, 2020).

Similarly, in one of the references the authors cite for their definition of stereotype (Gorman, 2005), the author wrote: "Gender stereotypes are cultural constructs, shared at the societal level, that describe what men and women are "known" to be like (Fiske 1998:377–78)". In sum, stereotypes in general and gender stereotypes in particular are usually defined more broadly than by alluding to the distribution of people from a given demographic group into a social category (which, in the context of this paper, refers to an occupational role or social role such as "cousin"). I would strongly recommend to either revise the definition or clarify that it refers to the operationalization of stereotype within the current research.

Thank you for raising this interesting and nuanced question. As you note, the stereotype literature is rife with competing definitions, some focusing on evaluative/discriminatory judgments against members of social groups, and others focusing more generally on expectations or features associated with someone as a function of their group membership. As you highlight, the literature on which we build (e.g., Kay et al. 2015 and Metaxa et al. 2021) typically rely on a simplified operationalization of stereotypes as count-based associations of each gender with a given category. Like this work, we viewed our approach as resonating with Ellemers' (2018) statistical sense of stereotypes as "general expectations about members of particular social groups," since our measures of images and text can be thought of as capturing which gender is most frequently associated with (and therefore, most statistically expected to be a member of) particular occupational and related categories. However, despite this connection, the landscape of competing definitions surrounding the term "stereotype" led us to worry that using this term in our manuscript may cause confusion among a diverse readership. In light of *Nature's* succinct format and focus on cross-disciplinary accessibility, we have decided to remove all mentions of stereotypes from our article, and instead refine the description of our study as focusing on "gender bias" as a form of statistical bias in terms of both overrepresentation and the statistical strength of gender associations for occupations and related categories, in comparison with the empirical distribution of gender by category in people's opinions and the U.S. census. We have added the following paragraph to our manuscript to clarify this focus.

From pg. 3 of the manuscript:

“Using these measures, we quantify gender bias **as a form of statistical bias** along three key dimensions. First, we examine the extent to which social categories are associated with a specific gender in images and texts. Second, we examine the overall extent to which women are represented compared to men across social categories in images and texts. Third, we compare the gender associations in our image and text data to the empirical representation of women and men in public opinion and US census data on occupations. These analyses allow us to test not only whether gender bias is statistically stronger in images than texts, but also whether this bias reflects a distorted representation of the underlying distribution of gender in society.”

Memo Reference:

Ellemers, Naomi. "Gender stereotypes." *Annual review of psychology* 69 (2018): 275-298.

8. *It was unclear what was meant by “the digital gender gap.”*

We agree that despite the popularity of this phrase, it remains ambiguous. Inspired by your comment, we decided to change the title of our paper to more clearly articulate our core contribution. Our title now reads: **“Online Gender Bias is Stronger in Images than Text”**. We have removed all other mentions of the digital gender gap in the manuscript.

9. *It seems like one possible explanation for why online images may show more bias than online text is that online text may be more likely than online images to refer to real people. Even if there are few women in a given male-dominated occupation, something may be written about those women. Images, in contrast, are less likely to refer to real people—in many cases, online images involve models (e.g., istock photos). Thus, online images are more likely to show a category exemplar or prototype, whereas text may include a wider range of gender representations.*

This is a fascinating suggestion. Several of our newly added analyses help to shed light on this possibility, which we describe in detail below. First, we want to highlight a key revision we have made to our manuscript to highlight this possible mechanism. We feel that our supplementary analyses on this point are not yet conclusive enough to properly evaluate this potential mechanism, which we believe points to a promising direction for future research. For this reason, we have added the following paragraph to our discussion highlighting this direction (from pg. 10):

“A promising direction for future research is the social and algorithmic processes contributing to bias in online images, pertaining not only to gender, but also to race and other demographic dimensions. The Google images we examine stem from various sources, with the most common source being personal blogs, followed by business, news, and stock photo websites (Fig. S33). This suggests that the bias we observe is driven partly by content that internet users elect to display on their

blogs, and also by audiences' preferences for which news to consume or images to purchase. Our supplementary results regarding celebrities on IMDb and Wikipedia (Fig. S4) reflect additional contributing factors relating to status dynamics and hiring biases in entertainment media. **In all cases, the human preference for familiar, stereotypical representations likely plays a role in perpetuating bias in online images [29].** We anticipate that the study of online bias will benefit from extending our multimodal approach across additional modes of communication, such as audio and video, and also by comparing both human and AI-generated content.”

Relevant Supplementary Analyses

We have added supplementary analyses which show (i) that our main findings hold in a context where we only examine images depicting real people; (ii) we quantify the rate at which images in our main image dataset present digital avatars or cartoons, which are more likely to skew toward fictional representations, and we show that all of our main results hold while controlling for whether images are photographic or avatars; and (iii) regarding potentially fictional photographic images (e.g. stock photos), we show that a fairly low fraction of our images are likely to derive from stock photo websites, and that our main results hold when accounting for stock photos. Altogether, these analyses indicate that the differential rate of fictional representations across online images and texts is unlikely to be the primary driver underlying our results, though it is likely one among several interesting factors at play. Most importantly, our focus in this study is on the rate at which particular genders are perceived by internet users as being associated with particular occupations and related categories, and for this purpose, our theory generalizes across whether users' gender perceptions derive from real or fictional content.

(1) Analyzing images of real celebrities over Wikipedia. Our *SOM* now includes an analysis of a canonical image dataset of celebrities as depicted over Wikipedia, where each image in this dataset is associated with the self-identified gender of the celebrity depicted based on the gender listed on their biographical Wikipedia page (see Rothe et al. 2018). We compare these image results to the gender association of the category “celebrity” according to leading word embedding models of Wikipedia text, as well as the U.S. census. This analysis speaks to your intuition that images may be more skewed toward depicting fictional rather than real people, as compared to text, since this is a context where we know the images are in fact depicting real people. Nevertheless, this analysis replicates our core finding: even when the images analyzed are known to represent real people, we find that images exhibit a significantly stronger male bias than both textual representations and the census, *and* images present stronger absolute gender associations overall as compared to textual representations and the census (see section A.2.2).

(2) Accounting for avatars and stock photography. Another way in which we've explored this intuition is by analyzing the frequency with which (i) fictionalized avatars and cartoons, as well as (ii) stock photography images appear in our image data. In

our revised materials, our image results are based on aggregating the classification judgments across three distinct annotators, using a coder sample that is nearly three times larger than our original study (N=6,392). When recollecting annotator judgments, we included an additional question which asked coders to indicate whether each face is a cartoon/digital avatar. We find that 8% of images were flagged as avatars, which suggests that fictionalized content is likely to contribute to our image results, but to an extent that is unlikely to be a primary driver of our findings. This latter point is substantiated by a supplementary analysis which uses a logistic regression to predict the likelihood of observing a male face per category search while controlling for whether faces are identified as cartoon avatars (please see table S8), and we refer to this robustness test on pg. 5 of the manuscript. We show that whether a face is an avatar is statistically unrelated to the likelihood of it being labeled as a particular gender, and that our main result holds while controlling for images' avatar classification, suggesting that statistical biases in gender representation in the domain of avatars are unlikely to be driving our results. We have chosen to keep all faces in our main dataset because we believe that both photographic and illustrated representations of people are important for understanding the gender associations of a particular category.

Controlling for the presence of stock photography is considerably more difficult, since it is not always apparent from the image alone that it derives from a stock marketplace, and classifying whether each image derived from a stock photography website would be an unfeasibly large task, given how many images constitute our main dataset. As a proxy, we hired a team of annotators to classify the websites associated with the images uploaded by participants in our experiment, since each participant was also asked to provide a link to the website from which they downloaded each description. The results of this analysis are presented in figure S33 in the final section of the appendix. Most images in our experimental data were coded as deriving from personal blogs. Stock photo websites were only identified as the fifth most common source of images, constituting less than 10% of images. This needs to be taken with a grain of salt, since it is possible for people to purchase and copy stock photo images and reuse them on their personal and professional websites. However, we think this is unlikely to bias our results in this context since less than 5% of images in our experimental data repeated across sources. The same is true in our observational data, where we find that only 11% of images repeat within or across searches, and all of our results are robust to the exclusion of repeat images. This analysis provides tentative support for the claim that image-based biases toward fictional representation, while present, are unlikely to serve as the primary mechanism underlying our main results, though this is a fascinating direction for future research as highlighted in our revised discussion.

10. I understand the focus in the discussion on the potential negative effects on the well-being and economic opportunities of women. But it may also be worth mentioning that the effects apply to occupational gender segregation at large. For example, these

algorithmic biases can also contribute to the low representation of men in care-oriented domains.

Thank you for this valuable suggestion. We have added the following note to our discussion to reflect this insight (from pg. 10 of the manuscript, in bold):

“In this study, we show not only that gender bias pervades the images of the world’s most widely used search engine, Google, but also that this bias is stronger than those in online texts, public opinion, and the US census. The same findings hold when comparing images and texts from Wikipedia (Fig. S3), in contrast with its perceived status as a neutral source of information. The scale of these findings is especially alarming in light of mounting evidence that images from these sources are not only widely used by people [26], but also by machine learning algorithms [27, 28], with the potential to amplify biases in each. Indeed, our experiments show that images are stronger than texts at amplifying explicit and implicit gender bias in people’s beliefs. **These findings draw attention to the potential for online images to negatively impact the well-being, social status, and economic opportunities for those who identify as women, with related implications for men in female-stereotyped categories, such as care-oriented occupations [6, 7, 23].** Thus, while recent work has reported improvements in the prevalence of gender bias in everyday language [1, 2], we must evaluate these narratives of progress with caution. The fight against inequality lags behind in the realm of images, which are ubiquitous and impactful.

Referee #4 (Remarks to the Author):

1. I am highly impressed by this huge undertaking, and I believe that it reports great studies with a good potential for high impact. I believe that the publication of this research would make a positive contribution to human knowledge. I believe that this paper would attract much interest from various disciplines and the general public. There were, however, some aspects in the manuscript that the authors might want to consider for a possible revision:

Thank you!

2. My expertise pertains only to the study of implicit social cognition, rather than biases in Internet media. I am unfamiliar with any study that examined the effect of Google search on self-reported gender stereotypes and performance in the IAT. This is a very nice experiment, with relatively high ecological validity.

There are a bunch of studies about the effect of exposure to information on the IAT, even specifically to the IAT that was designed to measure gender stereotypes (e.g., Asgari et al., 2012; Desgupta & Asgari, 2004; Ramsey & Sekaquaptewa, 2010; Stout et al., 2011). Some of those studies were longitudinal, showing an effect over time. Still, there is something unique in having participants search by themselves, rather than

providing them the information directly. The present experiment is a good demonstration of the possible effects of the biases reported by the authors on people's beliefs and stereotypes. In my field, this experiment would be highly interesting and impactful. However, the authors should probably acknowledge that the IAT is not a pure measure of (implicit) stereotypes, and it is sometimes sensitive to factors other than evaluations and beliefs. Critics of the IAT would be quick to argue that the present experiment found an effect of accessible information on IAT performance, which may, or may not, reflect a change in people's attitudes and beliefs

As Reviewer 4 astutely notes, there is ongoing debate around the interpretability and reliability of the IAT, which (like most psychological measures) faces limitations. This debate is crystallized in a recent pair of papers published in *Psychological Inquiry*: echoing studies referenced by Reviewer 4, Gawronski et al. (2022) argues that the IAT mostly detects explicit bias that is primed by the task environment; whereas Melnikoff & Kurdi (2022) directly reply to Gawronski et al. (2022) arguing that, when the IAT is reliable, it does indeed detect more implicit biases that are not captured by participants' self-reported explicit bias. Melnikoff & Kurdi (2022) argue that explicit self-report measures of bias (e.g., concerning gender stereotypes) are vulnerable to a social desirability confound, where participants will provide the answer they expect to be more socially appropriate and politically correct, while hiding their true beliefs. The IAT, while imperfect, does a better job of avoiding this social desirability confound, since it focuses on measuring participants' reaction times, and participants are unaware that reaction time is the behavioral signature being used to measure their bias, limiting their ability to game the measure by deliberately providing less biased responses. It is in this sense, according to Melnikoff & Kurdi (2022) and others, that the IAT is held to detect implicit biases that lurk beneath the aspects of participants' beliefs and attitudes that they may be self-aware of and thereby capable of strategically communicating.

To ensure that the IAT serves as a stable and interpretable measure, Melnikoff & Kurdi (2022) recommend leveraging validated IAT designs that have been shown to reliably detect a well-defined and well-recognized stereotype. In our study design, we deliberately followed this recommendation by using a specific IAT (focusing on the bias toward associating men with science and women with liberal arts), which has been shown to be among the most robust IATs in the literature; for instance, Charlesworth et al. (2022) annually deployed this exact same IAT on distinct samples for over a decade and observed striking patterns of stability and reliability in the implicit bias it detected. The stability of this IAT is attributed to the fact that the bias it detects matches a stark pattern of inequality in the distribution of genders across industry and academia (Miller et al. 2015; England et al. 2020; Master et al. 2021). Related work has also shown that the measure of implicit bias provided by this same IAT is predictive of a number of relevant behavioral outcomes, including people's actual math grades in undergraduate education (Smeding 2012). Using the same IAT as these prior studies allows our experimental design to avoid some of the problems of other IATs in the literature which are designed to detect new or less salient biases,

leading to unstable results. In particular, the fact that prior studies have replicated the specific IAT we use across a variety of task environments, and even in the absence of any priming interventions, suggests that it provides a robust measure of implicit bias that is unlikely to be entirely driven by explicit information in the task environment used in our experiment (e.g., by the mere fact that participants in our experiment were asked to rate categories in terms of their gender across all conditions, thereby priming them to think explicitly about their gender associations for occupations). Indeed, the dscore levels captured by our control condition are nearly identical to those measures in prior studies using the same IAT, despite considerable differences in the task environment across these studies.

Two additional features of our experiment demonstrate the robustness of the IAT in our setting: (1) in our prior pre-registered study on which our original submission was based (<https://osf.io/26kbr>), our IAT results cleanly replicated with a nearly identical but improved design; and (2) as shown in our revised experimental design, participants' IAT performance was not different between the Text and Control condition, suggesting that these conditions capture a stable baseline measure of implicit bias that is significantly amplified in the Image condition.

We think these experimental findings relative to the control condition help address the concern about whether “accessible” explicit information is driving participants' IAT responses. We show (in Fig. 3A) that participants in both the text and image condition were exposed to accessible information that was “gendered” and capable of priming their behavior in the IAT. However, we find that there was still no difference between participants' implicit bias in the text condition and the control condition, which lacked exposure to accessible gendered information. This reveals that the presence of accessible gendered information in the text condition was not sufficient on its own to skew participants' IAT responses relevant to a neutral baseline. This lends support to the claim that the IAT in both the text and control condition provided a reliable, though impure, measure of baseline levels of implicit bias unmediated by priming from explicit information. By contrast, it is only in the image condition that exposure to gendered information led to a notable change of participants' implicit bias beyond the control condition, which lasted for several days after the experiment. This finding is consistent with our claim that gender bias is not only more prevalent in online images than text, but also that images more strongly prime gender biases in people's associative cognition than text. This latter claim is further substantiated by the supplementary analyses provided for Fig. S32.

We have revised our manuscript to include these important clarifications.

From the main text on pg. 7:

“After completing the search task for all occupations, participants undertook an implicit association test (IAT) [22], a standard method in psychology for detecting implicit biases. We adopt an IAT designed to detect the bias toward associating women with liberal arts and men with science (Figs. S26-29), since prior work

demonstrates the robustness of this IAT [18] and its ability to predict judgments and behaviors [23] relating to a stark pattern of inequality in industry and academia [3, 6, 7]³.”

Footnote 3 in the manuscript:

“We acknowledge ongoing debate about the reliability of the IAT [24, 25]. While the IAT faces limitations like any psychological measure, prior studies demonstrate the reliability of the specific IAT we employ: the meta-analysis from Charlesworth et al. (2022) [18] confirm the reliability of this IAT over two decades of studies. Two additional features support the robustness of our IAT: (1) our results cleanly replicated in a separate pre-registered study with a nearly identical design (<https://osf.io/26kbr>); and (2) participants’ IAT performance did not differ between the text and control condition, suggesting that these conditions capture a stable baseline measure of implicit bias that is amplified in the image condition.”

Cited References:

- [3] Paula England, Andrew Levine, and Emma Mishel. Progress toward gender equality in the United States has slowed or stalled. *PNAS*, 117(13):6990–6997, 2020.
- [6] David I Miller, et al. Women’s representation in science predicts national gender-science stereotypes: Evidence from 66 nations. *Journal of Educational Psychology*, 107(3):631, 2015.
- [7] Allison Master, Andrew N Meltzoff, and Sapna Cheryan. Gender stereotypes about interests start early and cause gender disparities in computer science and engineering. *PNAS*, 118(48), 2021.
- [18] Tessa ES Charlesworth & Mahzarin R Banaji. Patterns of implicit and explicit stereotypes iii: Longterm change in gender stereotypes. *Social Psychological and Personality Science*.13(1), 2022.
- [23] Smeding, Annique. “Women in Science, Technology, Engineering, and Mathematics (STEM): An Investigation of Their Implicit Gender Stereotypes and Stereotypes’ Connectedness to Math Performance.” *Sex Roles* 67, no. 11 (December 1, 2012): 617–29.
- [24] Bertram Gawronski, Alison Ledgerwood, and Paul W Eastwick. Implicit bias ≠ bias on implicit measures. *Psychological inquiry*, 33(3):139–155, 2022.
- [25] David E Melnikoff and Benedek Kurdi. What implicit measures of bias can do. *Psychological Inquiry*, 33(3):185–192, 2022.

3. As for the main part that does not pertain to the IAT, I think the authors might want to emphasize more clearly what the novelty of this research is. It seems that there is already previous work that showed that Google images provides gender-biased results (Kay et al., 2015; Metaxa et al., 2021). Therefore, is the novelty of this research only the comparison between images and text? Or is it perhaps the magnitude of the present research (e.g., more categories)? This point was not clear to me.

Thank you for identifying this important need for clarification. In short, our comparison of images and text is the central novel contribution of our paper, and we have revised our title and introduction to more clearly convey this novel perspective and its importance. Our new title is: “Online Gender Bias is Stronger in Images than Text.” At the same time, we maintain that our paper makes a number of additional novel and important contributions. We summarize these novel contributions across four overarching themes, below:

(1) Comparing text to images (observational). The prior studies referenced by the Reviewers focus on gender bias either in images (e.g., Kay et al. 2015; Metaxa et al. 2021) or text alone (e.g., Caliskan et al., 2017). We can find no prior work that identifies differences between texts and images in how they encode gender bias. We address this gap by developing computational methods for comparing gender bias across images and text, including a novel experiment to compare the ability of images and text to reinforce gender bias in people's beliefs.

(2) Comparing text to images (experimental). Our experiment, which we have improved and replicated with a control condition, not only reveals that gender bias is more prevalent in online images than text, but also that images more strongly prime gender bias than online texts. This involves a novel integration of prior research on the "picture superiority effect" and prior work on priming effects in the study of gender bias. Neither Kay et al. (2015) nor all related studies we are familiar with identify this foundational difference between images and text and demonstrate its psychological consequences, which have far-reaching implications for image-based search engines and social media.

(3) Insights into the effects of images on implicit bias in an ecologically-valid experiment. Kay et al. (2015) and Metaxa et al. (2021) examine the ability for gender bias in online images to prime biases in people's explicit beliefs. We go beyond by measuring the impact of texts and images on both explicit and implicit bias over time. This required integrating the Implicit Association Test (IAT) into a novel online experiment in which participants used Google to search for content in real-time. These efforts reveal that only online images significantly and enduringly increased participants' implicit gender bias. See our reply to (4.) below for an explanation of the robustness of our IAT. Another novel feature of our experiment is that it required participants to leverage Google to search for information in a realistic manner, whereas Kay et al. (2015) and Metaxa et al. (2021) test the effects of exposing people to artificially created image sets without engaging them in a search task. This lends our design considerable ecological validity, which is highlighted by Reviewer 4 as another impactful contribution of our experiment.

(4) Scale. Kay et al. (2015) and Metaxa et al. (2021) only investigated approximately 5100 images mapped to 51 occupations. The dataset collected and analyzed in our study is almost 100 times larger. We study gender bias in the images associated with nearly 700 occupations, and 3,495 social categories in total, not only from Google images, but also Wikipedia; altogether, our sample consists of over 600,000 images. Furthermore, our revised *SOM* also tests our theory using a dataset of over 500k images from IMDb and Wikipedia. Combined, our entire image dataset comprises over one million images, and is mapped to the full known taxonomy of social categories in the English language, enabling a range of analyses infeasible for prior research.

We have considerably revised the introduction and discussion sections of our study to highlight these key dimensions of novelty, with a focus on the contrast between images and texts and its foundational implications for sociology,

psychology, and the study of search engines and social media. As a stylistic note, our discussion now links current debates in psychology and sociology to concerns expressed about the sociocultural impact of images on stereotypes and discrimination as expressed by the political philosopher Frederick Douglass in 1861. We have copied the comments illustrating this connection below from pg. 10 (most relevant sentences in bold), since we think they help to highlight the foundational nature of our contribution:

“In this study, we show not only that gender bias pervades the images of the world’s most widely used search engine, Google, but also that this bias is stronger than those in online texts, public opinion, and the US census. The same findings hold when comparing images and texts from Wikipedia (Fig. S3), in contrast with its perceived status as a neutral source of information. The scale of these findings is especially alarming in light of mounting evidence that images from these sources are not only widely used by people [26], but also by machine learning algorithms [27, 28], with the potential to amplify biases in each. Indeed, our experiments show that images are stronger than texts at amplifying explicit and implicit gender bias in people’s beliefs. These findings draw attention to the potential for online images to negatively impact the well-being, social status, and economic opportunities for those who identify as women, with related implications for men in female-stereotyped categories, such as care-oriented occupations [6, 7, 23]. Thus, while recent work has reported improvements in the prevalence of gender bias in everyday language [1, 2], we must evaluate these narratives of progress with caution. The fight against inequality lags behind in the realm of images, which are ubiquitous and impactful.

[...]

At the dawn of photography, political philosopher Frederick Douglass forewarned of the potential for images to either subvert or reinforce stereotypes at large, writing in his 1861 lecture “Pictures and Progress” that “the great cheapness and universality of pictures must exert a powerful though silent influence upon the ideas and sentiment of present and future generations” [30]. Nearly two centuries later, despite the rapid proliferation of images across popular media, the discriminatory impact of online pictures remains largely silent. We have aimed in this study to break this silence.”

4. In general, the methodology, data processing, and analyses seem highly professional and adequate with no serious red flags. However, some aspects were unclear to me, and might require further clarification. First, regarding the experiment, I did not see in the main text or in the supplement document analyses that tested directly the effect of the manipulation on the IAT. The closest I have seen to such a test is in Figure 3D, showing a relation between the measured value of the manipulated variable (Gender ratios in the search performed by the participant) and the IAT score. However, this still leaves a room for an effect of the participant’s stereotypes on the predictor (e.g., the participant’s choice of images to upload might be influenced by the participant’s gender

stereotypes). Therefore, it would be best to see analyses including the effect of the manipulation on the IAT score (especially because the analysis in the Figure 3D actually omits the participants from the text condition).

We agree with Reviewer 4 that this lack of clarity needs to be remedied. In our revised manuscript, we include the results of this test directly in Panel C of Figure 3. These results are provided in the context of a novel replication of our experiment, which we designed and pre-registered to address several of the Reviewers' concerns. This updated experimental design (pre-registration here <https://osf.io/3jhzx>) is the same as before, except that it includes a control condition and a modified text condition to address, respectively, comment 1 by Reviewer 3 and comment 5 by Reviewer 4. The IAT test of interest requested by Reviewer 4 is presented in the context of comparing both the image and text condition to the control condition (note all results hold regarding of the text condition used for our main comparisons, please see our response to comment 5 by Reviewer 4; our main text focuses on the comparisons with the Google News text condition because we believe this is the most appropriate comparison to maximize fit with our observational data). These results of this test are described in our manuscript on page 9, and are copied here for your direct review.

Fig. 3. Differential effects of googling for visual rather than textual descriptions of occupations on participants' explicit and implicit gender biases, as compared to a neutral control condition ($N=450$). (A) The average absolute strength of the gender associations that participants uploaded for each category in both the text and image condition. (B) The average absolute strength of the gender associations that participants reported for each occupation in each condition. (C) The average absolute strength of the implicit bias (dscore) that participants exhibited in each condition. (D) The correlation between the

average strength of participants' self-reported gender associations and their implicit bias toward associating men with science and women with liberal arts. (E) The correlation between the gender associations of the descriptions that participants uploaded and the gender associations they reported for each occupation, split by condition. M., Male; F., Female. (F) The correlation between the average strength of the gender associations of the descriptions that participants uploaded and average strength of the gender associations they reported for each occupation, split by condition. All error bands and error bars display 95% confidence intervals.

“Consistent with our observational results, panel A of Figure 3 shows that the descriptions participants uploaded were significantly more gendered in the image condition than the text condition ($p < .0001$, Wilcoxon Signed-rank Test, Two-sided; the gender associations for each occupation are highly correlated with those detected for the same occupations in our main observational analyses of Google Images and Google News, Fig. S30). Panel B of Figure 3 shows that exposure to more gendered stimuli in the image condition led participants to report significantly stronger gender associations as compared to participants in both the text ($p < .001$) and the control ($p < .001$) condition, while there was no significant difference between those in the text and control condition ($p = .56$) (Wilcoxon Signed-rank Test, Two-sided). These results hold when controlling for the number of online sources that participants encountered, and the amount of time they spent evaluating online descriptions (Fig. S31; Table S9).

Concerningly, Panel C of Figure 3 shows that exposure to more gendered descriptions in the image condition also led participants to exhibit significantly stronger implicit bias toward associating women with arts and men with science. There was no significant difference between participants' implicit bias in the text and control condition ($p = .19$), yet participants in the image condition exhibited significantly stronger implicit bias than those in the control condition ($p < .001$) (Wilcoxon Signed-rank Test, Two-sided). Indeed, panel D of Figure 3 identifies a robust correlation between the strength of participants' self-reported gender associations and the strength of their implicit bias, both of which are amplified in the image condition ($p < .0001$, Jonckheere-Terpsta Test). Together, these findings suggest that exposure to more gendered descriptions in the image condition more strongly primed participants' explicit and implicit gender associations.”

5. One issue that confused me throughout the reading is that most of the variables pertained to gender-bias in absolute terms – how different the score is from a score that reflects a 50-50 presentation (or association) of the social category with male and female. In fact, this was always the case excluding the IAT score (see the range of the y-axis in Figure 3D). That means that the graph in Figure 3D reflected a correlation between general gender-imbalance in Google image search and a specific bias (Men with science, women with Liberal arts) in the IAT. The use of absolute score was surprising because it ignores the question of whether the imbalance in the gender rate reflects the known gender stereotypes. It was not explained in the main text or the supplement why the authors preferred to look at such scores instead of trying to quantify how much their scores reflect the traditional stereotypes. I can think of good

justifications for that, and I suspect that this was probably the correct decision, but it was quite confusing, so it requires an explanation.

This is a very helpful insight. We focus on general scores because not all categories can be mapped to well-defined empirical distributions, in part because of differences in the level of abstraction at which these categories operate; for example, every human is a member of the category “person”, and many adults are a member of the category “professional” which has blurry boundaries. That said, Reviewer 4’s comment motivated us to add a supplementary analysis that examines our results while benchmarking qualifying categories against their underlying empirical distributions. Specifically, we compare the gender associations of occupations in Google Images and Google news while centering these associations on the “true” gender distribution associated with each occupation in the U.S. census. We copy this analysis here (starting on pg. 42 of the *SOM*):

“Another potential concern is whether the gender dimension we rely on, which is centered on gender neutrality (at 0), is an appropriate method for evaluating biases in the amplification of gender associations, since a measure of amplification could in theory account for what we know about the existing empirical distribution of gender within a category. In other words, we could compare the gender associations produced by texts and images after centering them on the true gender distribution for each category according to empirical data such as the US census. This is the analysis we conduct below.

For this analysis, we begin by calculating the same gender dimension for the Census data (which contained 687 occupational categories that matched our text and image datasets). Then, for each category, we treat the census value along the gender dimension as “0” and then center the gender associations in text and image on this census value. So, for example, if the census data positions *plumber* as a 0.4 along the gender dimension, we then set this value as the center point in the distribution against which we compare the position of *plumber* based on its gender association in image and text data.

Figure S18: The distribution of gender associations in images from Google Images and texts from Google News for 687 occupations. For each occupation, we set the value of 0 to the fraction of males that belong to each occupation, according to the US census bureau. To center the distributions accordingly, we first calculate the gender associations according to our standard measure for each data source; we apply the same procedure to the gender associations according to the census, and then for each category, we subtract the census' gender balance from each data source, thereby centering them on the census' representation. Positive values thereby indicate that a data source presents greater male representation for a given occupation than the census, and negative values present greater female representation. Solid green (purple) line indicates the average gender association according to text (images).

The results of this analysis are shown in figure S18. We find that the textual measures are significantly more female relative to the census (mean=-0.08, $p<.0001$), whereas the image measures are significantly more male relative to the census (mean=0.05, $p<.0001$), (Wilcoxon Signed-Rank Test). The census-centered text and image values are highly significantly different from each other ($p<.0001$)."

6. *The overlap between the pre-registration (<https://osf.io/26kbr>) and the reported study does not seem ideal. For example, the pre-registration mentioned using a set of "over 200 occupations" but the main text reported using 54 occupations. More importantly, the pre-registration did not include most of the tests reported in the main text and the supplement, and referred mostly to the comparison between the text and the image search. And, as I mentioned earlier, although the pre-registration includes the analysis of the condition (text vs. image) on the IAT scores, I did not see that test reported in the main text or the supplement.*

We appreciate this request for further clarification on the fit between our pre-registration and the narrative presentation of our experimental results. The reason the number of occupations differed between the pre-registration and the final implementation was because we had yet to identify the exact IAT we intended to use

for the experimental effects when we pre-registered this study design. We anticipated that we would have to refine the occupations used as stimuli to maximize their relevance to the implicit bias measured by the IAT we would eventually implement. This is what we intended to communicate by the following sentence in our original pre-registration, under the “hypotheses” section: “To optimize our search task design to match this IAT, we plan to focus subjects toward searching occupational categories in the domains of science and art.” We agree with the Reviewer that this ambiguity needs to be further addressed.

In light of the additional requests by the Reviewers, we reasoned that the most elegant approach to resolving this issue would be to pre-register a direct replication of our experiment, where the precise details of the experimental design are pre-registered without ambiguity. This approach also allowed us to incorporate a control condition and a modified text condition to address, respectively, comment 1 by Reviewer 3 and comment 5 by Reviewer 4. This updated pre-registration can be found here (<https://osf.io/3jhzx>). You will see that we provide precise details regarding the number of subjects and categories these subjects were asked to search for and evaluate (along with the specific categories used in this task). This includes a list of the categories that subjects evaluated across all conditions, as well as the list of random distractor categories that participants were asked to search for in the control condition.

Regarding Reviewer 3’s second concern regarding our pre-registered (text vs. image) comparison on the IAT outcome scores, we agree that our initial presentation of the results did not clearly present this test and its findings. In our revised manuscript, we include the results of this test directly in Panel C of Figure 3. We have copied this revised figure and its results description in our response to comment 1 by Reviewer 3. The findings show that this pre-registered comparison is statistically significant and in the direction predicted by our theory.

7. I wonder whether the textual condition in the experiment was the textual equivalent of the image condition. The textual condition seemed less constrained – it was less clear what the participant should do, in comparison to the image condition. For example, I guess that I would have added the word “define” in the textual Google search, and would have just pasted the definition. Or, perhaps I would have added the word “Wikipedia” and would entered the definition at the beginning of the Wikipedia entry. In both cases, it is not clear whether it is the same as searching in the Google image, which is more clearly related to the question “how does a geneticist look” or “what is a good example for a geneticist” rather than what a geneticist is (for the latter, people would probably not use Google images). Perhaps a more comparable search would have been for the category in Google News.

This is another excellent suggestion. For reasons elaborated above, we pre-registered a direct replication of our experiment that included a modified version of the text condition where participants were tasked with using Google News specifically (news.google.com) to search for and upload textual descriptions of occupations. In fact, we have chosen to focus our results in the manuscript on the comparison

between the control, image, and Google News version of the text condition because we believe this is the most appropriate text condition to compare against (in light of its fit with our observational contrast between Google News and Google Images). On page 58 of the *SOM*, we provide robustness tests which indicate that there are no significant differences between the Google News text condition and our original text condition in terms of the main behavioral outcome of interest. We have provided these supplementary analyses below for your review.

“A.3.3 Robustness to Participants Searching in Google Instead of Google News

In our main experiment, participants in the text condition were guided to the Google News search bar and were asked to identify and upload a description of an occupation from this context. This design choice was made to maximize the similarity of the experimental context to our observational comparisons between Google Images and Google News. However, participants in our experiment (N=150) were also randomized to a different version of the text condition, in which they were guided to the general Google search engine (i.e., simply google.com) and asked to perform the same task. Thus, in total, our experiment included 600 unique participants.

Figure S25: Differential effects of googling for images rather than textual descriptions of occupations on participants' explicit and implicit gender biases, as compared to a neutral control condition, while including a version of the text condition in which participants were directed to search for textual descriptions in the general Google search engine, rather than in Google News specifically (N = 600); the data shown for all other conditions are the same as those presented in our main results (see Fig. 3 in the main text). (A) The average absolute strength of the gender associations that participants reported for each occupation in each condition. (B) The average absolute strength of the implicit bias (dscore) that participants exhibited in each condition. The results are shown in solid purple (image condition), solid green (Google News text condition), dotted black (control condition), and dotted green (General Google text condition).

Building on the presentation of our main experimental results, Figure S25 compares the outcomes of each condition to an alternative version of the text condition in which participants were directed to retrieve textual descriptions of each occupation by searching via the general Google search engine (referred to as the “text neutral” condition) rather than Google News specifically. Panel A of Figure S25 shows that there was no significant difference between the neutral text condition and the Google News text and control condition in terms of the strength of participants’ self-reported gender associations for each occupation. Yet, again, we find that participants in the image condition produced significantly stronger gender associations compared to participants in the neutral text condition ($p < .001$, Wilcoxon Signed Rank Test, Two-tailed), along with all other conditions (as discussed in the main text). Panel B of Figure S25 shows that this result replicates when analyzing the overall strength of participants’ implicit bias toward associated men with science and women with liberal arts; that is, we find no significant difference between participants’ implicit bias in the text neutral and control condition, nor between the text neutral condition and main text condition; yet, participants in the image condition exhibited significantly stronger implicit biases than those in the text neutral condition ($p < .001$, Wilcoxon Rank Sum Test, Two-tailed). These results show that our main findings are not an artifact of comparing against Google News search results in particular, but instead generalize to a different means of acquiring textual descriptions of social categories via the Google search engine.”

8. I assume that it is not straightforward to determine whether the measurement of bias in text vs. image were easily comparable. I tried to think of methods to validate the similarity of these two measures using associations that are expected to be the same in both databases, but I could not think of such associations. So, I am raising this issue here, just in case the authors would have an idea.

We are grateful for this suggestion, which spurred us to include an additional analysis strategy that better allows for an apples-to-apples comparison of gender associations between images and text that does not rely on word embedding models. To maximize comparability between these modalities, we developed a simple technique that counts the frequency of gendered descriptions of occupations in both images and text, using an identical counting procedure. In what follows, we show not only that our main results replicate cleanly using this alternative measure, but also that this alternative measure is highly predictive of the gender associations detected by our original measures of gender bias in Google images and a word embedding model of Google News texts.

Alternative method for comparing gender bias across online images and texts. In our experiment, participants were asked to upload either textual or visual descriptions of each occupation. We hired a team of annotators to code whether each textual or visual description was female (1), neutral (0), or male (1). Each textual description was coded as either male, female, or neutral based on whether it used male or female pronouns or names to describe the occupation (e.g., referred to a

“doctor” as “he”); textual descriptions were identified as neutral if they did not ascribe a particular gender to the occupation described. Similarly, the gender of each image description was coded by coding the gender of the focal face in each image; faces were coded as neutral if they contained no face or an indecipherable face. Then for each occupation in each experimental condition, we calculate the average gender association across the descriptions provided by separate participants. Since the number of descriptions per occupation per condition are constant, and the outcome measure of gender association is similarly constant (such that each description is associated with a single gender label), this provides a more direct apples-to-apples comparison of the strength of gender associations between the textual and image modalities.

Here, we copy this revised analysis in the main text, along with a supplementary section which shows that the measures of gender associations provided by this alternative method highly correlate with our original measures.

Revisions to pg. 8 in the main text (most relevant changes in bold):

Figure 3: Differential effects of using Google to search for images rather than textual descriptions of occupations on participants' explicit and implicit gender biases, as compared to a control condition (N=450). The green (purple) vertical lines indicate average effects for the text (image) condition. (A) The average absolute strength of the gender associations in participants' uploads for each category in both the text and image condition. (B) The average absolute strength of the gender associations that participants reported for each occupation in each condition. (C) The average absolute strength of the implicit bias

(dscore) that participants exhibited in each condition. (D) The correlation between the average strength of participants' self-reported gender associations and their implicit bias toward associating women with liberal arts and men with science. (E) The correlation between the gender associations of the descriptions that participants uploaded and the gender associations they reported for each occupation, colored by condition. (F) The correlation between the average strength of the gender associations of the descriptions that participants uploaded and average strength of the gender associations they reported for each occupation, colored by condition. ***, $p < .001$. All error bands and error bars display 95% confidence intervals.

“Consistent with our observational results, panel A of Figure 3 shows that the descriptions participants uploaded were significantly more gendered in the image condition than the text condition ($p < .0001$, Wilcoxon Signed-rank Test, Two-sided; the gender associations for each occupation are highly correlated with those detected for the same occupations in our main observational analyses of Google Images and Google News, Fig. S30). Panel B of Figure 3 shows that exposure to more gendered stimuli in the image condition led participants to report significantly stronger gender associations as compared to participants in both the text ($p < .001$) and the control ($p < .001$) condition, while there was no significant difference between those in the text and control condition ($p = .56$) (Wilcoxon Signed-rank Test, Two-sided). These results hold when controlling for the number of online sources that participants encountered, and the amount of time they spent evaluating online descriptions (Fig. S31; Table S9).

[...]

This priming mechanism is further supported by panel E of Figure 3, which shows a strong correlation between the gender associations in the descriptions that participants uploaded and the gender associations in their own explicit responses across occupations ($\rho = .8$, $p < .0001$); as well as by panel F of Figure 3, which shows a strong correlation between the overall strength of gender associations in the descriptions that participants uploaded and the overall strength of the average gender associations they reported across occupations ($\rho = .57$, $p < .0001$). These results hold for both the image and text condition, and across occupations (Table S10).”

Corresponding revisions in the SOM (starting on pg. 67):

“A.4.1 Using our Observational Measures of Gender Bias to Predict Gender Bias in Participants' Uploaded Descriptions and Self-report Beliefs.

Here we confirm that our observational measures of gender bias in images from Google Images and texts from Google News are predictive of gender bias in the image/textual descriptions that participants uploaded in our experiment, as well as in their self-reported gender associations with each occupation. This analysis is important not only for demonstrating the ability for our observational measures to predict ecologically valid patterns of content exposure and response bias among human participants using google, but also to confirm the correspondence between

our alternative methods of measuring gender bias in text, namely the word embedding technique and the coarser method of counting the frequency of gendered descriptions in our experimental data.

Figure S30: The relationship between the distribution of gender in our observational sample of the top 100 Google images per occupation and (A) the distribution of gender associations in the images from Google uploaded by participants, and (B) participants' self-reported gender associations for each occupation, which they provided after uploading an image from Google. The relationship between the distribution of gender in our observational word embedding measures of Google News and (C) the distribution of gender associations in the textual descriptions from Google News uploaded by participants, and (D) participants' self-reported gender associations for each occupation, which they provided after uploading a textual description from Google News. All results are averaged at the occupation level, such that 58 data points (occupations) are shown. Error bands show 95% confidence intervals.

The results of this analysis are presented in Figure S30. Panel A of Figure S30 shows that the distribution of gender in our observational sample of the top 100 Google images per occupation is highly correlated with the distribution of gender in the images uploaded by participants in our experiment ($r = 0.58$, $p < .00001$, $N=58$ occupations). Similarly, panel B of Figure S30 shows that the distribution of gender in our observational sample of the top 100 Google images per occupation is highly correlated with participants' self-reported gender associations for each occupation, which they provided after uploading an image from Google ($r = 0.76$, $p < .00001$, $N=58$ occupations). The same results hold for the text condition. Panel C of Figure S30 shows that the distribution of gender in our observational word embedding measures of Google News is highly correlated with the distribution of gender in the textual descriptions uploaded by participants in our experiment ($r = 0.54$, $p < .00001$, $N=58$ occupations). This finding is particularly striking not only because gender bias is measured differently by our word embedding model and experimental analysis, but also because the Google News data underlying the word embedding model was collected in 2013, whereas our experimental data involves descriptions from Google News downloaded in 2022. Lastly, panel D of Figure S30 shows that the distribution of gender in our observational word embedding measures of Google News is highly correlated with participants' self-reported gender associations for each occupation, which they provided after uploading a textual description from Google News ($r = 0.79$, $p < .00001$, $N=58$ occupations)."

Reviewer Reports on the First Revision:

Referees' comments:

Referee #1 (Remarks to the Author):

The paper discusses the gender bias observed in Google Image Search results for 2,986 social categories compared to the gender associations of 3,495 words related to social categories in word embeddings trained on a Google News corpus. The study also considers public opinion and US census data for comparison. Another notable finding is that exposure to gender biased image results has a stronger impact on increasing gender bias Implicit Association Test (IAT) scores in the female+arts/male+science test compared to experiments that use biased text.

I appreciate the additional analysis the authors provided. However, concerns regarding “the potential differences in the text versus image data that could be biasing the results” have not been sufficiently addressed. This reviewer is not convinced that the comparison in the main experiment is valid, as detailed in the review.

Concerns regarding “limitations with the IAT setting and the strength of the conclusions that can be drawn from the IAT” have not been sufficiently addressed as detailed in the review.

“Training new word embedding models at the appropriate scale for our analysis takes an enormous amount of computing power, which is why social scientists and much of the machine learning community standardly harness pre-trained models released by companies such as Google, Facebook, and OpenAI...” Training static word embeddings is not computationally too expensive compared to training language models that have contextualized word embeddings.

The experimental design of the study involves several moving components, and it is unclear how the findings and metrics from different data sources can be directly compared. Language and vision modalities possess unique characteristics, and the validity of comparing gender bias in word embeddings, which incorporate contextual co-occurrences from a large corpus, to bias in Google Image Search results obtained from diverse countries remains unclear. In terms of the IAT experiments, it is important to know the amount of text participants read and wrote, and how this corresponds to the amount of information conveyed by each image retrieved from Google. Additionally, the paper lacks information on what IAT scores participants had for image and text-based stimuli before the experiments, as well as the influence of existing associations on the first and second IAT measurements. Furthermore, the relationship between the 2,986 words and the female+arts/male+science IAT requires justification.

The title of the paper should be more precise. The authors acknowledge that while text can minimize bias through gender-neutral terminology, online images often explicitly depict people, making demographic information inevitably conveyed. Given the distinct nature of signals in vision and language, further justification is necessary to support the current comparison of bias measurements in word embeddings and Google Image Search results. Word embeddings are trained on a corpus, while Google Image Search results come from a dynamic and context-dependent black-box, leading

to uncontrolled experimental settings that affect the robustness and reliability of the study.

The claim that prior studies on online gender bias primarily focus on text is challenged by the significant contributions of researchers like Abeba Birhane, Aylin Caliskan, and Olga Russakovsky, among many others, who have extensively analyzed bias and gender bias in image datasets or vision models trained on vast amounts of image or text-image pairs in multimodal settings. Similarly, Caliskan et al. have measured the gender associations of millions of words, highlighting the scalability of their approach.

In summary, the paper addresses an important issue, but it lacks clarity on how the current methodology can yield robust, valid, and reliable answers.

Referee #2 (Remarks to the Author):

Review to authors

1. I congratulate the authors on the extensive revision of this paper/ They seemed to have gone above and beyond to improve the paper and to address the reviewers' concerns (as far as I can see). I appreciate the long list of robustness analyses that they have added to the paper and the structural revisions that have been done. My prior concerns that led me to not yet fully trust the results and associated claims, have in most but not all cases resolved in this revision. I'll mention my prior points in the same order as before, though the authors have done a great job revising the paper such that my later points (6-11) all seem well addressed.

2. I'll start with my biggest concern. What I write here is ultimately up to the editors, and how they view the contribution of the paper. My prior concern was that the paper may overclaim its finding based on something that is inherently evident from the outset: namely that images carry more bias purely by the nature of the medium and that individual level effects of text versus images — i.e., images carrying more weight in the influence process — may also not be super surprising. I appreciate the author's response on this concern that goes into technical detail on how they rendered the two types media comparable. Yet, the inherent *theoretical* difference between images and texts and its influence on respondent remains. How surprising is the finding? Is it the scope in which this has been done in this study? The nuanced methodological approach? Or is there something substantively new we learn about human gender attitudes and how that is influenced? I see that this study addresses the first two questions *really* well, but the way in which it deals with the (perhaps more fundamental) second question does remain underdeveloped.

3. The authors note the differences in text and images and how they prime gender bias. My point 2 above doubts whether that is a substantively interesting finding. I do appreciate the labeling of the sources of the data. It may essentially be generated by gendered usage levels of tech and particularly blogs and the types of photos that are uploaded there.

4. The authors sufficiently address my prior point 4, yet a new concern arose when reading. Would participants homophilously select photos based their own gender? Perhaps I missed it, but that

seems like an important control. That relates to entrenched gender power dynamics in society writ large and how effects of gendered imagery may be conditional on participants' own gender.

5. The authors have done a decent job addressing this concern, though I still think the front end needs work. Statements such as "Images may this provide a clearer window into large-scale patterns of gender bias today." are challenging to test and seem somewhat underdeveloped from micro-level influence mechanism differences between images and text. This relates to my point 2: they do not provide a clearer window, just a substantively different window. Similarly, a statement that there is a "puzzling disconnect" between textual representation of gender online and gender disparity in society seems like an overclaim. Women politicians, women public figures, or even women regular users on Twitter would disagree given the amount of textual abuse they likely receive compared to their men counterparts. Do the authors need to phrase their premise as a paradox that itself is somewhat disconnected from what happens in the online sphere? In other words, the front end and setup of the paper's questions (as I also mention in point 2) can still use some improvement and development.

6-11. My prior points 6 through 11 all seem very sufficiently addressed. I want to emphasise the amount of work the authors have done between the prior version and this version of the manuscript.

Referee #3 (Remarks to the Author):

The authors' revisions successfully address the majority of the points I raised in my previous review, and I only have a few remaining comments in the spirit of improving the final version as much as possible and maximizing its impact.

1. It's great that the authors found a way to estimate the inter-rater reliability of the MTurk coders. However, percent agreement is a crude and somewhat poor metric of inter-rater reliability which does not correct for chance agreement, and is therefore often considered insufficient (Lombard et al., 2002; Neuendorf, 2002). Chance-corrected agreement is more appropriate because it makes the assumption that some portion of coders' agreement is due to chance and adjusts for that. It would be highly desirable for the authors to complement the simple agreement statistics with chance-corrected indicators such as Scott's pi, Cohen's kappa, Krippendorff's alpha, or (ideally) Gwet's AC. These different indicators vary in their computation, assumptions, and stability, and Gwet's AC in particular is highly recommended (e.g., Heyman et al., 2014), as it avoids the instability associated with the other coefficients.

2. I don't necessarily agree that there is a big problem of competing definitions around the term "stereotype", although different authors may emphasize some aspects and not others (e.g., focusing on "traits and behaviors" versus "social roles"). However, I see how replacing "stereotypes" with "gender bias" helps address the issue of conceptually defining stereotypes. But the current version of the manuscript still mentions "stereotypes" or derived words ("stereotypical") (e.g., twice on p. 10, once on p. 11). I would recommend either removing the term from the paper entirely or providing a definition for it.

3. I like the revisions the authors made related to my previous comment on an exclusive focus on the underrepresentation of women (instead of focusing on gender bias more broadly, which would involve paying similar attention to instances of the underrepresentation of men, e.g., in female-stereotypical occupations and roles). But I think more re-writing could be done in this area for the paper to be most impactful. There is a growing literature calling attention to the stark gender imbalance in female-dominated occupations, which tends to be more extreme than in male-dominated occupations, but which garners considerably less attention from researchers and the public (see for example Block et al., 2018, Croft et al., 2015, Croft et al., 2021). For example, on p. 5, the first full paragraph deals exclusively with the representation of women, but it omits the fact that, when searching for female-dominated occupations, the pattern of gender bias in images would likely flip. As Figure 2A shows, although the purple area on the right tail end is larger than the purple area on the left tail, both extremes show gender bias and it would be best if this parallelism were more strongly emphasized throughout the main text.

References:

- Block, K., Croft, A., De Souza, L., & Schmader, T. (2019). Do people care if men don't care about caring? The asymmetry in support for changing gender roles. *Journal of Experimental Social Psychology*, 83, 112-131.
- Croft, A., Schmader, T., & Block, K. (2015). An underexamined inequality: Cultural and psychological barriers to men's engagement with communal roles. *Personality and Social Psychology Review*, 19(4), 343-370.
- Croft, A., Atkinson, C., Sandstrom, G., Orbell, S., & Akin, L. (2021). Loosening the GRIP (Gender Roles Inhibiting Prosociality) to promote gender equality. *Personality and Social Psychology Review*, 25(1), 66-92.
- Heyman, R. E., Lorber, M. F., Eddy, J. M., & West, T. V. (2014). Behavioral observation and coding. In H. T. Reis & C. M. Judd (Eds.), *Handbook of research methods in social and personality psychology* (pp. 345–372). Cambridge University Press.
- Lombard, M., Snyder-Duch, J., & Bracken, C. C. (2002). Content analysis in mass communication: Assessment and reporting of intercoder reliability. *Human communication research*, 28(4), 587-604.
- Neuendorf, K. A. (2002). *The content analysis guidebook*. Thousand Oaks, CA.

Referee #4 (Remarks to the Author):

Signed: Yoav Bar Anan

I continue to be highly impressed by this work. It is clearly an outstanding undertaking, with very interesting results. It has a good potential to captivate a wide range of audiences. Naturally, I have a

few remaining comments that the authors might like to consider, in the case of further revision, although I consider only the first of them critical.

1. The critical issue pertains to the results of the experiment, reported in the final paragraph of Page 8. The authors report that “There was no significant difference between participants’ implicit bias in the text and control condition ($p = .19$), yet participants in the image condition exhibited significantly stronger implicit bias than those in the control condition ($p < .001$)”. Isn’t the comparison between text and images the critical comparison? The pre-registration was rather loose, which is a (non-critical) weakness in this work, but it did indicate: “We will use paired, non-parametric between-subject tests (e.g. the Wilcoxon test) to evaluate whether the strength of the gender associations (and implicit stereotypes) reported in the image condition are stronger than those reported in the text condition and the control condition.” These comparisons were reported for the self-report, but not for the IAT. To be clear, this is critical to fix, but I do not consider the actual result critical. In my view, in the present case, the IAT is not much different than the self-report: there were statistical associations between gender and occupation in the content collected by the participants, and those associations were registered in people’s memory, and therefore reported directly by the participants and influenced performance in the IAT. Therefore, the self-report results are interesting and important enough, even if the IAT results were not always clear-cut. Still, it would be important to report the same comparisons for both direct and indirect measures, preferably the pre-registered comparisons.

2. The best possible contribution of this research to science, beyond any specific finding, is the vast amount of data collected, processed, and analyzed. One can teach a whole university course about this data, and even only about how the authors analyzed it. Further, it is obvious that researchers could conduct tons of secondary research with these datasets (including the experiment, which was an unusual undertaking). I strongly recommend, implore and encourage the authors to provide immediate access to any data and code that they can share openly, and a less immediate access (e.g., through request forms) to data that may not be shared freely online (e.g., due to copyright restrictions). I believe that dozens of research studies could make use of that data. Notably, many researchers would be interested in the data and materials of the experiment. There are a few particularly strong aspects in that experiment, in comparison to the typical experiment: the manipulation was quite elaborate and strong (yet also ecological valid), there was more than one control condition, gender bias was measured again a few days after the induction. Data from such a strong experiment would be of interest to many, so, hopefully, it will be shared openly.

3. I think that Figure 1B was an important addition. The issue that bothered me the most in my previous review was that it was not easy to make sense of the two different biases reported by the authors: one was that the absolute gender bias was stronger in the image condition than in the text condition, and the other was that males were over-represented (i.e., were associated with more categories). Over-representation of males could have reflected gender biases that are quite different than those that are typically the focus of political and scientific attention. For example, perhaps “nurse” was more strongly related to Male than to Female in images than in text. That would have been a very different finding than what Figure 1B suggests.

Figure 1B is a big step toward explaining that the over-representation of men in image content is not the reason (or not the whole reason) for the larger gender bias in image content. However, there might be a concern about that plot, related to how categories were chosen to each separate bar in that plot. If I understand correctly, the Male Text categories were selected as those that showed a stronger association with women in the texts, the Male Images categories were selected as those that showed a stronger association with men in the image content, and the same is true for the Female bars. It means that the categories included in the data of each bar, were probably different. For example, perhaps “nurse” was included in the “male” white bar (the second from the top) and in the “female” grey bar (the third from the top). In that case, although Images content showed a different bias than the stereotype, the plot would (wrongly) seem to suggest that gender stereotypes were reflected more strongly in Images content than textual content. It might be helpful to show the same plot in the supplemental materials four times: once with the categories separated (for both text and image associations) by the image associations (i.e., only the “image” association would determine whether a category is “female” or “male”), once separated by the text associations, once by Human Judgments, and once by Census. Separating the categories by Human Judgment would allow seeing whether Images content amplifies gender stereotypes, in comparison to text content. Separating the categories by the Census data would allow seeing whether Images content is biased, in comparison to the gender-category associations in reality. That is an important piece of the results that is still eluding me.

4. Still related to the benefit of Figure 1B, it might be informative to see a version of Figure 2C that is separated to “female” and “male” categories. Currently, it seems like the Text associations might be more biased than the Images associations because they seem farther from the actual gender distribution (the Census data). This might reflect a larger bias in feminine categories. For example, perhaps “nurse” is more female-biased in Text than in Census and Images. That may reflect a stronger problem in Text than Images. If you break those bars to “female” and “male” categories (e.g., by Human Judgments), we might learn more about the pattern of the results. Probably, Text would be the least gender biased for both Male and Female categories, in comparison to Census and Images, reaffirming the authors’ current conclusions from this plot.

5. In p. 10, the authors wrote “we find that images differ from text not only in the prevalence of gender bias they contain, but also in their ability to prime gender bias in people’s beliefs. Participants who uploaded gendered images reported significantly stronger gender bias compared to participants who uploaded gendered textual descriptions of the same occupations ($p < .0001$, Student T-test, Two-tailed; Fig. S32).” It is not clear whether that effect is due to a larger impact for imagery (or something else inherent in images vs. text) or simply because even across those categories, the image content was more gender biased than the textual content. A possibly more appropriate analysis for the former claim would control for the content’s bias and test whether an effect of the content type (image vs. text) remains.

6. Figure 1C was described as “The gender associations for a sample of occupations according to these online images and texts.” This is of little informative quality without knowing how those

occupations were sampled. With thousands of occupations one can probably even find a sample that would show the opposite results of those found across the whole sample.

7. I think that Figure 1C is missing a legend.

8. In Figure 2B, I was not sure what the x-axis numbers meant. Why did they run from 0 to 25? Was it arbitrary (e.g., bin number) or meaningful?

9. Note that it is probably unusual to spell the IAT's D score "dscore".

Author Rebuttals to First Revision:

Reply to Reviewer 1 (R1)

1. I appreciate the additional analysis the authors provided. However, concerns regarding “the potential differences in the text versus image data that could be biasing the results” have not been sufficiently addressed. This reviewer is not convinced that the comparison in the main experiment is valid, as detailed in the review.

We are grateful for the helpful comments and recommendations provided by R1. Our responses detail a number of analyses which demonstrate that our results are not an artifact of statistically confounding differences between our text and image data, nor between our methods for analyzing our text and image data. We hope that these additional robustness tests and clarifications will satisfy the remaining concerns of R1.

2. Concerns regarding “limitations with the IAT setting and the strength of the conclusions that can be drawn from the IAT” have not been sufficiently addressed as detailed in the review.

Based on the comments from the prior review cycle, we added footnote 3 to our manuscript which discusses the known limitations of the IAT. To clarify, our main experimental results do not depend on the IAT. The image condition in our experiment increased bias in participants’ self-report gender associations which they deliberately (i.e., non-implicitly) indicated using a slider, prior to completing the IAT. We would like to share R4’s interpretation of these findings since it captures the relevance of our results beyond the IAT (see comment 1 from R4, who is an expert in implicit bias and the IAT): “In my view, in the present case, the IAT is not much different than the self-report: there were statistical associations between gender and occupation in the content collected by the participants, and those associations were registered in people’s memory, and therefore reported directly by the participants and influenced performance in the IAT. Therefore, the self-report results are interesting and important enough, even if the IAT results were not always clear-cut.” To align with this, we have revised our introductory paragraph so that it does not focus solely on implicit bias as the exclusive outcome variable of interest in our experiment, and instead is inclusive of the experimental effects on participants’ self-reported gender associations.

3. “Training new word embedding models at the appropriate scale for our analysis takes an enormous amount of computing power, which is why social scientists and much of the machine learning community

*standardly harness pre-trained models released by companies such as Google, Facebook, and OpenAI...”
Training static word embeddings is not computationally too expensive compared to training language
models that have contextualized word embeddings.*

We are grateful for R1’s continued emphasis on this point, since it has spurred us to train our own word2vec model on a large sample of online news published between 2021 and 2023. The gender associations of social categories are strikingly consistent across the 2013 Google News word2vec model and our own word2vec model retrained on recent online news. All of our main results strongly replicate when comparing gender associations in our Google Image sample to those in our retrained word2vec model. These analyses address the concern about whether our results are an artifact of the temporal distance between the collection of textual data for the original Google News word2vec model and the collection of our Google Image data. This robustness test helps rule out this concern, as all our results remain consistent when examining the same word2vec model trained on online news text data from the same time period as our Google Image dataset collection. For reference, we copy this newly added supplementary analysis below.

“Robustness to Comparing Against a Word2vec Model Trained on a Recent Sample of Online News.

Since the canonical word2vec model we analyze in the main text was trained on Google News data from 2013, we trained our own word2vec model. On a more recent sample of online news data published between 2021 and 2023. We compiled a dataset of 2,717,000 randomly sampled news articles published in English across various topics between January 2021 and August 2023. These articles were sourced from the following prominent online news sources: 1,000,000 articles from the BBC; 500,000 articles from the Huffington Post; 480,000 articles from CNBC; 400,000 articles from Bloomberg; 160,000 articles from Time Magazine; 150,000 articles from Techcrunch; and 27,000 articles from CNN. These datasets were purchased from the online web-scraping service, Crawl Feeds. We trained our own word2vec model on this dataset of online news using the exact specifications of the original word2vec model trained on the Google News dataset. Specifically, our word2vec model used 300 dimensions with skip-gram training based on a window size of 10. We then apply the same method we applied to the Google News word2vec model for identifying the gender associations of all social categories in Wordnet. The aim of comparing against this retrained word2vec model is to show that our main results are not an artifact of the time difference between the textual data collection for training the original Google News word2vec model (2013) and the timeframe during which we collected our Google Image data (2021).

Fig. S7. (A) The correlation between the gender associations of all social categories in WordNet according to the 2013 Google News word2vec model and our word2vec model retrained using online news data from 2021 to 2023. The trend line reflects the linear correlation between these models' gender associations, with error bands displaying 95% confidence intervals. Each data point corresponds to a distinct category. 2,992 social categories could be matched across these models. (B) The strength of gender associations across these same social categories according to each word2vec model as compared to our sample of Google Images collected in 2021.

First, we show that the gender associations of the social categories in Wordnet are remarkably consistent across the 2013 Google News word2vec model and our own word2vec model, which was retrained on a sample of online news from 2021 to 2023. Panel A of Figure S7 shows that the Pearson correlation in gender association across these models, paired at the category level, is 0.8 and highly statistically significant ($p < .00001$). Accordingly, the gender associations in the 2013 Google News word2vec model and our 2023 retrained word2vec model are both highly correlated with the gender associations captured in our main Google Image sample, both yielding a Pearson correlation of 0.5 ($p < .00001$). Panel B of Figure S7 displays the overall strength of the gender associations in both word2vec models, as compared to our main Google Image sample. There is no significant difference in the strength of gender associations between the 2013 word2vec model ($\mu = 0.22$) and our 2023 retrained word2vec model ($\mu = 0.22$) ($p = .14$, Student T-test, Two-sample). However, the strength of gender associations in our Google Image data ($\mu = 0.39$) is significantly higher than that of both word2vec models ($p < .00001$, Student T-test). Together, these analyses provide strong evidence that our main results are not driven by the temporal difference in data collection between the 2013 Google News word2vec model and our 2021 dataset of Google images. Moreover, these analyses suggest that gender associations for social categories have remained relatively stable in online news over the last decade, a trend consistent with recent work examining gender bias in Google books [9,10]. In the next supplementary section, we contextualize these results within a broader permutation analysis, which demonstrates the robustness of our findings across alternative word embedding models employing different data sources and algorithmic specifications."

We have also incorporated this new model into our prior robustness analyses, which compare a suite of word embedding models in terms of the skew and strength of their gender associations.

Please see the revised table S1 for a comprehensive overview of the specifications of all models examined, and the revised figure S8 for a graphical representation of these findings.

4. The experimental design of the study involves several moving components, and it is unclear how the findings and metrics from different data sources can be directly compared. Language and vision modalities possess unique characteristics, and the validity of comparing gender bias in word embeddings, which incorporate contextual co-occurrences from a large corpus, to bias in Google Image Search results obtained from diverse countries remains unclear.

We deploy several methods to validate the comparison between gender bias in word embeddings and Google Image search results, which we summarize below.

First, we replicate our results without relying on embedding approaches. To achieve this, we hired human coders to label every textual description and image uploaded into our experiment as either female (-1), male (1), or undecided/non-gendered (0). This labeling scheme was consistently applied at the level of individual descriptions for each modality – text and images. In addition, the descriptions for both modalities were collected using an identical experimental procedure, implemented during the same time frame via the randomization of participants across conditions. Using this approach, we replicate the core statistical result captured in our initial observational analyses, namely that gender bias is more frequent in images compared to text. In this way, we show that our findings are not an artifact of a potential statistical confound arising from the hypothetical incommensurability of gender associations as measured by embedding models and images.

Second, we validate the empirical validity of the gender associations as captured by word embedding models by demonstrating their strong correlation with gender associations detected via the simpler manual labeling scheme described above. Figure S33C shows that the gender associations measured by the Google News word2vec model are highly correlated ($r = .54, p < .0001$) with the gender associations of the occupational descriptions that participants downloaded and uploaded in the experiment. These associations were measured using the gender ratings provided by human coders using an identical labeling scheme across texts and images. Furthermore, figure S33D shows that the gender associations measured by the Google News word2vec model are also highly correlated with participants' subjective ratings of the gender association of each occupation in the experiment ($r = .79, p < .0001$). This provides statistical evidence that the gender associations measured by our word embedding models effectively capture the empirical traces of gender associations in the textual content that participants downloaded and engaged with via Google search. It also underscores that the associations measured by word embedding models mirror participants' subjective beliefs about the gender association of these occupations.

It is worth noting that recent work (Grand et al. 2022, reference below) leverages a very similar method for extracting dimensions from word embedding models based on word co-occurrences. Grand et al. show that this technique is strikingly effective at predicting individual-level human semantic judgments relating to basic concepts and their features. This concept is referred to as 'mental scales', and Grand et al. show how people leverage mental scales to make semantic judgments across a wide array of domains, including animals, weather, sports, cities, and clothing. This work further validates the claim that semantic relations as detected by the identification of dimensions in word embedding models can capture rich psychological information about the semantic associations that characterize how individual people understand and mentally represent

categories. We extend this method to the study of gender associations for social categories, and we find that this technique is similarly predictive of individuals' semantic judgments.

Next, we discuss how our image-based measure is also highly predictive of individual-level psychological judgments. This predictive power is nearly identical, statistically speaking, to that of word embedding models, further validating their congruence as measures, both in terms of their statistical compatibility and psychological relevance.

Our observational sample of Google Images is comparably predictive of the gender associations in the images that participants uploaded in the experiment, and also of participants' self-report gender ratings of occupations. Figure S33A shows that the gender associations measured by our sample of Google Images are highly correlated ($r = .58, p < .0001$) with the gender associations of the occupational images that participants downloaded and uploaded in the experiment. Figure S33B shows that the gender associations measured by our observational sample of Google Images are also highly correlated with participants' subjective ratings of the gender association of each occupation in the experiment ($r = .76, p < .0001$). Our ability to predict gender associations in participants' uploaded content and self-reported beliefs is essentially identical when using either word embedding models of gender associations or the frequency of genders in Google Images. This further establishes the statistical consistency and complementarity of these measures, despite the fundamental differences in these modalities.

Lastly, we generalize these validating findings beyond our experimental data by showing that the gender associations detected by our embedding measures of text and our observational sample of Google Images are both equally and highly correlated with human coders' subjective ratings of gender across all 3,496 social categories in our dataset (see Fig. 2B).

Memo Reference

Grand, Gabriel, Idan Asher Blank, Francisco Pereira, and Evelina Fedorenko. "Semantic Projection Recovers Rich Human Knowledge of Multiple Object Features from Word Embeddings." *Nature Human Behaviour* 6, no. 7 (July 2022): 975–87.

5. In terms of the IAT experiments, it is important to know the amount of text participants read and wrote, and how this corresponds to the amount of information conveyed by each image retrieved from Google.

This is an interesting suggestion. It is important to acknowledge that there is no consensus in the academic community for how to clearly and robustly measure the amount of information conveyed in images in a way that would numerically correspond to the number of words that participants read and wrote. Developing such a method would constitute a major methodological advance warranting its own separate study. For this reason, we believe this falls beyond the scope of our study. As articulated by a recent paper by Otto et al. (2019), "The automatic understanding of semantic correlations between text and associated images as well as their interplay has a great potential [...] However, automatic understanding of multimodal information is still an unsolved research problem." In the years since, there have been a few developments in this direction - for example, Birhane et al. (2021), Wolfe et al. (2023), and Cheema et al. (2023) have developed methods for identifying correlations between image and textual information specifically in the context of image-text pairs (e.g. between image and image captions) - but no standard method or comprehensive

framework has been developed for measuring semantic information in a modality-independent manner that would allow the number of concepts to be counted and directly compared between images and texts.

In our study, this limitation does not pose an issue, since our theory only requires a simple measure of the statistical gender skew in the representation of social categories within and across modalities. For this reason, our study develops a simple and highly interpretable framework for comparing texts and images in terms of the gender associated with depictions of a social category. We show how this approach can be robustly measured using automated methods such as word embedding models and deep learning image classification, as well as subjective judgments from human coders. We recruited the latter to classify the gender association of textual and image descriptions in a way that enables direct numerical comparisons; see our experimental design for details. We note that descriptions of occupations in texts and images often convey a lot of complex contextual information, including information about gender itself – such as the other social features that gender co-occurs with, e.g. age or social status – and other contextual elements, such as physical setting or historical period. We account for this uncontrolled signal in both images and text through a myriad of robust statistical models. These models show that all of our results hold (both in our observational and experimental studies) when controlling for unexplained variation, such as hidden variables relating to the social category or description, at various levels – the level of the specific description, the social category being described, and even the human coder classifying the description (where applicable). These efforts are comprehensive and demonstrate the robustness of our results to this concern. We thank Reviewer 1 for this suggestion, which points toward a promising direction for future innovation. The citations for the aforementioned references are provided below:

Memo References

- Birhane, Abeba, Vinay Uday Prabhu, and Emmanuel Kahembwe. “Multimodal Datasets: Misogyny, Pornography, and Malignant Stereotypes.” arXiv, October 5, 2021.
- Cheema, Gullal S., Sherzod Hakimov, Eric Müller-Budack, Christian Otto, John A. Bateman, and Ralph Ewerth. “Understanding Image-Text Relations and News Values for Multimodal News Analysis.” *Frontiers in Artificial Intelligence* 6 (2023).
- Otto, Christian, Matthias Springstein, Avishek Anand, and Ralph Ewerth. “Understanding, Categorizing and Predicting Semantic Image-Text Relations.” In *Proceedings of the 2019 on International Conference on Multimedia Retrieval*, 168–76, 2019.
- Wolfe, Robert, Yiwei Yang, Bill Howe, and Aylin Caliskan. “Contrastive Language-Vision AI Models Pretrained on Web-Scraped Multimodal Data Exhibit Sexual Objectification Bias.” In *Proceedings of the 2023 ACM Conference on Fairness, Accountability, and Transparency*, 1174–85. FAccT ’23. New York, NY, USA: Association for Computing Machinery, 2023.

6. Additionally, the paper lacks information on what IAT scores participants had for image and text-based stimuli before the experiments, as well as the influence of existing associations on the first and second IAT measurements.

We debated the pros and cons of running a within-subject or a between-subject design, and ultimately decided to run a between-subject design. An important reason to prefer a between-subject design over a within-person subject design in our context is to avoid anchoring effects. If we asked subjects to provide their explicit ratings of the gender of social categories - or their implicit associations via the IAT - prior to the experimental treatment, this is at risk of creating anchor

effects that will constrain how participants will rate the gender of categories after exposure to the experimental manipulation. The between-subject design within a randomized controlled experiment is a standard and valid way to avoid these anchoring effects and other statistical confounds, while maintaining the ability to robustly estimate the effect of the experimental treatments relative to the control condition.

To elaborate, in our between-subject experimental design, the control condition serves the statistical function of capturing subjects' beliefs and behaviors at baseline in an identical context that lacks exposure to the experimental manipulation. Given that our experiment is sufficiently statistically powered, and since participants were randomized across conditions, the standard statistical interpretation of our experimental design is that the subjects in the control condition approximate how we would expect participants in the image and text condition to behave in the absence of exposure to the experimental treatments. As an additional indicator of validity, we used the exact same IAT as a number of other studies, and the baseline rates of implicit bias measured in our control condition are nearly identical to those measured in large-scale longitudinal studies of this same IAT in a large representative population (see Charlesworth & Banaji 2022, ref. below).

It is thus reasonable to conclude that the baseline level of implicit bias captured by the IAT in our control condition approximates the expected distribution of participants' implicit bias levels prior to exposure to the experimental treatments in the text and image condition. Hence, differences in the average levels of implicit bias measured in our experimental conditions as compared to the control condition can be interpreted as the outcome of our experimental manipulation. This would correspond to a change in the baseline distribution of implicit bias levels represented in the control condition induced by the experimental treatments.

Memo reference

Charlesworth, Tessa E. S., and Mahzarin R. Banaji. "Patterns of Implicit and Explicit Stereotypes III: Long-Term Change in Gender Stereotypes." *Social Psychological and Personality Science* 13, no. 1 (January 1, 2022): 14–26.

7. Furthermore, the relationship between the 2,986 words and the female+arts/male+science IAT requires justification.

We provide this justification in the supplementary material. We clarify that we exposed subjects to a subset of occupational categories from the full Wordnet set. This subset focused on occupations related to the sciences and arts, omitting numerous non-occupational categories. We selected an equal number of science-related and art-related categories on a *prima facie* valid basis. In our pre-registration, we explicitly outlined all the occupational categories used in our design. We also pre-registered our prediction that exposure to the descriptions of these occupations in online images would amplify gender bias in participants' beliefs, compared to those who were exposed to text-based descriptions of these same occupations. We replicated this experiment and its results multiple times as part of our revisions (see footnote 3). More details can be found in the supplementary sections "A.3.2. Participant Experience" and "A.3.4. Measuring Implicit Bias Using the IAT", where we include the requested justification.

8. The title of the paper should be more precise. The authors acknowledge that while text can minimize bias through gender-neutral terminology, online images often explicitly depict people, making demographic information inevitably conveyed. Given the distinct nature of signals in vision and

language, further justification is necessary to support the current comparison of bias measurements in word embeddings and Google Image Search results. Word embeddings are trained on a corpus, while Google Image Search results come from a dynamic and context-dependent black-box, leading to uncontrolled experimental settings that affect the robustness and reliability of the study.

We entirely agree that our title could be made more precise and have refined it to better match our revised framing, which focuses on the relevance of our findings in light of the rising trend toward image-based information consumption and social media. Accordingly, our revised title is now - "How Online Images Amplify Gender Bias" - which captures our two main findings - (i) that images present an amplified signal of gender bias compared to textual sources, empirical census data, and participant's subjective beliefs (based on our observational analyses), and (ii) that exposure to images more strongly amplifies gender bias in experimental participants subject beliefs as compared to exposure to texts (based on our experimental results).

To reiterate, we demonstrate the robustness of our results beyond the use of embedding models. We did this using a standardized coding scheme implemented by human coders, who manually classified textual and image descriptions of social categories in terms of the gender they depict. Moreover, we validate the embedding methods we employ by showing their strikingly strong and robust correlation with various empirical traces of gender associations, including (i) people's subjective ratings of gender associations for social categories (as collected in two separate studies, our crowdsourcing study presented in Figure 2B and our experiment presented in Figure 3 and Figure S33), (ii) gender associations in the the content that participants uploaded in our experiment (Figure S33), and (iii) ground truth empirical data on the distribution of gender across occupations according to the U.S. Census.

9. The claim that prior studies on online gender bias primarily focus on text is challenged by the significant contributions of researchers like Abeba Birhane, Aylin Caliskan, and Olga Russakovsky, among many others, who have extensively analyzed bias and gender bias in image datasets or vision models trained on vast amounts of image or text-image pairs in multimodal settings. Similarly, Caliskan et al. have measured the gender associations of millions of words, highlighting the scalability of their approach.

To address any remaining concerns regarding the novelty of our study, we have compiled an in-depth literature review that engages the entire body of work associated with the authors mentioned by R1, namely Abeba Birhane, Aylin Caliskan, and Olga Russakovsky. We also included the key papers cited by these authors. In total, we have reviewed 34 papers. We have organized our literature review into Memo Table 1 (below), which identifies the methods and theoretical contributions of each study. Based on this review, we can confirm that none of these prior studies (and indeed no prior work that we can identify) has systematically compared and identified differences in the prevalence of gender bias in large online corpora of images and texts. Furthermore, no prior work has shown the direct consequences of this finding in terms of how images and text differ in their ability to activate psychological biases in people's beliefs. This is demonstrated by Memo Table 1 below; the numbers in this table indicate the study provided in the reference list at the end of this reply.

Data Type/Source	Measures Gender Bias	Measures Explicit Bias in Human Participants	Measures Implicit Bias in Human Participants	Compares Against Gender Representation in Census Data	Compares Gender Bias between Images and Text	Compares Effects of Images and Text on Gender Bias in Humans
Image Only	1–14	1,3,4	None	1,3	No	No
Text Only	9,15–24	None	15	15,16	No	No
Text-to-Image Generative AI	25–30	None	None	26,30	No	No
Multimodal (Both Text and Image)	28,32–34	None	None	None	No	No

Memo Table 1. A summary of our literature review of relevant studies on gender and related social biases in images and texts, including recent work on multimodal language-and-text AI models. The numbers indicate the corresponding study provided in the reference list shared at the end of this reply.

In this table, we organize papers along seven distinct dimensions: (1) their data type/source; (2) whether they measure gender bias in their data; (3) whether they measure explicit gender bias in human participants; (4) whether they measure implicit gender bias in human participants (e.g., the IAT); (5) whether they compare the biases in their observational sample to gender representations in the US Census; (6) whether they compare gender bias directly between images and text in their observational analyses; and (7) whether they compare the effects of images and text on gender biases in people’s explicit or implicit beliefs (e.g. via a psychological experiment).

In terms of data source, we organize papers into four groups (1) those that only examine image data, (2) those that only examine textual data, (3) those that examine generative text-to-image AI models, and (4) “multimodal” studies that examine both image and text data. We separate the study of text-to-image AI models from multimodal studies because, while text-to-image AI models are trained on both text and images, it is often challenging to disentangle the representations of text and images within these models. Often, researchers study these models by feeding them textual prompts and evaluating their image outputs, rather than studying the highly complex multimodal embeddings that underlie these models (which are frequently inaccessible to the user). Among the studies of text-to-image models we review, almost all of them measure gender bias only through studying the image data that these text-to-image models produce in response to prompting. However, these studies lack a systematic comparison between the representation of bias in the textual and image data on which these models are trained. We found only one study, described below, which also examined gender bias in the image captions produced by language-and-text models. But this study did not develop any comparison of gender bias levels between these image captions and the images themselves, nor any empirical investigation of the psychological implications of such an analysis.

A similar problem arises in papers exploring multimodal datasets and settings beyond text-to-image generative models. In all of the multimodal AI papers recommended by R1, the multimodal AI models examined are based on deep learning algorithms that incorporate text and image data into the same model. This results in the creation of multimodal embeddings for which it is challenging to disentangle the textual and image features of these embeddings. Again, in all of the multimodal AI studies we review, there is no systematic comparison of how bias is encoded separately in images and text, nor in how these modalities differ in their psychological impact on human participants.

We have compiled this in-depth literature review to make clear a key point, which we initially aimed to convey in our prior response. After reviewing all of the papers associated with the authors highlighted by Reviewer 1, as well as all of the relevant papers cited by the papers associated with these authors, we cannot find any studies that perform the analyses that are core to the theory and findings of our paper. Specifically, we have found no studies that engage in a direct comparison of gender bias across separate large online sources of images and text through a transparent and interpretable multimodal framework. Similarly, we have been unable to identify studies which simultaneously show how text and images differ in their psychological effects on gender bias in human users exposed to such content.

Furthermore, we found that while some studies satisfied the criteria of one or two columns in Memo Table 1, very few spanned two or more columns, whereas our study occupied all columns. Furthermore, none of the studies occupied the two columns (highlighted in red) that mark our chief contributions. Our paper thus marks a novel and significant advance upon prior work on gender bias in online data and its psychological consequences.

Given the scope of this literature review, its level of detail exceeds the space limitations of a manuscript at *Nature*. Consequently, we have only included references to the most relevant papers in our manuscript. We are happy to include this literature review table in our supplementary material if preferred by the Editor and R1. Specifically, we have added the following references to our manuscript:

- [12] Aylin Caliskan, et al. Semantics derived automatically from language corpora contain human-like biases. *Science*, 356(6334):183–186, 2017.
- [13] April H Bailey, et al. Based on billions of words on the internet, people = men. *Science Advances*, 8(13):eabm2463, 2022.
- [31] Federico Bianchi, et al. Easily accessible text-to-image generation amplifies demographic stereotypes at large scale. In *Proceedings of the 2023 ACM Conference on Fairness, Accountability, and Transparency*, pages 1493–1504, 2023.
- [32] Robert Wolfe, et al. Contrastive language-vision ai models pretrained on web-scraped multimodal data exhibit sexual objectification bias. In *Proceedings of the 2023 ACM Conference on Fairness, Accountability, and Transparency*, pages 1174–1185, 2023.

Our intent is to make especially clear the novelty of our research in the context of existing literature. For this reason, we would like to provide additional commentary on some of the papers in our reference list, which may, solely based on their title, appear to be more similar to our study than they truly are upon closely reading their methods and findings. See below.

- Steed, Ryan, and Aylin Caliskan. “Image Representations Learned With Unsupervised Pre-Training Contain Human-like Biases.” In *Proceedings of the 2021 ACM Conference on Fairness, Accountability, and Transparency*, 701–13, 2021.

This study exclusively employs images as stimuli. The authors build upon prior methods for measuring bias in text by adapting these methods to examine biases within images. However, these authors do not use these measures to compare the levels of gender bias between text and images; consequently, they do not identify or report the finding that gender bias is stronger in images than text. Furthermore, this paper does not contain any psychological experimentation that would establish that images and texts differ in their effects on gender bias in human participants. Similarly, while this study claims to examine implicit human biases in images, the authors do not use an IAT to measure implicit bias in a human population, nor do they use self-report judgments of human bias; instead, they use statistical methods for interpreting correlations in image data as evidence of implicit human biases.

- Wolfe, Robert, and Aylin Caliskan. “American == White in Multimodal Language-and-Image AI.” In *Proceedings of the 2022 AAAI/ACM Conference on AI, Ethics, and Society*, 800–812. AIES '22. New York, NY, USA: Association for Computing Machinery, 2022.

This paper examines racial biases in multimodal AI models that are trained on both textual and image data. The authors do not explicitly or systematically make a comparison of the level of gender bias between textual and image data. In fact, it is very challenging to disentangle the textual and image embeddings within the multimodal embeddings formed by such models. The authors also examine racial biases in a text-to-image model purely by examining the image outputs of this model, again with no systematic comparison in gender bias (or any kind of bias) between the image and textual data. What is more, this study involves no psychological experimentation to show how images and text different in their ability to activate biases in human participants.

- Bianchi, Federico, ..., and Aylin Caliskan. “Easily Accessible Text-to-Image Generation Amplifies Demographic Stereotypes at Large Scale.” In *Proceedings of the 2023 ACM Conference on Fairness, Accountability, and Transparency*, 1493–1504. FAccT '23.

This paper examines the prevalence of gender and racial stereotypes in the images produced by popular generative text-to-image AI models. While the paper explores the variance in bias elicited by different textual prompts in the resulting images, there is no direct or systematic comparison between the prevalence of gender bias in textual and image data separately. In fact, it is known to be difficult to separate the textual and image-based embeddings in these text-to-image models. Moreover, the authors do not examine the relationship between the observed biases and psychological biases in users of these models. There is no examination or discussion of how images and texts differ in their ability to activate bias in people’s beliefs.

- Wolfe, Robert, ..., and Aylin Caliskan. “Contrastive Language-Vision AI Models Pretrained on Web-Scraped Multimodal Data Exhibit Sexual Objectification Bias.” In *Proceedings of the 2023 ACM Conference on Fairness, Accountability, and Transparency*, 1174–85. FAccT '23.

This study examines a specific aspect of gender bias – namely, sexualization – present in the images and accompanying textual captions generated by multimodal AI models. These models are based on deep learning methods, incorporating both textual and image data scraped from

the internet into the same model. This integration creates multimodal embeddings that prove challenging to disentangle into distinct textual and image features. In fact, the textual data examined in this study is inherently entangled with the images since these captions are descriptions of the images themselves; in this way, this textual data is limited as a measure of how women are represented in textual media separate from images. Most importantly, no direct or systematic comparison is made between the prevalence of gender bias in textual and image data separately, nor in their psychological impacts. Instead, this study applies computational techniques to estimate implicit bias in the image data from these models, without including or comparing against measures of implicit bias in raw human participants.

- Wolfe, Robert, Mahzarin R. Banaji, and Aylin Caliskan. "Evidence for Hypodescent in Visual Semantic AI." In *Proceedings of the 2022 ACM Conference on Fairness, Accountability, and Transparency*, 1293–1304. FAccT '22.

This study examines a language-and-image AI model, focusing solely on examining racial and gender biases in the image representations produced by this model. No systematic or direct comparison is made between how gender bias is represented in images and text. There is also no comparison to psychological data from human participants, and no examination of the different effects of images and texts on gender bias in people's beliefs.

- Wolfe, Robert, and Aylin Caliskan. "Markedness in Visual Semantic AI." In *Proceedings of the 2022 ACM Conference on Fairness, Accountability, and Transparency*, 1269–79. FAccT '22.

Although the study aims to uncover biases in the image captioning behavior of multimodal models, their study does not measure or systematically compare gender bias across image and text, nor does it identify the differences in how each data modality contributes to gender bias in its multimodal embeddings. As well, there is no systematic or direct comparison in how gender bias is represented in images and texts, nor of their psychological effects on humans.

- Birhane, Abeba, Vinay Uday Prabhu, and Emmanuel Kahembwe. "Multimodal Datasets: Misogyny, Pornography, and Malignant Stereotypes." arXiv, October 5, 2021.

This paper focuses on identifying biases and offensive content present in the dataset used to train LAION-400M, a large language-and-text AI model trained on text and image pairs, such as images and their captions scraped from across the web. All of the paper's main analyses examine biases within these image-text pairs. There is some effort to correlate the semantic features in the images and texts, but these are determined by comparing models' textual taggings of images against associated texts. It is important to note two key aspects: (i) these correlations are not leveraged to examine gender bias. Instead, they are leveraged to verify semantic coherence across these modalities (see their Fig. 3). And (ii) the analysis does not identify differences in the level of gender bias encoded in images and texts. Accordingly, there is no psychological investigation into how images and texts differ in how they impact biases in people's beliefs.

Memo References

1. Kay, M., Matuszek, C. & Munson, S. A. Unequal Representation and Gender Stereotypes in Image Search Results for Occupations. in *Proceedings of the 33rd Annual ACM Conference on Human Factors in Computing Systems* 3819–3828 (Association for Computing Machinery, 2015).

2. Buolamwini, J. A. Gender shades: intersectional phenotypic and demographic evaluation of face datasets and gender classifiers. (Massachusetts Institute of Technology, 2017).
3. Metaxa, D., Gan, M. A., Goh, S., Hancock, J. & Landay, J. A. An Image of Society: Gender and Racial Representation and Impact in Image Search Results for Occupations. *Proc. ACM Hum.-Comput. Interact.* **5**, 26:1-26:23 (2021).
4. Vlasceanu, M. & Amodio, D. M. Propagation of societal gender inequality by internet search algorithms. *Proceedings of the National Academy of Sciences* **119**, e2204529119 (2022).
5. Meister, N. *et al.* Gender Artifacts in Visual Datasets. Preprint at <https://doi.org/10.48550/arXiv.2206.09191> (2022).
6. Steed, R. & Caliskan, A. Image Representations Learned With Unsupervised Pre-Training Contain Human-like Biases. in *Proceedings of the 2021 ACM Conference on Fairness, Accountability, and Transparency* 701–713 (2021).
7. Nagpal, S., Singh, M., Singh, R. & Vatsa, M. Deep Learning for Face Recognition: Pride or Prejudiced? Preprint at <https://doi.org/10.48550/arXiv.1904.01219> (2019).
8. Wang, Z. *et al.* Towards Fairness in Visual Recognition: Effective Strategies for Bias Mitigation. in 8919–8928 (2020).
9. Caliskan, A., Ajay, P. P., Charlesworth, T., Wolfe, R. & Banaji, M. R. Gender Bias in Word Embeddings: A Comprehensive Analysis of Frequency, Syntax, and Semantics. in *Proceedings of the 2022 AAAI/ACM Conference on AI, Ethics, and Society* 156–170 (Association for Computing Machinery, 2022).
10. Birhane, A. & Prabhu, V. U. Large image datasets: A pyrrhic win for computer vision? in *2021 IEEE Winter Conference on Applications of Computer Vision (WACV)* 1536–1546 (2021).
11. Birhane, A., Prabhu, V. U. & Whaley, J. Auditing saliency cropping algorithms. in *2022 IEEE/CVF Winter Conference on Applications of Computer Vision (WACV)* 1515–1523 (2022).
12. Yang, K., Qinami, K., Fei-Fei, L., Deng, J. & Russakovsky, O. Towards fairer datasets: filtering and balancing the distribution of the people subtree in the ImageNet hierarchy. in *Proceedings of the 2020 Conference on Fairness, Accountability, and Transparency* 547–558 (Association for Computing Machinery, 2020).
13. Schwemmer, C. *et al.* Diagnosing Gender Bias in Image Recognition Systems. *Socius* **6**, 2378023120967171 (2020).
14. Wang, A. & Russakovsky, O. Overwriting Pretrained Bias with Fine Tuning Data. Preprint at <https://doi.org/10.48550/arXiv.2303.06167> (2023).
15. Caliskan, A., Bryson, J. J. & Narayanan, A. Semantics derived automatically from language corpora contain human-like biases. *Science* **356**, 183–186 (2017).
16. Garg, N., Schiebinger, L., Jurafsky, D. & Zou, J. Word embeddings quantify 100 years of gender and ethnic stereotypes. *PNAS* **115**, E3635–E3644 (2018).
17. Kozłowski, A. C., Taddy, M. & Evans, J. A. The Geometry of Culture: Analyzing the Meanings of Class through Word Embeddings. *Am Sociol Rev* **84**, 905–949 (2019).
18. Jones, J. J., Amin, M. R., Kim, J. & Skiena, S. Stereotypical Gender Associations in Language Have Decreased Over Time. *Sociological Science* **7**, 1–35 (2020).
19. Lewis, M. & Luyyan, G. Gender stereotypes are reflected in the distributional structure of 25 languages. *Nature Human Behaviour* **4**, 1021–1028 (2020).
20. Charlesworth, T. E. S., Caliskan, A. & Banaji, M. R. Historical representations of social groups across 200 years of word embeddings from Google Books. *Proceedings of the National Academy of Sciences* **119**, e2121798119 (2022).
21. Bailey, A. H., Williams, A. & Cimpian, A. Based on billions of words on the internet, people = men. *Science Advances* **8**, eabm2463 (2022).
22. Mei, K. X., Fereidooni, S. & Caliskan, A. Bias Against 93 Stigmatized Groups in Masked Language Models and Downstream Sentiment Classification Tasks. in *2023 ACM Conference on Fairness, Accountability, and Transparency* 1699–1710 (2023).
23. Swinger, N., De-Arteaga, M., Heffernan IV, N. T., Leiserson, M. D. & Kalai, A. T. What are the biases in my word embedding? Preprint at <http://arxiv.org/abs/1812.08769> (2019).
24. Guo, W. & Caliskan, A. Detecting Emergent Intersectional Biases: Contextualized Word Embeddings Contain a Distribution of Human-like Biases. in *Proceedings of the 2021 AAAI/ACM Conference on AI, Ethics, and Society* 122–133 (Association for Computing Machinery, 2021).

25. Kim, E., Kim, S., Shin, C. & Yoon, S. De-stereotyping Text-to-image Models through Prompt Tuning. (2023).
26. Bianchi, F. *et al.* Easily Accessible Text-to-Image Generation Amplifies Demographic Stereotypes at Large Scale. in *Proceedings of the 2023 ACM Conference on Fairness, Accountability, and Transparency* 1493–1504 (Association for Computing Machinery, 2023).
27. Fraser, K. C., Kiritchenko, S. & Nejadgholi, I. A Friendly Face: Do Text-to-Image Systems Rely on Stereotypes when the Input is Under-Specified? Preprint at <https://doi.org/10.48550/arXiv.2302.07159> (2023).
28. Wolfe, R., Yang, Y., Howe, B. & Caliskan, A. Contrastive Language-Vision AI Models Pretrained on Web-Scraped Multimodal Data Exhibit Sexual Objectification Bias. in *Proceedings of the 2023 ACM Conference on Fairness, Accountability, and Transparency* 1174–1185 (Association for Computing Machinery, 2023).
29. Seshadri, P., Singh, S. & Elazar, Y. The Bias Amplification Paradox in Text-to-Image Generation. Preprint at <https://doi.org/10.48550/arXiv.2308.00755> (2023).
30. Naik, R. & Nushi, B. Social Biases through the Text-to-Image Generation Lens. Preprint at <https://doi.org/10.48550/arXiv.2304.06034> (2023).
31. Wolfe, R. & Caliskan, A. American == White in Multimodal Language-and-Image AI. in *Proceedings of the 2022 AAAI/ACM Conference on AI, Ethics, and Society* 800–812 (Association for Computing Machinery, 2022).
32. Wolfe, R., Banaji, M. R. & Caliskan, A. Evidence for Hypodescent in Visual Semantic AI. in *Proceedings of the 2022 ACM Conference on Fairness, Accountability, and Transparency* 1293–1304 (Association for Computing Machinery, 2022).
33. Wolfe, R. & Caliskan, A. Markedness in Visual Semantic AI. in *Proceedings of the 2022 ACM Conference on Fairness, Accountability, and Transparency* 1269–1279 (Association for Computing Machinery, 2022).
34. Birhane, A., Prabhu, V. U. & Kahembwe, E. Multimodal datasets: misogyny, pornography, and malignant stereotypes. Preprint at <https://doi.org/10.48550/arXiv.2110.01963> (2021).

10. *In summary, the paper addresses an important issue, but it lacks clarity on how the current methodology can yield robust, valid, and reliable answers.*

We are grateful to R1 for their feedback, which has been instrumental in improving our manuscript. We appreciate that R1 sees the value of the core issue we are addressing.

Reply to Reviewer 2 (R2)

1. I congratulate the authors on the extensive revision of this paper/ They seemed to have gone above and beyond to improve the paper and to address the reviewers' concerns (as far as I can see). I appreciate the long list of robustness analyses that they have added to the paper and the structural revisions that have been done. My prior concerns that led me to not yet fully trust the results and associated claims, have in most but not all cases resolved in this revision. I'll mention my prior points in the same order as before, though the authors have done a great job revising the paper such that my later points (6-11) all seem well addressed.

Thank you! Your recommended revisions have led to enormous improvements in the clarity, thoroughness, and robustness of our findings. We greatly appreciate your insightful comments.

2. I'll start with my biggest concern. What I write here is ultimately up to the editors, and how they view the contribution of the paper. My prior concern was that the paper may overclaim its finding based on something that is inherently evident from the outset: namely that images carry more bias purely by the nature of the medium and that individual level effects of text versus images — i.e., images carrying more weight in the influence process — may also not be super surprising. I appreciate the author's response on

*this concern that goes into technical detail on how they rendered the two types media comparable. Yet, the inherent *theoretical* difference between images and texts and its influence on respondent remains. How surprising is the finding? Is it the scope in which this has been done in this study? The nuanced methodological approach? Or is there something substantively new we learn about human gender attitudes and how that is influenced? I see that this study addresses the first two questions *really* well, but the way in which it deals with the (perhaps more fundamental) second question does remain underdeveloped.*

We have given this astute comment careful thought and have arrived at a stronger framing that more clearly articulates the novelty and importance of our findings. We fully agree with R2 that our findings rest upon an inherent theoretical difference between texts and images. As R2 aptly emphasizes, texts can more easily minimize or obscure information about gender (e.g., via gender-neutral terminology), whereas images featuring people necessitate the communication of demographic information such as gender. We view this as a strength that lends our findings foundational significance and robustness. While the implication of this theoretical difference for gender bias – namely, that gender bias is stronger in images than text – follows almost deductively from this inherent theoretical difference, our contribution lies in being the first to empirically document this difference at scale and demonstrate its social and psychological implications. Our findings not only illuminate the distribution of gender bias online, but also the psychology of how gender bias can reinforce people’s beliefs. In terms of highlighting, in R2’s words, ‘*something substantively new we learn about human gender attitudes and how that is influenced*’, our revised study highlights two major and important insights into this question, which we describe below.

First, as we emphasize in our revised framing, we show not only that gender bias is amplified in images compared to texts (which had yet to be demonstrated at scale), but also, more importantly, that this foundational difference between images and texts has far reaching social and psychological implications. This distinction becomes particularly significant given the well-documented rise of image-based content in our everyday information diets and social interactions online. As our new introduction explains, the average internet user now spends less time reading text and more time viewing images. Advertisers, news agencies, and social media companies are increasingly relying on images to grab people’s attention online. Over the last decade, the number of images on the internet – and the number of images that people are interacting with in general – has skyrocketed. This growing trend is expected to further increase with the surging popularity of image-based social media platforms like Instagram, Snapchat, and TikTok. Additionally, the rapid proliferation of text-to-image AI models makes it even easier for people to generate and disseminate visual content. In essence, there is a large-scale shift in human information consumption away from texts and toward images. Our study highlights the far-reaching implications of this shift for the mass diffusion and reinforcement of gender bias online.

Secondly, our study is the first, to our knowledge, to devise an experimental paradigm for showing that increasing exposure to images over texts online amplifies gender biases in the explicit and implicit beliefs of internet users. This provides direct causal evidence for the psychological impact that the rising cultural trend toward image-based media can have by amplifying gender biases. While prior work in cognitive psychology has examined the foundational role of images in human cognition - for example, showing that mental imagery likely developed before verbal reasoning and is actively recruited to support the production and comprehension of text - surprisingly, no prior work has explored the implications of these insights for the representation and transmission of gender bias. From the perspective of embodied cognition - which emphasizes the central role played

by sensory data and mental imagery throughout human reasoning - the rising trend toward image-based media can be identified as a large-scale, collective shift in the kind of sensory data and cognitive processes that mediate human comprehension and communication. Given this substantial shift, our experimental finding that gender bias is not only more prevalent but also more psychologically potent in images than texts has far reaching implications, especially for the cognitive and cultural development of humans and the ethical challenges that face the future of society, which we elaborate upon in our revised discussion.

In response to comment 9 from R1, we have developed a comprehensive literature review of computational work on the topic of gender bias in online data, covering 34 recent papers). This review indicates that no prior work, to the best of our knowledge, documents the foundational difference between images and text that we highlight, nor does it demonstrate its large-scale implications regarding the distribution of gender bias online and its psychological consequences. Similarly, as far as we know, we are the first to articulate the critical importance of this theoretical difference in light of the seismic shift in human culture away from texts and toward images as our primary mode of information consumption and exchange.

To better align with this framing, we have revised our title to read as follows: “How Online Images Amplify Gender Bias”. This improved title is intended to refer to our two main, novel contributions; first, the amplification of gender bias in images as compared to text, census data, and public opinion (as captured by our observational data), and second, the amplification of gender bias in people’s beliefs when exposed to images as compared to text (as captured by our experimental findings).

While we maintain that our study offers key innovations at the methodological level (including the development of a cross-platform image dataset of social categories that far exceeds the scale of prior studies), we think that the main innovations of interest regarding R2’s comment concern the theoretical and empirical identification of this foundational difference between images and text, and the demonstration of its social and psychological implications. We believe this contribution is more strongly conveyed by our revised framing, which we provide below for your direct review.

From the introduction:

“Images increasingly pervade the information we consume daily. The number of images shared online has skyrocketed in recent years [1-3], leading internet users to spend less time reading and more time viewing images [4]. Images from platforms like Google and Wikipedia are downloaded and circulated by millions of people every day [5,6]. This growing trend is widely recognized by the tech and venture capital industry [2,3], prompting advertisers and news agencies to more heavily rely on images to capture people’s attention online [7,8]. Here, we provide large-scale evidence that the proliferation of online images may significantly exacerbate gender bias in people’s beliefs. Unlike text, which can be crafted to minimize bias via gender-neutral terminology, images inherently convey demographic information when depicting people, suggesting that gender bias may be more prevalent in images than text. Psychological research also indicates that humans process images more quickly, implicitly, and memorably than text [9,10], suggesting that images may be stronger than text at reinforcing gender bias in people’s beliefs. To test these predictions, we develop novel computational and experimental techniques for comparing gender bias and its psychological impact across online images and texts. First, we study the gender associations of 3,495 social categories in over one million images from Google, Wikipedia, and IMDb, as well as in textual representations of these categories using word

embedding models trained on billions of words from these platforms. We find that online images exhibit significantly stronger gender bias for both female- and male-typed categories than online text. Building on prior work showing that women are underrepresented in online text [11-13] and images [14-16], we further show that the underrepresentation of women is significantly worse in online images compared to not only text, but also public opinion and US census data on occupations. Next, we conducted a pre-registered experiment with a nationally representative US sample ($N = 450$), which shows that googling for images rather than textual descriptions of occupations amplifies gender bias in people's beliefs. This includes amplifying the implicit bias toward associating women with arts and men with science, a bias linked to systemic inequalities in academia and industry [17,18]. Strikingly, only exposure to online images increased gender bias for several days following our experiment.”

From the discussion:

“Altogether, these findings suggest that the algorithmically-augmented rise of images may come at a critical social cost. Gender bias online is more prevalent and more psychologically potent in images than text. The rising centrality of visual content in our information diets may thus exacerbate gender bias by magnifying its digital presence and deepening its psychological entrenchment. This problem is expected not only to impact the well-being, social status, and economic opportunities for women, who are systematically underrepresented in online images according to our measures, but also men in highly female-typed categories such as care-oriented or community-oriented occupations [28,29].

Our findings are especially alarming given that image-based social media platforms like Instagram, SnapChat, and TikTok are surging in popularity, accelerating the mass production and circulation of images. In parallel, popular search engines like Google are increasingly incorporating images into their core functionality, for example, by including images as a default part of text-based searches [30]. The apex of these developments is perhaps the widespread adoption of text-to-image AI models that allow users to automatically generate images via textual prompts, further accelerating the production and circulation of images. Recent work identifies rampant gender and racial biases in the images that text-to-image models generate [31], suggesting that these innovations may intensify the large-scale spread of social biases. Consistent with recent work [32], our study suggests that these biases in multimodal AI may stem in part from their reliance on public images from platforms like Google and Wikipedia, which encode pervasive gender bias as revealed by our findings.”

3. The authors note the differences in text and images and how they prime gender bias. My point 2 above doubts whether that is a substantively interesting finding. I do appreciate the labeling of the sources of the data. It may essentially be generated by gendered usage levels of tech and particularly blogs and the types of photos that are uploaded there.

We hope that the revisions we have made to address point 2 from R2 (above) also contribute to addressing this related concern. As we discuss in reply to comment 2, while the difference in priming effects of text and images may be intuitive, we are the first (to our knowledge) to demonstrate the implications of this difference for the transmission of gender bias. This may have far reaching implications in light of the growing trend toward image-based information diets and social media. In reply to comment 6 by R4, we provide additional analyses that support our finding that images prime gender bias more strongly than text. Specifically, our expanded analysis shows that this result

holds when controlling for the level of gender bias in the online content associated with each category according to images and texts. This suggests that images amplify psychological biases in internet users' beliefs beyond exposure to text, even when holding the frequency of gender bias in both images and text constant.

4. The authors sufficiently address my prior point 4, yet a new concern arose when reading. Would participants homophilously select photos based their own gender? Perhaps I missed it, but that seems like an important control. That relates to entrenched gender power dynamics in society writ large and how effects of gendered imagery may be conditional on participants' own gender.

This is a very interesting idea that identifies an important set of controls. We explore these questions through a series of additional supplementary analyses (copied below for reference). In sum, we do not find evidence of gender homophily in participants' choices for which descriptions to upload for occupations. We observe no significant differences in the gender of content uploaded by male and female participants for the same occupation. We also do not find that participants are biased toward uploading content that matches their own gender. Instead, our data suggests that both male and female participants focused on uploading descriptions that matched their judgment of which gender is most associated with each occupation. These results hold for both the text and image condition.

We went a step further to ensure the robustness of our findings. We also show that all of our main experimental results hold when controlling for the gender of the participant and whether their uploaded occupation description matched their own gender. We do find that the absolute strength of participants' gender associations is significantly higher when the description they upload for a given occupation matches their own gender; however, we find that this correlation has no detectable influence on the main statistical effect of each experimental condition. While including this as a control, we continue to find that participants in the image condition report significantly stronger gender associations than participants in the text and control conditions. Based on these analyses, we conclude that the effect of the image condition observed in our experiment is unlikely to be driven by underlying homophily-related biases in our study population. In other words, the effects of gendered imagery do not appear to be conditional on participants' own gender within the scope of our experimental design and outcome measures of interest. We acknowledge the valuable suggestion of R2 that further examining the effects of gender homophily is an important direction for future research. The associated supplementary analyses on this topic are provided below.

“Robustness to Controlling for Participants' Gender

Here, we test the robustness of our experimental results when accounting for the gender of the participants completing the task. First, we examine whether participants are more likely to upload descriptions of occupations that correspond to their own gender (i.e., whether women are more likely to upload depictions of women in occupations, and similarly for men).

$y = \text{Gender of Uploaded Occupation Description}$

Text Condition [vs. Image]	0.13*** (0.02)
Participant Gender [Male]	0.03 (0.02)
Occupation Fixed Effects	X
Constant	0.19* (0.07)
N	5525
R^2	0.20

Standard errors in parentheses (clustered by participant)

* $p < 0.05$, ** $p < 0.01$, *** $p < 0.001$

Table S10. An OLS regression predicting the gender of participants' uploaded descriptions of occupations (-1=Female, 0=Neutral, 1=Male), as a function of participants' gender and experimental condition, with fixed effects by occupation. The control condition is excluded from this model because none of the content uploaded in the control condition was gendered (e.g., *apple*). Standard errors are clustered at the participant level.

Table S10 presents a model testing whether participants' gender significantly predicts the gender of the descriptions they upload across different occupations. The results show that there is no significant correlation between participants' gender and the gender of the descriptions they upload ($\beta[\text{MALE}] = 0.03$, $\text{SE} = 0.02$, $p = .12$). Qualitatively, in the image condition, 56% of descriptions uploaded by female participants depicted women, and 53% of descriptions uploaded by male participants depicted men, illustrating stable trends across participant gender. The same pattern holds in the text condition, albeit it had fewer gendered descriptions overall. In the text condition, both female and male participants uploaded nearly identical fractions of male and female content: 12% (10.9%) of textual descriptions uploaded by women were female (male), and 11.9% (11.4%) of textual descriptions uploaded by men were female (male). These results collectively suggest that the experimental outcomes observed are not driven by underlying biases among participants toward uploading content that mirrors their own gender.

We further evaluate the robustness of our experimental outcomes to controlling for participant gender using two complementary statistical models. First, we test whether our main outcome of interest holds – namely, that the strength of participants' gender associations for occupations is significantly higher in the image condition – while controlling for the gender of the participant (self-identified as "Male" or "Female"). Second, we test whether the same outcome holds when controlling for whether the self-identified gender of the participant matches the gender of the description they uploaded for a given occupation. This second analysis directly controls for any psychological biases participants may have toward exaggerating or reducing gender associations as a function of sharing the perceived gender of the target occupation.

	y = Strength of Gender Associations in Participant Responses		
	(1)	(2)	(3)
Image Condition [vs. Control]	0.06** (0.02)	0.06** (0.02)	0.06** (0.02)
Text Condition [vs. Control]	0.00 (0.02)	0.00 (0.02)	0.00 (0.02)
Participant Gender [Male]		-0.03 (0.02)	-0.03* (0.02)
Participant Gender Match Upload Gender			0.15*** (0.01)
Occupation Fixed Effects	X	X	X
Constant	0.44*** (0.02)	0.45*** (0.03)	0.39*** (0.03)
N	8780	8780	8780
R ²	0.15	0.16	0.21

Standard errors in parentheses (clustered by participant)
 * $p < 0.05$, ** $p < 0.01$, *** $p < 0.001$

Table S11. Models present OLS regressions predicting the absolute strength of participants' gender associations for occupations, corresponding to the experimental outcome presented in fig. 3F in the main text. All models include standard errors clustered at the participant level.

Table S11 presents the aforementioned models. Model 1 shows the main experimental effect without controlling for the demographic of participants. Here, we see that being randomly assigned to the image condition is associated with a significant increase in the average strength of participants' gender associations, while controlling for the occupation being evaluated and clustering standard errors at the participant level ($\beta = 0.06$, $SE = 0.02$, $p < .01$).

Model 2 presents the same model at Model 1 while controlling for the gender of the experimental participant. Model 2 finds no significant correlation between the gender of the participant and the absolute strength of participants gender associations for occupations ($\beta[\text{MALE}] = -0.03$, $SE = 0.02$, $p = .13$); adding this demographic control variable has no impact on the main statistical effect of being randomly assigned to the image condition.

Model 3 replicates Model 2 while also including a dummy variable which captures whether a participant's gender matches the gender of the description they uploaded for a given occupation. Adding this variable significantly improves the R^2 of the model, an increase from 0.15 (Model 1) and 0.16 (Model 2) to 0.21 (Model 3) ($p < .05$, ANOVA). When a participants' self-identified gender matched the gender of the content they uploaded for a given occupation, this was associated with a significant and sizable increase in the absolute strength of their gender association for this occupation ($\beta = 0.15$, $SE = 0.01$, $p < .001$). Importantly, including this variable again had no statistical impact on the strength or significance of the main experimental effect of being randomized to the image condition. That is, controlling for whether participants' self-identified gender matched the gender of their uploaded content, we continue to find that being randomized to the image condition was associated with a significant increase in the absolute strength of participants' gender associations relative to those randomized to the text and control condition ($\beta = 0.06$, $SE = 0.02$, $p < .01$; effect strength is identical to Model 1 and 2). In none of the models do we find that being randomly assigned to the text condition leads participants to exhibit statistically different gender associations for occupations compared to those randomized to the control condition. From these analyses, we conclude that our main

experimental outcome of interest is robust to controlling for the gender of our experimental participants.”

5. The authors have done a decent job addressing this concern, though I still think the front end needs work. Statements such as “Images may this provide a clearer window into large-scale patterns of gender bias today.” are challenging to test and seem somewhat underdeveloped from micro-level influence mechanism differences between images and text. This relates to my point 2: they do not provide a clearer window, just a substantively different window. Similarly, a statement that there is a “puzzling disconnect” between textual representation of gender online and gender disparity in society seems like an overclaim. Women politicians, women public figures, or even women regular users on Twitter would disagree given the amount of textual abuse they likely receive compared to their men counterparts. Do the authors need to phrase their premise as a paradox that itself is somewhat disconnected from what happens in the online sphere? In other words, the front end and setup of the paper’s questions (as I also mention in point 2) can still use some improvement and development.

This insightful comment from R2 was crucial in spurring us to majorly revise our introduction and discussion. As mentioned in our reply to comment 2 by R2, we realized that a stronger way to communicate the importance of our findings is to highlight their significance with respect to the rising trend toward image-based information consumption and social media. Acknowledging R2’s comment, we agree that our prior framing evoked a tenuous tension between textual representations of gender and the socioeconomic distribution of gender in society. We have now revised our introduction and discussion to eliminate this prior framing and, instead, to focus on highlighting the timeliness and importance of our study in light of empirical data indicating the growing centrality of images online. In our reply to comment 2 from R2, we copy the direct framing revisions to the introduction and discussion that we have made to address these concerns. Regarding the specific sentence mentioned in the comment, we recognized that it was not very precise, and we deleted it.

6-11. My prior points 6 through 11 all seem very sufficiently addressed. I want to emphasise the amount of work the authors have done between the prior version and this version of the manuscript.

Thank you!

Reply to Reviewer 3 (R3)

The authors’ revisions successfully address the majority of the points I raised in my previous review, and I only have a few remaining comments in the spirit of improving the final version as much as possible and maximizing its impact.

1. It’s great that the authors found a way to estimate the inter-rater reliability of the MTurk coders. However, percent agreement is a crude and somewhat poor metric of inter-rater reliability which does not correct for chance agreement, and is therefore often considered insufficient (Lombard et al., 2002; Neuendorf, 2002). Chance-corrected agreement is more appropriate because it makes the assumption that some portion of coders’ agreement is due to chance and adjusts for that. It would be highly desirable for the authors to complement the simple agreement statistics with chance-corrected indicators such as Scott’s pi, Cohen’s kappa, Krippendorff’s alpha, or (ideally) Gwet’s AC. These different indicators vary in their computation, assumptions, and stability, and Gwet’s AC in particular is highly

recommended (e.g., Heyman et al., 2014), as it avoids the instability associated with the other coefficients.

We agree with R3 that percent agreement is a limited metric of inter-rater reliability. Following R3's helpful recommendations, we have taken the necessary steps to quantify the inter-rater reliability of our sample using Gwet's AC. According to this measure, our sample achieves substantial and reliable inter-coder agreement. We have copied these additions to the supplementary appendix below:

"For robustness, we also calculate the chance-corrected inter-coder reliability of raters in our sample using GWET's AC [S18]. We calculated GWET's AC using the irrCAC package in R. Our coders achieved a coefficient of 0.48. This GWET coefficient falls within the 'good' range of reliability according to standard interpretations of this measure, especially considering that our sample was limited to only three coders per image (see [S18]). Combined with our percent-agreement results, these analyses suggest that our coders provided reliable statistical judgments of the gender of faces in our sample."

2. I don't necessarily agree that there is a big problem of competing definitions around the term "stereotype", although different authors may emphasize some aspects and not others (e.g., focusing on "traits and behaviors" versus "social roles"). However, I see how replacing "stereotypes" with "gender bias" helps address the issue of conceptually defining stereotypes. But the current version of the manuscript still mentions "stereotypes" or derived words ("stereotypical") (e.g., twice on p. 10, once on p. 11). I would recommend either removing the term from the paper entirely or providing a definition for it.

Thank you for catching this. We agree with you and have removed all remaining mentions of the word stereotype from the manuscript.

3. I like the revisions the authors made related to my previous comment on an exclusive focus on the underrepresentation of women (instead of focusing on gender bias more broadly, which would involve paying similar attention to instances of the underrepresentation of men, e.g., in female-stereotypical occupations and roles). But I think more re-writing could be done in this area for the paper to be most impactful. There is a growing literature calling attention to the stark gender imbalance in female-dominated occupations, which tends to be more extreme than in male-dominated occupations, but which garners considerably less attention from researchers and the public (see for example Block et al., 2018, Croft et al., 2015, Croft et al., 2021). For example, on p. 5, the first full paragraph deals exclusively with the representation of women, but it omits the fact that, when searching for female-dominated occupations, the pattern of gender bias in images would likely flip. As Figure 2A shows, although the purple area on the right tail end is larger than the purple area on the left tail, both extremes show gender bias and it would be best if this parallelism were more strongly emphasized throughout the main text.

We are greatly sympathetic to this comment and the very useful references on this topic provided by R3. We have made minor revisions throughout our manuscript to make sure that these implications for men are also captured in our framing. To this end, we have incorporated several of the references provided by R3 (specifically, Croft et al., 2015 & Block et al., 2019). Below are two examples of edits we have made to preserve this message (in bold).

From page 1 (first introductory paragraph):

“We find that online images exhibit significantly stronger gender bias for both **female- and male-typed categories** than online text.”

From the discussion:

“This problem is expected not only to impact the well-being, social status, and economic opportunities for women, who are systematically underrepresented in online images according to our measures, but also **men in highly female-typed categories such as care-oriented or community-oriented occupations** [28,29].”

References cited:

- [28] Alyssa Croft, Toni Schmader, and Katharina Block. An underexamined inequality: Cultural and psychological barriers to men’s engagement with communal roles. *Personality and Social Psychology Review*, 19(4):343–370, 2015.
- [29] Katharina Block, Alyssa Croft, Lucy De Souza, and Toni Schmader. Do people care if men don’t care about caring? The asymmetry in support for changing gender roles. *Journal of Experimental Social Psychology*, 83:112–131, 2019.

Reply to Reviewer 4 (R4)

I continue to be highly impressed by this work. It is clearly an outstanding undertaking, with very interesting results. It has a good potential to captivate a wide range of audiences. Naturally, I have a few remaining comments that the authors might like to consider, in the case of further revision, although I consider only the first of them critical.

We are grateful for R4’s enthusiastic support of this study!

1. The critical issue pertains to the results of the experiment, reported in the final paragraph of Page 8. The authors report that “There was no significant difference between participants’ implicit bias in the text and control condition ($p = .19$), yet participants in the image condition exhibited significantly stronger implicit bias than those in the control condition ($p < .001$)”. Isn’t the comparison between text and images the critical comparison? The pre-registration was rather loose, which is a (non-critical) weakness in this work, but it did indicate: “We will use paired, non-parametric between-subject tests (e.g. the Wilcoxon test) to evaluate whether the strength of the gender associations (and implicit stereotypes) reported in the image condition are stronger than those reported in the text condition and the control condition.” These comparisons were reported for the self-report, but not for the IAT. To be clear, this is critical to fix, but I do not consider the actual result critical. In my view, in the present case, the IAT is not much different than the self-report: there were statistical associations between gender and occupation in the content collected by the participants, and those associations were registered in people’s memory, and therefore reported directly by the participants and influenced performance in the IAT. Therefore, the self-report results are interesting and important enough, even if the IAT results were not always clear-cut. Still, it would be important to report the same comparisons for both direct and indirect measures, preferably the pre-registered comparisons.

Thank you for requesting this helpful clarification. We are happy to report the results of all pre-registered statistical comparisons between conditions. We have now added the following revision (in bold) which reports the results of the statistical test comparing the image and text condition directly in terms of participants' resulting *D* score.

“There was no significant difference between participants' implicit bias in the text and control condition ($p = .19$), yet participants in the image condition exhibited significantly stronger implicit bias than those in the control condition ($p < .001$); the difference in implicit bias between the image and text condition did not reach conventional levels of statistical significance ($p = .09$)...”

2. The best possible contribution of this research to science, beyond any specific finding, is the vast amount of data collected, processed, and analyzed. One can teach a whole university course about this data, and even only about how the authors analyzed it. Further, it is obvious that researchers could conduct tons of secondary research with these datasets (including the experiment, which was an unusual undertaking). I strongly recommend, implore and encourage the authors to provide immediate access to any data and code that they can share openly, and a less immediate access (e.g., through request forms) to data that may not be shared freely online (e.g., due to copyright restrictions). I believe that dozens of research studies could make use of that data. Notably, many researchers would be interested in the data and materials of the experiment. There are a few particularly strong aspects in that experiment, in comparison to the typical experiment: the manipulation was quite elaborate and strong (yet also ecological valid), there was more than one control condition, gender bias was measured again a few days after the induction. Data from such a strong experiment would be of interest to many, so, hopefully, it will be shared openly.

We are grateful that R4 shares our enthusiasm for the richness of the data we have collected. We plan to make all of the raw meta-data used in our analyses (including links to the google images and Wikipedia images downloaded) immediately available upon publication. They are currently available via the public github associated with this project, see here: www.github.com/drguilbe/ImgVSText. In terms of the raw images themselves, all of the Wikipedia images we have classified will be made publicly available (since all of these are in the public domain). We also plan to give access to all of the raw images from Google either through a public repository or a request form (pending confirmation around copyright restrictions). We will also make all of the code related to the analysis in the manuscript and supplementary appendix publicly available, which will replicate all of our analyses from the raw metadata (including both our observational and experimental datasets). In addition, we will include the code for extracting and analyzing gender dimensions in word embedding models, along with the code for retraining our own word2vec model (which we have added in reply to comment 3 from R1; see Fig. S7).

3. I think that Figure 1B was an important addition. The issue that bothered me the most in my previous review was that it was not easy to make sense of the two different biases reported by the authors: one was that the absolute gender bias was stronger in the image condition than in the text condition, and the other was that males were over-represented (i.e., were associated with more categories). Over-representation of males could have reflected gender biases that are quite different than those that are typically the focus of political and scientific attention. For example, perhaps “nurse” was more strongly related to Male than to Female in images than in text. That would have been a very different finding than what Figure 1B suggests.

We fully agree with this interpretation and are grateful for the Reviewers' invaluable recommendations in helping us to refine the clarity and impact of our analyses.

4. Figure 1B is a big step toward explaining that the over-representation of men in image content is not the reason (or not the whole reason) for the larger gender bias in image content. However, there might be a concern about that plot, related to how categories were chosen to each separate bar in that plot. If I understand correctly, the Male Text categories were selected as those that showed a stronger association with women in the texts, the Male Images categories were selected as those that showed a stronger association with men in the image content, and the same is true for the Female bars. It means that the categories included in the data of each bar, were probably different. For example, perhaps "nurse" was included in the "male" white bar (the second from the top) and in the "female" grey bar (the third from the top). In that case, although Images content showed a different bias than the stereotype, the plot would (wrongly) seem to suggest that gender stereotypes were reflected more strongly in Images content than textual content. It might be helpful to show the same plot in the supplemental materials four times: once with the categories separated (for both text and image associations) by the image associations (i.e., only the "image" association would determine whether a category is "female" or "male"), once separated by the text associations, once by Human Judgments, and once by Census. Separating the categories by Human Judgment would allow seeing whether Images content amplifies gender stereotypes, in comparison to text content. Separating the categories by the Census data would allow seeing whether Images content is biased, in comparison to the gender-category associations in reality. That is an important piece of the results that is still eluding me.

This is an excellent suggestion! We have included this recommended robustness test in the supplementary material. We carefully considered your suggestion and have implemented it with an update in our approach to best align with the spirit of your recommendation. We encountered a theoretical challenge in determining the ground truth for gender associations across modalities (text, image, human judgments, or census). For instance, people's self-aware beliefs as reported via the gender association slider may not consistently reflect large-scale patterns of co-occurrence in the verbal description of occupations. This inconsistency can be attributed to factors like social desirability bias, which can lead people's self-reported gender associations to downplay their underlying gender biases. Instead, to ensure a robust and parsimonious approach to this test, we propose to combine multiple modalities to obtain the most reliable and consistent gender associations for different categories. Our approach involves identifying categories with the same gender associations in images, texts, and human judgments. We do not factor by census associations in this analysis, first, because people's beliefs often do not track census representation, and second, because we are interested in people's beliefs about the gender of social categories broadly, not just occupations. This step allows us to identify all female (male) categories as those which carry female (male) associations in images, texts, and human judgments.

With this method, we then compare the textual and image representations for these robustly identified gender associations, holding the social categories constant in each condition. We then replicate the results in Fig. 1B while comparing the strength of gender associations between images and texts for these robustly identified gender associations. Using this method, our results replicate precisely as expected. We copy this newly added robustness test and relevant statistical details below.

“Replication using only Categories that have the Same Gender Association across All Modalities when Comparing Images and Text.

Here, we test whether the main results in figure 1B hold while studying only those categories which have consistent female (male) associations across all modalities. We then compare the strength of gender associations between image and text, specifically for these consistently gender-skewed categories. We ensure that the specific categories in each condition remain constant throughout the analysis. Specifically, we compare them to the same robustly estimated female-skewed and male-skewed categories. To achieve this, we first identified all female (male) categories based on their associations in images, texts, and human judgments. We then replicate the analysis presented in Fig. 1B while focusing on these categories with robustly identified gender associations. We do not factor by census associations in this analysis, first, because people’s beliefs often do not track census representation, and second, because we are interested in people’s beliefs about the gender of social categories broadly, not just occupations. This method yielded 1,472 social categories, comprising 1,281 consistently male-skewed categories and 191 consistently female-skewed categories. We note that this asymmetry is partly due to the general bias toward male over-representation across categories in images, text, and human judgments. Since we hold the female and male categories constant across these measures in this analysis, all statistical tests to follow assume comparisons that are paired at the category level.

Fig. S3. The strength of gender association according to these online images and texts for categories consistently identified as either female- or male-skewed by each measure. The female (male) categories shown are those categories that were associated with women (men) in images, text, and human judgments. The gender associations for the same female (male) skewed categories are examined, paired at the category level. **, $p < .01$; ***, $p < .001$.

Figure S3 shows the results of this analysis. We find that images present a significantly stronger level of gender bias for both female- and male-skewed categories. Specifically, when we compare the textual and image representations of these 191 strongly female-skewed categories, we find that texts indicate an average gender association of -0.43 while images present a significantly higher gender association of -0.49, marking a significant 0.06 increase in absolute magnitude ($p = .003$, Student T-test, Two-tailed, with paired comparisons at the category level). Similarly, when we contrast textual and image representations of the 1,281 strongly male-

skewed categories, we find that texts indicate an average gender association of 0.23, whereas images indicate an average gender association of 0.46. This represents a two-fold increase in the absolute strength of male representation ($p < .001$, Student T-test, Two-tailed, with paired comparisons at the category level). In this way, we find not only that images present a significantly stronger gender bias for both female and male categories that are strongly gendered across modalities, but we also find that this effect of images is particularly strong in the context of amplifying male gender bias. This provides additional insight into the role that images play in amplifying male representation across categories.”

5. Still related to the benefit of Figure 1B, it might be informative to see a version of Figure 2C that is separated to “female” and “male” categories. Currently, it seems like the Text associations might be more biased than the Images associations because they seem farther from the actual gender distribution (the Census data). This might reflect a larger bias in feminine categories. For example, perhaps “nurse” is more female-biased in Text than in Census and Images. That may reflect a stronger problem in Text than Images. If you break those bars to “female” and “male” categories (e.g., by Human Judgments), we might learn more about the pattern of the results. Probably, Text would be the least gender biased for both Male and Female categories, in comparison to Census and Images, reaffirming the authors’ current conclusions from this plot.

We are grateful for this suggestion. We have conducted this analysis, which has helped to clarify and deepen the results in figure 2C. We have added these additional analyses into the appendix.

Related to the comments provided in reply to comment 4 from R4, we are wary of treating human judgments as the definitive empirical baseline for stereotypes. This is because subjects’ gender ratings may not always accurately reflect their true underlying stereotypes or the stereotypes diffused via language and images. For example, discrepancies of this kind can be caused by social desirability bias, where participants tend to provide answers they believe the researchers consider appropriate. Instead, we propose a more robust approach to address this question. We suggest categorizing occupations into female-typed and male-typed occupations based on the predominant gender for each occupation according to census data. We then split the occupations along these lines and compare the gender associations of female- and male-typed occupations separately using our text and image measures. This holds constant the female and male typed categories being compared. This allows us to ask: when an occupation is female- (male-) typed according to the census, how gender-skewed are the representations of this occupation in online texts and images?

In this specific analysis, we find that text is *not* more biased than images. On the one hand, texts skew significantly more female for female-typed occupations compared to images. Yet on the other hand, images skew significantly more male for male-typed occupations compared to texts. The extent to which images are male-skewed for male-typed occupations is several fold greater than the extent to which texts are female-skewed for female-typed occupations. In sum, images exhibit greater statistical bias overall.

Furthermore, if we compare the extent to which texts are skewed female for female occupations and skewed male for male occupations, we do not see a significant bias toward the representation of a particular gender in text across occupations. However, in the case of images, we see that the extent to which they are skewed male for male occupations is significantly greater than the extent to which they are skewed female for female occupations, revealing a bias toward male

representation. This analysis reveals that this difference between images and text is largely driven by the overrepresentation of men in images depicting female typed occupations, which aligns with our overall argument.

This analysis shows that the gender skew of both female and male occupations is significantly muted in the text modality, as predicted by our theory. Crucially, we find that this muted effect is not biased in the text modality toward either male or female occupations; instead, across occupations, the aggregate gender skew according to the text modality is neutral, with no discernible bias toward either male or female representation. This provides statistical evidence for a bias toward the overrepresentation of men in the image modality relative to both ground-truth census data and textual representations of occupations. Given these findings, we chose to maintain the original presentation of figure 2C in the main text because it more directly and simply captures the overall gender skew of occupations according to text and images, relative to the underlying empirical gender skew in census data. Yet, we think these additional analyses provide important supplementary insights into understanding these results, so we have added them to our appendix and copied them here for your review.

“In our main text, the results presented in figure 2C focus on the overall gender skew of occupations according to text and images. This comparison is made relative to the underlying empirical skew in the census data for the occupations available in our dataset. Figure 2C shows that for the occupations in our dataset, the census data indicates a significantly overall male skew in the gender distribution of occupations. Yet, if we examine the skew of these occupations according to the image measure, we observe that the image representations of these same occupations are more biased toward male representation than the census data. Conversely, textual representations of the same occupations show no underlying gender bias, despite the clear male skew in the census data. An important question arises: are these results primarily a consequence of the over-representation of men in the image modality (consistent with our main argument)? Alternatively, could they be driven by the over-representation of women in the textual modality, or perhaps some combination of both factors? To address this question and offer further clarity on the statistical patterns underlying these findings, we build upon the analyses presented in our manuscript.

Fig. S4. The gender association of 687 occupations according to three sources (i) textual patterns in Google News (green), (ii) the empirical distribution of gender in the 2019 US

census Bureau of Labor Statistics (grey), and (iii) Google Images (purple). These associations are categorized into whether a specific category is male-skewed or female-skewed according to the census data. Error bars display 95% confidence intervals.

Figure S4 presents the same outcomes as Figure 2C, while comparing the image, text, and census measures separately for occupations categorized as female-skewed or male-skewed occupations according to the census data. For both male- and female-skewed occupations in the census, the absolute strength of gender association is significantly stronger in the census compared to both the text and image modalities ($p < .001$, Student T-test, Two-tailed). Yet, our main interest is in comparing the image and text measures to discern whether they are skewed toward female or male representation relative to each other and the census.

On the one hand, texts skew significantly more female for female-typed occupations compared to images ($p < .001$, Student T-test, Two-tailed). Yet on the other hand, images skew significantly more male for male-typed occupations compared to texts ($p < .001$, Student T-test, Two-tailed). The extent to which images are male-skewed for male-typed occupations is several fold greater than the extent to which texts are female-skewed for female-typed occupations ($p < .001$, Student T-test, Two-tailed). In sum, images exhibit greater statistical bias overall, and in the direction of male over-representation. Indeed, we find that images exhibit significant male skew for male-skewed occupations in the census ($p < .001$), whereas images do not exhibit a significant female skew for female-skewed occupations in the census ($p = .41$), (Student T-test, Two-tailed). This finding indicates that the observed amplification toward male representation in the image modality compared to the census (as presented in Figure 2C) is driven by the over-representation of men in images depicting female-skewed occupations (according to the census), consistent with our main argument.

Relatedly, Figure S4 shows that the gender skew of both female and male occupations in the census is significantly muted in the text modality, consistent with our theory. Crucially, we find that this muted effect is not biased in the text modality toward either male or female occupations; instead, across occupations, the aggregate gender skew according to the text modality is neutral, with no discernible bias toward either male or female representation. In this way, we provide statistical evidence for a bias toward the over-representation of men in the image modality relative to both ground truth census data and textual representations of occupations.”

6. In p. 10, the authors wrote “we find that images differ from text not only in the prevalence of gender bias they contain, but also in their ability to prime gender bias in people’s beliefs. Participants who uploaded gendered images reported significantly stronger gender bias compared to participants who uploaded gendered textual descriptions of the same occupations ($p < .0001$, Student T-test, Two-tailed; Fig. S32).” It is not clear whether that effect is due to a larger impact for imagery (or something else inherent in images vs. text) or simply because even across those categories, the image content was more gender biased than the textual content. A possibly more appropriate analysis for the former claim would control for the content’s bias and test whether an effect of the content type (image vs. text) remains.

This is another excellent suggestion. We have conducted this analysis and it further strengthens our original finding. We have added this analysis to our appendix (copied below). We also provide the revised section of the manuscript which references this supplementary analysis. We have made minor adjustments to the wording to better highlight the causal limitations of this analysis.

From the manuscript (revisions in bold):

“Importantly, we find evidence suggesting that images differ from text not only in the prevalence of gender bias they contain, but also in their ability to prime gender bias in people's beliefs. Participants who uploaded gendered images reported significantly stronger gender bias compared to those who uploaded gendered textual descriptions of the same occupations ($p < .0001$, Student T-test, Two-tailed; Fig. S33). **This holds even when controlling for the level of gender bias in the distribution of images and texts to which participants were exposed (Table S13).** Thus, even when gender was salient in both text and images, exposure to images led to stronger bias in people's beliefs.”

From the supplementary appendix:

“A limitation of the analysis strategy above is that we are unable to distinguish the priming effect of the specific descriptions uploaded by participants from the priming effect of the content that participants were exposed to prior to selecting content to upload. To address this limitation, we conduct an additional analysis which tests whether the main hypothesized priming effect of images - namely, that gendered images more strongly prime gendered responses than gendered textual descriptions - holds when controlling for the level of gender bias associated with each occupation in our observational sample (i.e., within our main sample of Google images and within our word embedding measures of gender association in Google News). In this way, we leverage our observational sample as an estimate of the level of gender bias in the distribution of content that participants were exposed to when searching for descriptions of each occupation (see Fig. S31 for statistical analyses showing that our observational sample of images and text correlates effectively with the distribution of content uploaded by participants in the experiment). The results of this analysis are presented in table S13.

$y = \text{Strength of Participants' Gender Associations}$	
Image Condition [vs. Text]	0.05* (0.02)
Strength of Gender Association in Observational Dataset	-0.03 (0.04)
Occupation Fixed Effects	X
Constant	0.40*** (0.03)
N	2875
R^2	0.14

Standard errors in parentheses (clustered by participant)
* $p < 0.05$, ** $p < 0.01$, *** $p < 0.001$

Table S13. An OLS regression predicting the absolute strength of participants' gender associations for the occupations in our experiment, while controlling for experimental condition (the main treatment effect) and the level of gender bias associated with each occupation in our observational sample (i.e., within our main sample of Google images and within our word embedding measures of gender association in Google News). This model includes fixed effects by occupation and clusters standard errors at the participant level. Data from the control condition is excluded from this analysis because in this condition the

connection is severed between the observational sample and the occupations for which participants indicated their associated gender.

The model in table S13 predicts participants' absolute strength of gender associations, as a function of experimental condition, controlling for the absolute strength of gender bias associated with each occupation in our observational sample. This model also includes fixed effects by occupation and clusters standard errors at the participant level. Participant data from the control condition is excluded from this model because, in this condition, the connection is severed between the content participants encountered online and the occupations for which participants indicated their associated gender. We find that, even when controlling for the level of bias in the content distribution in images and text, being randomized to encounter and upload image content as opposed to textual content leads to a significant increase in the absolute strength of participants' gender associations ($\beta[\text{IMAGE}] = 0.05$, $\text{SE} = 0.02$, $p < .05$). This analysis provides further evidence that, holding constant the expected level of gender bias in the content distribution participants encountered, exposure to images rather than textual descriptions of occupations leads to significantly stronger gender associations. This finding is compatible with a priming mechanism."

7. Figure 1C was described as "The gender associations for a sample of occupations according to these online images and texts." This is of little informative quality without knowing how those occupations were sampled. With thousands of occupations one can probably even find a sample that would show the opposite results of those found across the whole sample.

We have added a clarification to the figure caption to explain that the subset of categories in Figure 1C were manually selected to give the reader a clear qualitative sense of the kinds of categories evaluated and the kinds of gender biases measured. It is not intended as a representative sample. The main results and statistical trends are more rigorously and appropriately captured by our other plots and analyses, which measure the gender associations for all available categories. We are happy to further iterate on this plot or extend it for the supplementary materials if recommended by the review team. The updated caption to Figure 1 is copied below:

"Fig. 1. (A) The correlation between gender associations in images from Google Images and texts from Google News for 2,986 social categories, organized by deciles. Our image-based measure captures the frequency of female and male faces associated with each category in Google Images (-1 means 100% female; 1 means 100% male). Our text-based measure captures the frequency at which each category is associated with male or female terms in the Google News corpus (-1 means 100% female; 1 means 100% male; measure is min-max normalized, *SOM*). Error bars display 95% confidence intervals. ***, $p < .001$. (B) The strength of gender association according to these online images and texts for categories identified as female- or male-skewed by each measure. (C) The gender associations for a sample of occupations according to these online images and texts; **this sample was manually selected to highlight the kinds of social categories and gender biases examined.**"

8. I think that Figure 1C is missing a legend.

The intended legend for Fig. 1C was present in our submission, but we recognized it may not have been appropriately highlighted. In the revised Fig. 1C, we have added a box around the legend so it is more visually salient.

9. In Figure 2B, I was not sure what the x-axis numbers meant. Why did they run from 0 to 25? Was it arbitrary (e.g., bin number) or meaningful?

The x-axis for Fig. 2B reflects an arbitrary bin number. The bins represent evenly spaced subsamples of the data (akin to quartiles or deciles), as specified in the caption to Fig. 2. This binning number was selected because it is visually illustrative, though all statistical analyses associated with the results are calculated on the raw data and are not dependent on the binning. Moreover, the visual results are robust to varying the number of bins. We are happy to update this or add supplementary plots if R4 deems this to be important for clarifying the results.

10. Note that it is probably unusual to spell the IAT's D score "dscore".

Thank you for catching this. We have corrected this to "D score" throughout the manuscript and supplementary material.

Reviewer Reports on the Second Revision:

Referees' comments:

Referee #1 (Remarks to the Author):

I appreciate the comprehensive research conducted by the authors in their rebuttal. The rebuttal is quite informative. The new title now precisely reflects the paper's contributions. However, my primary concern remains unaddressed. As the authors note, there is still an open research question regarding the quantification of information conveyed by text as compared to images. I can think of a few straightforward human subject experiments that might yield preliminary findings in this direction. Nevertheless, I acknowledge that this is a complex problem. The remaining comments I had provided have since been addressed.

Referee #2 (Remarks to the Author):

I have re-read this paper with much interest, and I congratulate the authors once again with the manuscript -- this is the third time I read the manuscript, and in each round it improved significantly. The responses on my prior set of comments all seem well thought out, and my concerns are sufficiently addressed. In particular, I applaud the authors for the change in framing in this round: the shift in focus and the move away from the framing-as-a-paradox increases the relevance of the study and much better connects the premise to the analytical setup.

My prior concerns seem sufficiently addressed and I have not much to add anymore. One tiny final comment is to perhaps take a good look at some of the captions and the axes labels of the figures, as those were sometimes slightly difficult to fully understand. It was a pleasure to see this paper develop to where it is now.

Referee #3 (Remarks to the Author):

I continue to be enthusiastic about this work and impressed with the huge undertaking by the authors. My previous concerns have been adequately addressed. However, I could not find any mention of the Gwet's AC metric in the main manuscript. I think this information is more compelling and valuable than the simple discussion of percent agreement, and it would make sense to report it in the main text. Other than this, I wish the authors good luck as they continue with this research.

Referee #4 (Remarks to the Author):

The revision has further strengthened this excellent contribution. My only recommendation is that the authors continue to improve the accessibility of the data and analysis scripts with extensive readme file(s) that provide as much information and examples as possible about the databases they collected and uploaded.

Author Rebuttals to Second Revision:

Reply to Reviewers

Referees' comments:

Referee #1 (Remarks to the Author):

I appreciate the comprehensive research conducted by the authors in their rebuttal. The rebuttal is quite informative. The new title now precisely reflects the paper's contributions. However, my primary concern remains unaddressed. As the authors note, there is still an open research question regarding the quantification of information conveyed by text as compared to images. I can think of a few straightforward human subject experiments that might yield preliminary findings in this direction. Nevertheless, I acknowledge that this is a complex problem. The remaining comments I had provided have since been addressed.

We thank Reviewer 1 for their enormously helpful comments throughout this review process. Reviewer 1's insightful comments have inspired several ideas for how to push forward this research frontier. Our paper has substantially improved as a result of our exchange. Thank you for your careful thought and attention in helping to bring out the best in our research. We really value your time and expertise, thank you!

Referee #2 (Remarks to the Author):

I have re-read this paper with much interest, and I congratulate the authors once again with the manuscript -- this is the third time I read the manuscript, and in each round it improved significantly. The responses on my prior set of comments all seem well thought out, and my concerns are sufficiently addressed. In particular, I applaud the authors for the change in framing in this round: the shift in focus and the move away from the framing-as-a-paradox increases the relevance of the study and much better connects the premise to the analytical setup.

We are delighted to learn that our revision has effectively addressed Reviewer 2's remaining concerns. We agree that our revised framing is a major improvement upon our initial submission, and we are enormously grateful to Reviewer 2's insightful comments which spurred a much deeper understanding and framing of our study.

My prior concerns seem sufficiently addressed and I have not much to add anymore. One tiny final comment is to perhaps take a good look at some of the captions and the axes labels of the figures, as those were sometimes slightly difficult to fully understand. It was a pleasure to see this paper develop to where it is now.

We have reviewed all of our figure captions and axes, making minor tweaks to enhance their clarity. Thank you, once again, for sharing such careful and insightful analyses throughout this review process. Your comments really helped to bring out the best in our research.

Referee #3 (Remarks to the Author):

I continue to be enthusiastic about this work and impressed with the huge undertaking by the authors. My previous concerns have been adequately addressed. However, I could not find any mention of the Gwet's AC metric in the main manuscript. I think this information is more compelling and valuable than the simple discussion of percent agreement, and it would make

sense to report it in the main text. Other than this, I wish the authors good luck as they continue with this research.

We are delighted about Reviewer 3's enthusiasm regarding our latest revision. In the manuscript, we now explicitly reference the Gwet's AC metric with the following sentence (revision highlighted in bold):

“Codiers reached unanimous agreement in their gender classifications for 91% of images. **A standard chance-corrected measure of classification agreement indicates satisfactory intercoder reliability in our sample (Gwet's AC = 0.48; SOM).**”

We want to thank Reviewer 3, once again, for their enormously helpful and insightful comments throughout the review process. Our paper has majorly improved as a result of your incisive analyses and interesting suggestions. Thank you for your careful thought and attention in helping to bring out the best in our research.

Referee #4 (Remarks to the Author):

The revision has further strengthened this excellent contribution. My only recommendation is that the authors continue to improve the accessibility of the data and analysis scripts with extensive readme file(s) that provide as much information and examples as possible about the databases they collected and uploaded.

We are delighted to hear that Reviewer 4 continues to enthusiastically approve of our study. Your thoughtful comments were enormously impactful in helping to bring out the best in our research, and your insightful analyses have inspired several ideas for interesting next projects that we are eager to explore. Thank you for embodying the ideals of reviewership, *par excellence*. In terms of spurring new projects, we are excited about the potential for other research teams to leverage our data and methods, which are now publicly available. We have further updated our project GitHub page so that it now includes all of the raw images we classified both from Google and Wikipedia. Each image is associated with a unique “image_id”, which can be directly matched to the classification metadata also provided in .csv format via our project GitHub. We have worked to provide detailed readmes for how to understand, download, and analyze the data, and we hope that our replication analysis script will also be helpful for interested parties in this regard. We have also made publicly available our code for (i) building our own word2vec model from scratch using any text data, and (ii) for extracting a gender dimension (and indeed, any sociodemographic or semantic dimension of interest) from any given word embedding model. We are excited to see the potential research projects that these materials can enable beyond our own team's research program. Thank you!